# ASPEST: Bridging the Gap Between Active Learning and Selective Prediction

**Jiefeng Chen**\*  _jiefeng@cs.wisc.edu_
_University of Wisconsin-Madison_

**Jinsung Yoon**  _jinsungyoon@google.com_
_Google_

**Sayna Ebrahimi**  _saynae@google.com_
_Google_

**Sercan Ö. Arık**  _soarik@google.com_
_Google_

**Somesh Jha**  _jha@cs.wisc.edu_
_University of Wisconsin-Madison_
_Google_

**Tomas Pfister**  _tpfister@google.com_
_Google_

**Reviewed on OpenReview:** _https://openreview.net/forum?id=3nprbNR3HB_

## Abstract

Selective prediction aims to learn a reliable model that abstains from making predictions when uncertain. These predictions can then be deferred to humans for further evaluation. As an everlasting challenge for machine learning, in many real-world scenarios, the distribution of test data is different from the training data. This results in more inaccurate predictions, and often increased dependence on humans, which can be difficult and expensive. Active learning aims to lower the overall labeling effort, and hence human dependence, by querying the most informative examples. Selective prediction and active learning have been approached from different angles, with the connection between them missing. In this work, we introduce a new learning paradigm, _active selective prediction_, which aims to query more informative samples from the shifted target domain while increasing accuracy and coverage. For this new paradigm, we propose a simple yet effective approach, ASPEST, that utilizes ensembles of model snapshots with self-training with their aggregated outputs as pseudo labels. Extensive experiments on numerous image, text and structured datasets, which suffer from domain shifts, demonstrate that ASPEST can significantly outperform prior work on selective prediction and active learning (e.g. on the MNIST→SVHN benchmark with the labeling budget of 100, ASPEST improves the AUACC metric from 79.36% to 88.84%) and achieves more optimal utilization of humans in the loop.

## 1 Introduction

Deep Neural Networks (DNNs) have shown notable success in many applications that require complex understanding of input data (He et al., 2016a; Devlin et al., 2018; Hannun et al., 2014), including the

---

\*Work done during internship at Google.
Our code is available at: `https://github.com/google-research/google-research/tree/master/active_selective_prediction`.

ones that involve high-stakes decision making (Yang, 2020). For safe deployment of DNNs in high-stakes applications, it is typically required to allow them to abstain from their predictions that are likely to be wrong, and ask humans for assistance (a task known as selective prediction) (El-Yaniv et al., 2010; Geifman & El-Yaniv, 2017). Although selective prediction can render the predictions more reliable, it does so at the cost of human interventions. For example, if a model achieves 80% accuracy on the test data, an ideal selective prediction algorithm should reject those 20% misclassified samples and send them to a human for review.

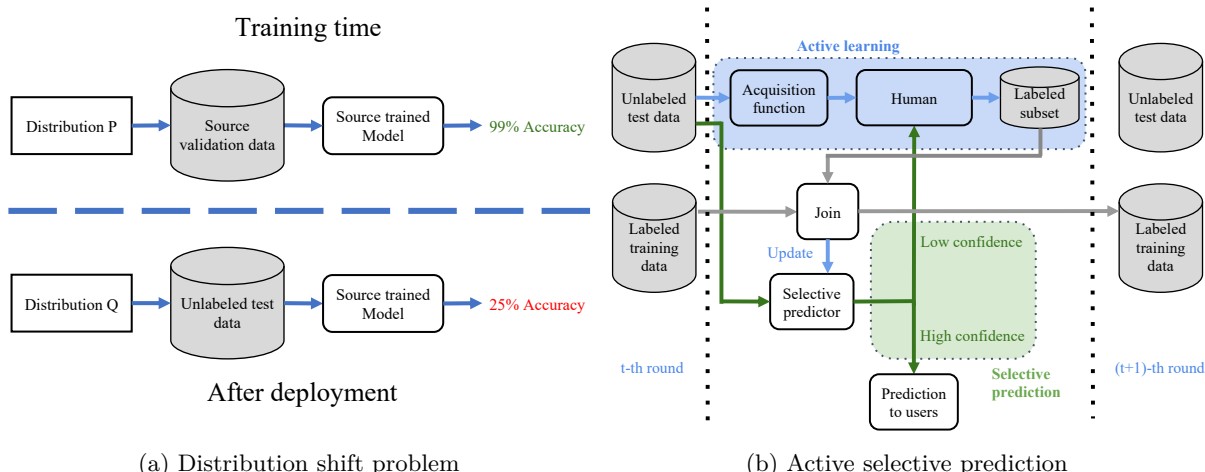

(a) Distribution shift problem       (b) Active selective prediction

Figure 1: Illustration of the distribution shift problem and the proposed *active selective prediction* solution. Under distribution shift, the model trained on the source training dataset will suffer a large performance degradation on the unlabeled test dataset. We propose to use active selective prediction to solve this problem, where active learning is used to improve selective prediction under distribution shift. The selective predictor shown in Fig. (b) is built upon the source-trained model depicted in Fig. (a). In this setting, active learning selects a small subset of data for labeling which are used to improve selective prediction on the remaining unlabeled test data. This yields more reliable predictions and more optimized use of humans in the loop.

Distribution shift can significantly exacerbate the need for such human intervention. The success of DNNs often relies on the assumption that both training and test data are sampled independently and identically from the same distribution. In practice, this assumption may not hold and can degrade the performance on the test domain (Barbu et al., 2019; Koh et al., 2021). For example, for satellite imaging applications, images taken in different years can vary drastically due to weather, light, and climate conditions (Koh et al., 2021). Existing selective prediction methods usually rely on model confidence to reject inputs (Geifman & El-Yaniv, 2017). However, it has been observed that model confidence can be poorly calibrated, especially with distribution shifts (Ovadia et al., 2019). The selective classifier might end up accepting many misclassified test inputs, making the predictions unreliable. Thus, selective prediction might yield an accuracy below the desired target performance, or obtain a low coverage, necessitating significant human intervention.

To improve the performance of selective prediction, one idea is to rely on active learning and to have humans label a small subset of selected test data. The correct labels provided by humans can then be used to improve the accuracy and coverage (see Sec. 3.2) of selective prediction on the remaining unlabeled test data, thus reducing the need for subsequent human labeling efforts. In separate forms, selective prediction (Geifman & El-Yaniv, 2017; 2019) and active learning (Settles, 2009) have been studied extensively, however, to the best of our knowledge, this paper is first to propose performing active learning to improve selective prediction jointly, with the focus on the major real-world challenge of distribution shifts. Active domain adaptation (Su et al., 2020; Fu et al., 2021; Prabhu et al., 2021) is one area close to this setting, however, it does not consider selective prediction. In selective prediction, not only does a classifier need to be learned, but a selection scoring function also needs to be constructed for rejecting misclassified inputs. Thus, going beyond conventional active learning methods that focus on selecting examples for labeling to improve the accuracy,

| Machine Learning Paradigm | Accuracy | Coverage | Acquisition Function | Selectivity | Adapt $f$ | Adapt $g$ | Target Metric |
|---|---|---|---|---|---|---|---|
| Selective Prediction | High | Low | None | If $g(x) \geq \tau$, output $f(x)$; otherwise, output $\perp$ | ✗ | ✗ | $cov\|acc \geq t_a$, $acc\|cov \geq t_c$ or AUACC |
| Active Learning | Medium | 100% | $a(B, f)$ | Always output $f(x)$ | ✓ | ✗ | $acc$ |
| Active Selective Prediction | High | High | $a(B, f, g)$ | If $g(x) \geq \tau$, output $f(x)$; otherwise, output $\perp$ | ✓ | ✓ | $cov\|acc \geq t_a$, $acc\|cov \geq t_c$ or AUACC |

Table 1: Comparing different machine learning paradigms. With distribution shift and a small labeling budget, active learning can achieve 100% coverage (predictions on all test data), but it fails to achieve the target accuracy. Selective prediction can achieve the target accuracy, but the coverage is low, which results in significant human intervention. Active selective prediction achieves the target accuracy with much higher coverage, significantly reducing human labeling effort. $B$: selected batch for labeling. $f$: classifier and $g$: selection scoring function. $acc$: accuracy and $cov$: coverage.

we propose to also use those selected labeled examples to improve the selection scoring function. The optimal acquisition function (used to select examples for labeling) for this new setting is different compared to those in traditional active learning – e.g. if a confidence-based selection scoring function is employed, the selected labeled samples should have the goal of improving the estimation of that confidence score.

In this paper, we introduce a new machine learning paradigm: active selective prediction under distribution shift (see Fig. 1), which combines selective prediction and active learning to improve accuracy and coverage, and hence use human labeling in a more efficient way. Table 1 shows the differences among selective prediction, active learning and active selective prediction. Active selective prediction is highly important for most real-world deployment scenarios (e.g., the batch prediction scenario where users give a batch of test inputs potentially with distribution shifts, and request predictions from a deployed pre-trained model). To the best of our knowledge, we are the first to formulate and investigate this problem, along with the judiciously chosen evaluation metrics for it (Sec. 3). We also introduce a novel and simple yet effective method, ASPEST, for this active selective prediction problem (Sec. 4). The key components of ASPEST, checkpoint ensembling and self-training, are designed to address the key challenge (i.e., the overconfidence issue) in the active selective prediction problem. On numerous real-world datasets, we show that ASPEST consistently outperforms other baselines proposed for active learning and selective prediction (Sec. 5).

## 2 Related Work

*Selective prediction* (also known as prediction with rejection/deferral options) constitutes a common deployment scenario for DNNs, especially in high-stakes decision making scenarios. In selective prediction, models abstain from yielding outputs if their confidence on the likelihood of correctness is not sufficiently high. Such abstinence usually incurs deferrals to humans and results in additional cost (Mozannar & Sontag, 2020). Increasing the coverage – the ratio of the samples for which the DNN outputs can be reliable – is the fundamental goal (El-Yaniv et al., 2010; Fumera & Roli, 2002; Hellman, 1970; Geifman & El-Yaniv, 2019). Geifman & El-Yaniv (2017) considers selective prediction for DNNs with the 'Softmax Response' method, which applies a carefully selected threshold on the maximal response of the softmax layer to construct the selective classifier. Lakshminarayanan et al. (2017) shows that using deep ensembles can improve predictive uncertainty estimates and thus improve selective prediction. Rabanser et al. (2022) proposes a novel method, NNTD, for selective prediction that utilizes DNN training dynamics by using checkpoints during training. Our proposed method ASPEST also uses checkpoints to construct ensembles for selective prediction. In contrast to NNTD and other aforementioned methods, we combine selective prediction with active learning to improve its data efficiency while considering a holistic perspective of having humans in the loop. This new active selective prediction setup warrants new methods for selective prediction along with active learning.

*Active learning* employs acquisition functions to select unlabeled examples for labeling, and uses these labeled examples to train models to utilize the human labeling budget more effectively while training DNNs (Settles, 2009; Dasgupta, 2011). Commonly-used active learning methods employ acquisition functions by considering uncertainty (Gal et al., 2017; Ducoffe & Precioso, 2018; Beluch et al., 2018) or diversity (Sener & Savarese, 2017; Sinha et al., 2019), or their combination (Ash et al., 2019; Huang et al., 2010). One core challenge for active learning is the "cold start" problem: often the improvement obtained from active learning is less

significant when the amount of labeled data is significantly smaller (Yuan et al., 2020; Hacohen et al., 2022). Moreover, active learning can be particularly challenging under distribution shift (Kirsch et al., 2021; Zhao et al., 2021). Recently, active domain adaptation has been studied, where domain adaptation is combined with active learning (Su et al., 2020; Fu et al., 2021; Prabhu et al., 2021). Different from traditional active learning, active domain adaptation typically adapts a model pre-trained on the labeled source domain to the unlabeled target domain. In our work, we also try to adapt a source trained model to the unlabeled target test set using active learning, while focusing on building a selective classification model and reducing the human labeling effort. More related work are discussed in Appendix A.

## 3 Active Selective Prediction

In this section, we first formulate the active selective prediction problem and then present the proposed evaluation metrics to quantify the efficacy of the methods.

### 3.1 Problem Setup

Let $\mathcal{X}$ be the input space and $\mathcal{Y} = \{1, 2, \ldots, K\}$ the label space.[1] The training data distribution is given as $P_{X,Y}$ and the test data distribution is $Q_{X,Y}$ (both are defined in the space $\mathcal{X} \times \mathcal{Y}$). There might exist distribution shifts such as covariate shifts (i.e., $Q_{X,Y}$ might be different from $P_{X,Y}$). Suppose for each input $\mathbf{x}$, an oracle (e.g., the human annotator) can assign a ground-truth class label $y_x$ to it. Given a classifier $\bar{f} : \mathcal{X} \to \mathcal{Y}$ trained on a source training dataset $\mathcal{D}^{\mathrm{tr}} \sim P_{X,Y}$ ($\sim$ means "sampled from"), and an unlabeled target test dataset $U_X = \{\mathbf{x}_1, \ldots, \mathbf{x}_n\} \sim Q_X$, our goal is to employ $\bar{f}$ to yield reliable predictions on $U_X$ in a human-in-the-loop scenario. Holistically, we consider the two approaches to involve humans via the predictions they provide on the data: (i) selective prediction where uncertain predictions are deferred to humans to maintain a certain accuracy target; and (ii) active learning where a subset of $U_X$ unlabeled samples are selected for humans to improve the model with the extra labeled data to be used at the subsequent iterations. The concept of active selective prediction emerges from the potential synergistic optimization of these two approaches. By judiciously integrating active learning and selective prediction, we aim to minimize the demand for human labeling resources. Active learning in this context is tailored to select a minimal yet impactful set of samples for labeling, which in turn informs the development of a highly accurate selective predictor. This predictor effectively rejects misclassified samples, deferring them for human evaluation. As a result of this optimal integration, the number of samples requiring human intervention is significantly reduced, thereby enhancing the overall efficiency and resource utilization. Since these two approaches to involve humans have different objectives, their joint optimization to best use the human labeling resources is not straightforward.

As an extension of the classifier $f$ (initialized by $\bar{f}$), we propose to employ a selective classifier $f_s$ including a selection scoring function $g : \mathcal{X} \to \mathbb{R}$ to yield reliable predictions on $U_X$. We define the predicted probability of the model $f$ on the $k$-th class as $f(\mathbf{x} \mid k)$. Then, the classifier is $f(\mathbf{x}) = \arg\max_{k \in \mathcal{Y}} f(\mathbf{x} \mid k)$. $g$ can be based on statistical operations on the outputs of $f$ (e.g., $g(\mathbf{x}) = \max_{k \in \mathcal{Y}} f(\mathbf{x} \mid k)$). With $f$ and $g$, the selective prediction model $f_s$ is defined as:

$$f_s(\mathbf{x}; \tau) = \begin{cases} f(\mathbf{x}) & \text{if } g(\mathbf{x}) \geq \tau, \\ \bot & \text{if } g(\mathbf{x}) < \tau \end{cases}, \tag{1}$$

where $\tau$ is a threshold. If $f_s(\mathbf{x}) = \bot$, then the DNN system would defer the predictions to a human in the loop. To improve the overall accuracy to reach the target, such deferrals require manual labeling. To reduce the human labeling cost and improve the accuracy of the selective classifier, we consider labeling a small subset of $U_X$ and adapt the selective classifier $f_s$ on the labeled subset via active learning. The goal is to significantly improve the accuracy and coverage of the selective classifier $f_s$ and thus reduce the total human labeling effort.

Suppose the labeling budget for active learning is $M$ (i.e., $M$ examples from $U_X$ are selected to be labeled by humans). We assume that humans in the loop can provide correct labels. For active learning, we consider

---

[1]In this paper, we focus on the classification problem, although it can be extended to the regression problem.

the transductive learning paradigm (Vapnik, 1998), which assumes all training and test data are observed beforehand and we can make use of the unlabeled test data for learning. Specifically, the active learning is performed on $U_X$ to build the selective classifier $f_s$, with performance evaluation of $f_s$ only on $U_X$. We adapt the source-trained classifier $\bar{f}$ to obtain $f_s$ instead of training $f_s$ from scratch to maintain feasibly-low computational cost.

Let's first consider the single-round setting. Suppose the acquisition function is $a : \mathcal{X}^m \times \mathcal{F} \times \mathcal{G} \to \mathbb{R}$, where $m \in \mathbb{N}^+$, $\mathcal{F}$ is the classifier space and $\mathcal{G}$ is the selection scoring function space. This acquisition function is different from the one used in traditional active learning (Gal et al., 2017) since traditional active learning doesn't have the goal of improving $g$. In the beginning, $f$ is initialized by $\bar{f}$. We then select a batch $B^*$ for labeling by solving the following objective:

$$B^* = \arg \max_{B \subset U_X, |B|=M} a(B, f, g), \tag{2}$$

for which the labels are obtained to get $\tilde{B}^*$. Then, we use $\tilde{B}^*$ to update $f$ and $g$ (e.g., via fine-tuning).

The above can be extended to a multi-round setting. Suppose we have $T$ rounds and the labeling budget for each round is $m = [\frac{M}{T}]$. In the beginning, $f_0$ is initialized by $\bar{f}$. At the $t$-th round, we first select a batch $B_t^*$ for labeling by solving the following objective:

$$B_t^* = \arg \max_{B \subset U_X \setminus (\cup_{l=1}^{t-1} B_l^*), |B|=m} a(B, f_{t-1}, g_{t-1}), \tag{3}$$

for which the labels are obtained to get $\tilde{B}_t^*$. Then we use $\tilde{B}_t^*$ to update $f_{t-1}$ and $g_{t-1}$ to get $f_t$ and $g_t$ (e.g., via fine-tuning the model on $\tilde{B}_t^*$). With multiple-rounds setting, we define $B^* = \cup_{i=1}^T B_i^*$.

## 3.2 Evaluation Metrics

To quantify the efficacy of the methods that optimize human-in-the-loop adaptation and decision making performance, appropriate metrics are needed. The performance of the selective classifier $f_s$ (defined in Eq. (1)) on $U_X$ is evaluated by the accuracy and coverage metrics, which are defined as:

$$acc(f_s, \tau) = \frac{\mathbb{E}_{\mathbf{x} \sim U_X} \mathbb{I}[f(\mathbf{x}) = y_x \wedge g(\mathbf{x}) \geq \tau \wedge \mathbf{x} \notin B^*]}{\mathbb{E}_{\mathbf{x} \sim U_X} \mathbb{I}[g(\mathbf{x}) \geq \tau \wedge \mathbf{x} \notin B^*]} \tag{4}$$

$$cov(f_s, \tau) = \frac{\mathbb{E}_{\mathbf{x} \sim U_X} \mathbb{I}[g(\mathbf{x}) \geq \tau \wedge \mathbf{x} \notin B^*]}{\mathbb{E}_{\mathbf{x} \sim U_X} \mathbb{I}[\mathbf{x} \notin B^*]} \tag{5}$$

We can tune the threshold $\tau$ to achieve a certain coverage. There could be an accuracy-coverage trade-off – as we increase coverage, the accuracy could be lower. We consider the following metrics that are agnostic to the threshold $\tau$: (1) maximum accuracy at a target coverage denoted as $acc|cov \geq t_c$; (2) maximum coverage at a target accuracy denoted as $cov|acc \geq t_a$; (3) Area Under Accuracy-Coverage Curve denoted as AUACC. The definitions of these metrics are given in Appendix B.

The target coverage $t_c$ is tailored to each dataset, acknowledging the varying levels of difficulty inherent to different datasets. Similar to $t_c$, the target accuracy $t_a$ is set according to the dataset's complexity. For more challenging datasets, where achieving high accuracy is difficult, lower target accuracy thresholds are set. Conversely, for datasets where models typically achieve higher accuracy, more stringent accuracy targets are established. By adapting the target accuracy and coverage thresholds to the dataset-specific challenges, we ensure that our evaluation is both rigorous and relevant to each dataset's context. Additionally, we employ the AUACC metric as our primary metric for comparative analysis. This choice is strategic, as it helps standardize our evaluations across a variety of datasets and addresses potential inconsistencies that might arise from solely relying on threshold-based metrics.

## 3.3 Challenges

For active selective prediction, we want to utilize active learning to improve the coverage and accuracy of the selective classifier $f_s$, that consists of a classifier $f$ and a selection scoring function $g$. In contrast to

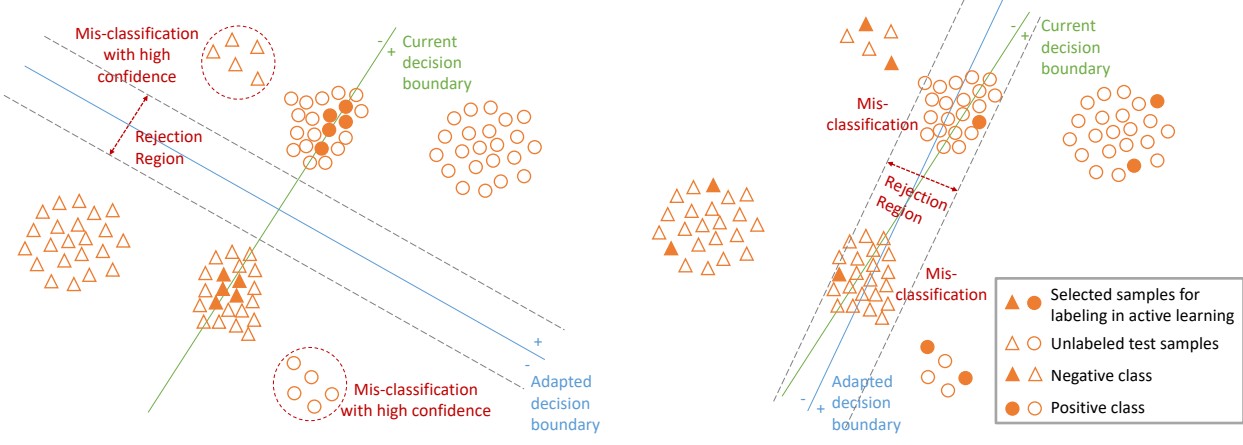

(a) Sample selection based on uncertainty        (b) Sample selection based on diversity

Figure 2: Illustration of the challenges in active selective prediction using a linear model to maximize the margin (distance to the decision boundary) for binary classification. We consider a single-round active learning setup, without the inclusion of already labeled data. The current decision boundary is derived from labeled source training data. Since we focus on assessing performance with respect to the unlabeled test data, where our evaluation metrics are applied, we omit source training data from this illustration. The model confidence is considered to be proportional to the margin (when the margin is larger, the confidence is higher and vice versa). Fig. (a) shows if the samples close to the current decision boundary are selected for labeling (uncertainty-based sample selection), then the adapted model suffers from the overconfidence issue (mis-classification with high confidence), which results in acceptance of some mis-classified points. Fig. (b) shows if diverse samples are selected for labeling (e.g., using the k-Center-Greedy algorithm (Sener & Savarese, 2017)), then the adapted model suffers from low accuracy. This leads to rejection of many points, necessitating significant human intervention.

conventional active learning, which only aims to improve the accuracy of $f$, active selective prediction also aims to improve $g$ so that it can accept those examples where $f$ predicts correctly and reject those where $f$ predicts incorrectly. Especially with distribution shift and a small labeling budget $M$, it can be challenging to train $f$ for high accuracy. Therefore, $g$ is critical in achieving high coverage and accuracy of $f_s$, for which we consider the confidence of $f$ (i.e., the maximum softmax score of $f$) and train $f$ such that its confidence can be used to distinguish correct and incorrect predictions. This might not be achieved easily since it has been observed that $f$ can have overconfident predictions especially under distribution shift (Goodfellow et al., 2014; Hein et al., 2019). Besides, for active learning, typically we select samples for labeling based on uncertainty or diversity. However, in active selective prediction, sample selection based on uncertainty may lead to overconfidence and sample selection based on diversity may lead to low accuracy of $f$, as illustrated in Fig. 2. Our experiments in Appendix F.2 show that these issues indeed exist – the methods based on uncertainty sampling (e.g., SR+Margin) achieve relatively high accuracy, but suffer from the overconfidence issue, while the methods based on diversity sampling (e.g. SR+kCG) don't have the overconfidence issue, but suffer from low accuracy of $f$. Moreover, the hybrid methods based on uncertainty and diversity sampling (SR+CLUE and SR+BADGE) still suffer from the overconfidence issue. To tackle these, we propose a novel method, ASPEST, described next.

## 4 Proposed Method: ASPEST

We propose a novel method Active Selective Prediction using Ensembles and Self-training (ASPEST), which utilizes two key techniques, checkpoint ensembles and self-training, to solve the active selective prediction problem. The key constituents, checkpoint ensembles and self-training, are designed to tackle the fundamental challenges in active selective prediction, with the ideas of selecting samples for labeling based on uncertainty to achieve high accuracy and using checkpoint ensembles and self-training to alleviate overconfi-

dence. We empirically analyze why they can tackle the challenges in Section 5.3. In Appendix D, we analyze the complexity of the ASPEST algorithm.

We first describe how the weights from the intermediate checkpoints during training are used to construct the checkpoint ensemble. Since we have all the test inputs, we don't need to save the checkpoints during training, but just record their outputs on the test set $U_X$. Specifically, we use a $n \times K$ matrix $P$ (recall that $n = |U_X|$ and $K$ is the number of classes) to record the average of the softmax outputs of the checkpoint ensemble and use $N_e$ to record the number of checkpoints in the current checkpoint ensemble. During training, we get a stream of checkpoints (assuming no two checkpoints arrive at the same time), and for each incoming checkpoint $f$, we update $P$ and $N_e$ as:

$$P_{i,k} \leftarrow \frac{1}{N_e + 1}(P_{i,k} \cdot N_e + f(\mathbf{x}_i \mid k)) \quad \text{for } 1 \leq i \leq n \text{ and } 1 \leq k \leq K, \quad N_e \leftarrow N_e + 1. \tag{6}$$

Since it has been observed that an ensemble of DNNs (known as "deep ensembles") usually produces a confidence score that is better calibrated compared to a single DNN (Lakshminarayanan et al., 2017), we consider $f$ to be in the form of deep ensembles and $g$ to be the confidence of the ensemble. Specifically, we continue fine-tuning $N$ models independently via Stochastic Gradient Descent (SGD) with different random seeds (e.g., the randomness can come from different random orders of training batches). At the beginning, we set each model $f_0^j = \bar{f}$ ($j = 1, \ldots, N$), and set $N_e = 0$ and $P = \mathbf{0}_{n \times K}$. Here, we initialize each model $f_0^j$ with the source-trained classifier $\bar{f}$ instead of random initialization, to minimize the computational cost. We fine-tune each model $f_0^j$ on $\mathcal{D}^{\text{tr}}$ for $n_s$ steps via SGD using the following training objective: $\min_{\theta^j} \mathbb{E}_{(\mathbf{x}, y) \in \mathcal{D}^{\text{tr}}} \ell_{CE}(\mathbf{x}, y; \theta^j)$, where $\ell_{CE}$ is the cross-entropy loss and $\theta^j$ is the model parameters of $f_0^j$. For every $c_s$ steps when training each $f_0^j$, we update $P$ and $N_e$ using Eq (6) with the checkpoint model $f_0^j$.

After constructing the initial checkpoint ensemble, we perform a $T$-round active learning process. In each round of active learning, we first select samples for labeling based on the margin of the checkpoint ensemble, then fine-tune the models on the selected labeled test data, and finally perform self-training. We describe the procedure below:

**Sample selection.** In the $t$-th round, we select a batch $B_t$ with a size of $m = \lceil \frac{M}{T} \rceil$ from $U_X$ via:

$$B_t = \arg \max_{B \subset U_X \backslash (\cup_{l=0}^{t-1} B_l), |B|=m} - \sum_{\mathbf{x}_i \in B} S(\mathbf{x}_i) \tag{7}$$

where $B_0 = \emptyset$, $S(\mathbf{x}_i) = P_{i,\hat{y}} - \max_{k \in \mathcal{Y} \backslash \{\hat{y}\}} P_{i,k}$ and $\hat{y} = \arg \max_{k \in \mathcal{Y}} P_{i,k}$. We use an oracle to assign ground-truth labels to the examples in $B_t$ to get $\tilde{B}_t$. Here, we select the test samples for labeling based on the margin of the checkpoint ensemble. The test samples with lower margin should be closer to the decision boundary and they are data points where the ensemble is uncertain about its predictions. Training on those data points can either make the predictions of the ensemble more accurate or make the ensemble have higher confidence on its correct predictions.

**Fine-tuning.** After the sample selection, we reset $N_e$ and $P$ as $N_e = 0$ and $P = \mathbf{0}_{n \times K}$, because we want to remove those checkpoints in the previous rounds with worse performance from the checkpoint ensemble (experiments in Appendix F.9 show that after each round of active learning, the accuracy of the ensemble will significantly improve). We then fine-tune each model $f_{t-1}^j$ ($j = 1, \ldots, N$) independently via SGD with different randomness on the selected labeled test data to get $f_t^j$ using the following training objective:

$$\min_{\theta^j} \mathbb{E}_{(\mathbf{x}, y) \in \cup_{l=1}^t \tilde{B}_l} \ell_{CE}(\mathbf{x}, y; \theta^j) + \lambda \cdot \mathbb{E}_{(\mathbf{x}, y) \in \mathcal{D}^{\text{tr}}} \ell_{CE}(\mathbf{x}, y; \theta^j), \tag{8}$$

where $\theta^j$ is the model parameters of $f_{t-1}^j$ and $\lambda$ is a hyper-parameter. Note that here we use joint training on $\mathcal{D}^{\text{tr}}$ and $\cup_{l=1}^t \tilde{B}_l$ to avoid over-fitting to the small set of labeled test data and prevent the models from forgetting the source training knowledge (see the results in Appendix F.5 for the effect of using joint training and the effect of $\lambda$). For every $c_e$ epoch when fine-tuning each model $f_{t-1}^j$, we update $P$ and $N_e$ using Eq. (6) with the checkpoint model $f_{t-1}^j$.

**Self-training.** After fine-tuning the models on the selected labeled test data and with the checkpoint ensemble, we construct a pseudo-labeled set $R$ via:

$$R = \{(\mathbf{x}_i, [P_{i,1}, \cdots, P_{i,K}]) \mid \mathbf{x}_i \in U_X \wedge (\eta \leq \max_{k \in \mathcal{Y}} P_{i,k} < 1)\}, \tag{9}$$

where $\max_{k \in \mathcal{Y}} P_{i,k}$ is the confidence of the checkpoint ensemble on $\mathbf{x}_i$ and $\eta$ is a threshold (refer to Section 5.2 for the effect of $\eta$). We do not add those test data points with confidence equal to 1 into the pseudo-labeled set because training on those data points cannot change the models much and may even hurt the performance (refer to Appendix F.8 for the justification of such a design). We then perform self-training on the pseudo-labeled set $R$. For computational efficiency, we only apply self-training on a subset of $R$. We construct the subset $R_{\text{sub}}$ by randomly sampling up to $[p \cdot n]$ data points from $R$, where $p \in [0, 1]$. We train each model $f_t^j$ ($j = 1, \ldots, N$) further on the pseudo-labeled subset $R_{\text{sub}}$ via SGD using the following training objective:

$$\min_{\theta^j} \quad \mathbb{E}_{(\mathbf{x},\mathbf{y}) \in R_{\text{sub}}} \quad \ell_{KL}(\mathbf{x}, \mathbf{y}; \theta^j) + \lambda \cdot \mathbb{E}_{(\mathbf{x},y) \in \mathcal{D}^{\text{tr}}} \quad \ell_{CE}(\mathbf{x}, y; \theta^j) \tag{10}$$

where $\ell_{KL}$ is the KL-Divergence loss, which is defined as: $\ell_{KL}(\mathbf{x}, \mathbf{y}; \theta) = \sum_{k=1}^{K} \mathbf{y}_k \cdot \log(\frac{\mathbf{y}_k}{f(\mathbf{x}|k;\theta)})$. We use the KL-Divergence loss with soft pseudo-labels to alleviate the overconfidence issue since the predicted labels might be wrong and training the models on those misclassified pseudo-labeled data using the typical cross entropy loss will cause the overconfidence issue, which hurts selective prediction performance. For every $c_e$ epoch of self-training each model $f_t^j$, we will update $P$ and $N_e$ using Eq (6) with the checkpoint model $f_t^j$. We add checkpoints during self-training into checkpoint ensemble to improve the sample selection in the next round of active learning.

After $T$ rounds active learning, we use the checkpoint ensemble as the final selective classifier: the classifier $f(\mathbf{x}_i) = \arg\max_{k \in \mathcal{Y}} P_{i,k}$ and the selection scoring function $g(\mathbf{x}_i) = \max_{k \in \mathcal{Y}} P_{i,k}$.

# 5 Experiments

This section presents experimental results, especially focusing on the following questions: (**Q1**) Can we use a small labeling budget to significantly improve selective prediction performance under distribution shift? (**Q2**) Does the proposed ASPEST outperform baselines across different datasets with distribution shift? (**Q3**) What is the effect of checkpoint ensembles and self-training in ASPEST?

## 5.1 Setup

**Datasets.** To demonstrate the practical applicability of active selective prediction, we conduct experiments not only on synthetic data but also on diverse real-world datasets, where distribution shifts are inherent and not artificially constructed. Specifically, we use the following datasets with distribution shift: (i) MNIST→SVHN (LeCun, 1998; Netzer et al., 2011), (ii) CIFAR-10→CINIC-10 (Krizhevsky et al., 2009; Darlow et al., 2018), (iii) FMoW (Koh et al., 2021), (iv) Amazon Review (Koh et al., 2021), (v) Domain-Net (Peng et al., 2019) and (vi) Otto (Benjamin Bossan, 2015). Details of the datasets are described in Appendix E.2.

**Model architectures and source training.** On MNIST→SVHN, we use CNN (LeCun et al., 1989). On CIFAR-10→CINIC-10, we use ResNet-20 (He et al., 2016b). On the FMoW dataset, we use DensetNet-121 (Huang et al., 2017b). On Amazon Review, we use the pre-trained RoBERTa (Liu et al., 2019). On the DomainNet dataset, we use ResNet-50 (He et al., 2016a). On the Otto dataset, we use a multi-layer perceptron. On each dataset, we train the models on the training set $\mathcal{D}^{\text{tr}}$. More details on model architectures and training on source data are presented in Appendix E.3.

**Active learning hyper-parameters.** We evaluate different methods with different labeling budget $M$ values on each dataset. By default, we set the number of rounds $T = 10$ for all methods (Appendix F.6 presents the effect of $T$). During the active learning process, we fine-tune the model on the selected labeled test data. During fine-tuning, we don't apply any data augmentation to the test data. We use the same fine-tuning hyper-parameters for different methods to ensure a fair comparison. More details on the fine-tuning hyper-parameters can be found in Appendix E.4.

**Baselines.** We consider Softmax Response (SR) (Geifman & El-Yaniv, 2017) and Deep Ensembles (DE) (Lakshminarayanan et al., 2017) with various active learning sampling methods as the baselines. SR+Uniform means combining SR with an acquisition function based on uniform sampling (similarly for DE and other acquisition functions). We consider sampling methods from both traditional active learning (e.g., BADGE (Ash et al., 2019)) and active domain adaptation (e.g., CLUE (Prabhu et al., 2021)). Appendix C further describes the details of the baselines.

**Hyper-parameters of ASPEST.** Table 2 comprehensively lists all the hyperparameters used in ASPEST, along with their respective default values. We set $\lambda = 1$, $n_s = 1000$ and $N = 5$ (see Appendix F.7 for the effect of $N$), which are the same as those for Deep Ensembles, for fair comparisons. For all datasets, we use $c_s = 200$, $p = 0.1$, $\eta = 0.9$, the number of self-training epochs to be 20 and $c_e = 5$. It is important to note that the majority of these hyperparameters are general training parameters, and as such, they would not significantly benefit from per-dataset tuning. This significantly reduces the complexity and the need for extensive validation datasets. Specifically, for hyperparameters $n_s$, $c_s$, $c_e$, and $p$, we employ fixed values based on empirical evidence, thus eliminating the need for their adjustment across different datasets. For other hyperparameters $\lambda$, $N$, $T$, and $\eta$, we perform tuning based on the performance observed on a validation dataset (i.e., DomainNet R→I). Once tuned, these values are kept consistent across all other datasets. This approach mirrors real-world scenarios where practitioners often rely on a single, representative validation dataset to tune key hyperparameters before applying the model to various datasets.

| Hyper-parameter | Notation | Default value |
|---|---|---|
| Trade-off parameter to balance the source and target losses in the training objectives | $\lambda$ | 1 |
| The number of training steps in the initial fine-tuning | $n_s$ | 1000 |
| Number of models in the ensemble | $N$ | 5 |
| Number of rounds in active learning | $T$ | 10 |
| The step interval to update the checkpoint ensemble in the initial fine-tuning | $c_s$ | 200 |
| The epoch interval to update the checkpoint ensemble | $c_e$ | 5 |
| The fraction to subsample the pseudo-labeled set | $p$ | 0.1 |
| The threshold used to construct the pseudo-labeled set | $\eta$ | 0.9 |

Table 2: The hyperparameters used in the proposed method ASPEST.

## 5.2 Results

| Method | | $cov^*\|acc \geq 90\%$ ↑ | Accuracy of $f$ ↑ |
|---|---|---|---|
| Selective Prediction | SR (M=0) | 0.08±0.0 | 24.68±0.0 |
| | DE (M=0) | 0.12±0.1 | 26.87±0.8 |
| Active Learning | Margin (M=1K) | N/A | 82.26±0.3 |
| | kCG (M=1K) | N/A | 59.98±4.6 |
| | CLUE (M=1K) | N/A | 81.65±0.3 |
| | BADGE (M=1K) | N/A | 82.11±0.8 |
| Active Selective Prediction | ASPEST (M=1K) | **94.91**±0.4 | **89.20**±0.3 |

Table 3: Results on MNIST→SVHN to describe the effect of combining selective prediction with active learning. The mean and std of each metric over three random runs are reported (mean±std). $cov^*$ is defined in Appendix F.4. All numbers are percentages. **Bold** numbers are superior results.

**Impacts of combining selective prediction with active learning.** We evaluate the accuracy of the source trained models on the test set $U_X$ of different datasets. The results in Appendix F.1 show that the models trained on the source training set $\mathcal{D}^{\mathrm{tr}}$ suffer a performance drop on the target test set $U_X$, and sometimes this drop can be large. For example, the model trained on MNIST has a source test accuracy of 99.40%. However, its accuracy on the target test set $U_X$ from SVHN is only 24.68%. If we directly build a selective classifier on top of the source trained model, then to achieve a target accuracy of 90%, the coverage would be at most 27.42%. In Table 3, we demonstrate that for a target accuracy of 90%, the coverage achieved by the selective prediction baselines SR and DE is very low (nearly 0%). It means that almost all

test examples need human intervention or labeling. This is a large cost since the test set of SVHN contains over 26K images. The results in Table 3 also show that the active learning baselines Margin, kCG, CLUE and BADGE fail to achieve the target accuracy of 90% with a labeling budget of 1K. However, by combining selective prediction with active learning with the proposed method ASPEST, we only need to label $1K$ test examples to achieve a target accuracy of 90% with a coverage of 94.91%. Thus, during active learning and selective prediction processes, only 5.09% test examples from SVHN need to be labeled by a human to achieve the target accuracy of 90%, resulting in a significant reduction of the overall human labeling cost. Similar results are observed for other datasets (see Appendix F.4).

| Dataset | DomainNet R→C (easy) | | Amazon Review | | Otto | |
|---|---|---|---|---|---|---|
| Metric | $cov\|acc \geq 80\% \uparrow$ | AUACC $\uparrow$ | $cov\|acc \geq 80\% \uparrow$ | AUACC $\uparrow$ | $cov\|acc \geq 80\% \uparrow$ | AUACC $\uparrow$ |
| SR+Uniform | 25.56±0.6 | 63.31±0.4 | 13.71±11.3 | 72.71±1.5 | 63.58±0.7 | 84.46±0.2 |
| SR+Confidence | 25.96±0.2 | 64.20±0.6 | 11.28±8.9 | 72.89±0.7 | 69.63±1.7 | 85.91±0.3 |
| SR+Entropy | 25.44±1.0 | 63.52±0.6 | 5.55±7.8 | 71.96±1.6 | 67.79±0.8 | 85.41±0.3 |
| SR+Margin | 26.28±1.2 | 64.37±0.8 | 14.48±10.9 | 73.25±1.0 | 68.10±0.1 | 85.56±0.1 |
| SR+kCG | 21.12±0.3 | 58.88±0.0 | 20.02±11.0 | 72.34±3.2 | 64.84±0.7 | 85.08±0.2 |
| SR+CLUE | 27.17±0.8 | 64.38±0.6 | 4.15±5.9 | 73.43±0.4 | 68.21±1.2 | 85.82±0.3 |
| SR+BADGE | 27.78±0.8 | 64.90±0.5 | 22.58±0.4 | 73.80±0.6 | 67.23±1.0 | 85.41±0.3 |
| DE+Uniform | 30.82±0.8 | 67.60±0.4 | 34.35±1.4 | 76.20±0.3 | 70.74±0.5 | 86.78±0.1 |
| DE+Entropy | 29.13±0.9 | 67.48±0.3 | 31.74±1.4 | 75.98±0.4 | 75.71±0.3 | 87.87±0.1 |
| DE+Confidence | 29.90±0.8 | 67.45±0.3 | 35.12±1.8 | 76.63±0.2 | 75.52±0.2 | 87.84±0.1 |
| DE+Margin | 31.82±1.3 | 68.85±0.4 | 33.42±1.3 | 76.18±0.2 | 75.49±0.8 | 87.89±0.2 |
| DE+Avg-KLD | 32.23±0.2 | 68.73±0.2 | 33.03±1.5 | 76.21±0.4 | 75.91±0.2 | 87.89±0.0 |
| DE+CLUE | 30.80±0.3 | 67.82±0.2 | 33.92±3.0 | 76.27±0.6 | 69.66±0.5 | 86.67±0.1 |
| DE+BADGE | 30.16±1.3 | 68.46±0.3 | 32.23±3.7 | 76.13±0.7 | 73.23±0.2 | 87.55±0.1 |
| ASPEST (ours) | **37.38**±0.1 | **71.61**±0.2 | **38.44**±0.7 | **77.69**±0.1 | **77.85**±0.2 | **88.28**±0.1 |

Table 4: Results of comparing ASPEST to the baselines on DomainNet R→C, Amazon Review and Otto. The mean and std of each metric over three random runs are reported (mean±std). The labeling budget $M$ is 500. All numbers are percentages. **Bold** numbers are superior results.

**Baseline comparisons.** We compare ASPEST with two existing selective prediction methods: SR and DE with various active learning methods. The results in Table 4 (complete results on all datasets for all metrics and different labeling budgets are provided in Appendix F.3) show that ASPEST consistently outperforms the baselines across different image, text and tabular datasets. For example, for MNIST→SVHN, ASPEST improves the AUACC from 79.36% to 88.84% when the labeling budget ($M$) is only 100. When $M = 500$, for DomainNet R→C, ASPEST improves the AUACC from 68.85% to 71.61%; for Amazon Review, ASPEST improves the AUACC from 76.63% to 77.69%; for Otto, ASPEST improves the AUACC from 87.89% to 88.28%.

## 5.3 Analyses and Discussions

In this section, we analyze why the key components checkpoint ensembles and self-training in ASPEST can improve selective prediction and perform ablation study to show their effect.

**Checkpoint ensembles can alleviate overfitting and overconfidence.** We observe that in active selective prediction, when fine-tuning the model on the small amount of selected labeled test data, the model can suffer overfitting and overconfidence issues and ensembling the checkpoints in the training path can effectively alleviate these issues (see the analysis in Appendix F.10).

**Self-training can alleviate overconfidence.** We observe that the checkpoint ensemble constructed after fine-tuning is less confident on the test data $U_X$ compared to the deep ensemble. Thus, using the softmax outputs of the checkpoint ensemble as soft pseudo-labels for self-training can alleviate overconfidence and improve selective prediction performance (see the analysis in Appendix F.11).

**Ablation studies.** Compared to DE+Margin, ASPEST has two additional components: checkpoint ensemble and self-training. We perform ablation experiments on MNIST→SVHN and DomainNet to analyze

| Dataset | MNIST→SVHN | | DomainNet R→C | |
|---|---|---|---|---|
| Metric | AUACC ↑ | | AUACC ↑ | |
| Labeling Budget | 100 | 500 | 500 | 1000 |
| DE+Margin | 78.59±1.4 | 94.31±0.6 | 68.85±0.4 | 71.29±0.3 |
| ASPEST without self-training | 78.09±1.3 | 94.25±0.4 | 69.59±0.2 | 72.45±0.1 |
| ASPEST without checkpoint ensemble | 83.78±2.9 | 96.54±0.2 | 69.94±0.1 | 72.20±0.4 |
| ASPEST ($\eta$=0.1) | 83.77±1.7 | 96.01±0.4 | 70.35±0.2 | 72.89±0.4 |
| ASPEST ($\eta$=0.5) | 83.99±1.3 | 96.24±0.2 | 70.92±0.3 | 73.37±0.1 |
| ASPEST ($\eta$=0.6) | 85.17±1.3 | 96.24±0.2 | 70.96±0.2 | 73.05±0.1 |
| ASPEST ($\eta$=0.8) | 85.40±2.3 | 96.74±0.1 | 71.05±0.2 | 72.99±0.3 |
| ASPEST ($\eta$=0.9) | **88.84**±1.0 | 96.62±0.2 | **71.61**±0.2 | 73.27±0.2 |
| ASPEST ($\eta$=0.95) | 87.67±1.3 | **96.74**±0.1 | 71.03±0.3 | **73.38**±0.2 |

Table 5: Ablation study results for ASPEST. The mean and std of each metric over three random runs are reported (mean±std). All numbers are percentages. **Bold** numbers are superior results.

the effect of these. We also study the effect of the threshold $\eta$ in self-training. The results in Table 5 show for MNIST→SVHN, adding the checkpoint ensemble component alone (ASPEST without self-training) does not improve the performance over DE+Margin, whereas adding the self-training component alone (ASPEST without checkpoint ensemble) can significantly improve the performance. For DomainNet, both checkpoint ensemble and self-training have positive contributions. For both cases, ASPEST (with both self-training and checkpoint ensemble) achieves much better results than DE+Margin or applying those components alone. We also show the performance is not highly sensitive to $\eta$, while typically setting larger $\eta$ (e.g. $\eta = 0.9$) yields better results.

**Integrating with UDA.** To study whether incorporating unsupervised domain adaption (UDA) techniques into training could improve active selective prediction, we evaluate DE with UDA and ASPEST with UDA in Appendix F.12. Our results show that ASPEST outperforms (or on par with) DE with UDA, although ASPEST doesn't utilize UDA. Furthermore, we show that by combining ASPEST with UDA, it might achieve even better performance. For example, on MNIST→SVHN, `ASPEST with DANN` improves the mean AUACC from 96.62% to 97.03% when the labeling budget is 500. However, in some cases, combining ASPEST with UDA yields much worse results. For example, on MNIST→SVHN, when the labeling budget is 100, combining ASPEST with UDA will reduce the mean AUACC by over 4%. We leave the exploration of UDA techniques to improve active selective prediction to future work – superior and robust UDA techniques can be easily incorporated into ASPEST to enhance its overall performance.

## 6  Conclusion

In this paper, we introduce a new learning paradigm called *active selective prediction* which uses active learning to improve selective prediction under distribution shift. We show that this new paradigm results in improved accuracy and coverage on a distributionally shifted test domain and reduces the need for human labeling. We also propose a novel method ASPEST using checkpoint ensemble and self-training with a low labeling cost. We demonstrate ASPEST's effectiveness over other baselines for this new problem setup on various image, text and structured datasets. Future work in this direction can investigate unsupervised hyperparameter tuning on test data, online data streaming, or further minimizing the labeling effort by designing time-preserving labeling interfaces.

### Broader Impact Statement

The proposed framework yields more reliable predictions with more optimized utilization of humans in the loop. One potential risk of such a system is that if the humans in the loop yield inaccurate or biased labels, our framework might cause them being absorbed by the predictor model and the selective prediction mechanism, and eventually the outcomes of the system might be inaccurate and biased. We leave the methods for inaccurate label or bias detection to future work.

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

# Supplementary Material

In Section A, we discuss some additional related work. In Section B, we give the definitions of some evaluation metrics. In Section C, we describe the baselines in detail. In Section D, we show the complete ASPEST algorithm and analyze its computational complexity. In Section E, we provide the details of the experimental setup. In Section F, we give some additional experimental results.

## A  More Related Work

**Distribution shift.** Distribution shift, where the training distribution differs from the test distribution, often occurs in practice and can substantially degrade the accuracy of the deployed DNNs (Koh et al., 2021; Yao et al., 2022; Barbu et al., 2019). Distribution shift can also substantially reduce the quality of uncertainty estimation (Ovadia et al., 2019), which is often used for rejecting examples in selective prediction and selecting samples for labeling in active learning. Several techniques try to tackle the challenge caused by distribution shift, including accuracy estimation (Chen et al., 2021; Chuang et al., 2020), error detection (Hendrycks & Gimpel, 2016; Granese et al., 2021), out-of-distribution detection (Salehi et al., 2021), domain adaptation (Ganin et al., 2016; Saito et al., 2019), selective prediction (Kamath et al., 2020) and active learning (Kirsch et al., 2021). In our work, we combine selective prediction with active learning to address the issue of distribution shift.

**Deep ensembles.** Ensembles of DNNs (or deep ensembles) have been successfully used to boost predictive performance (Moghimi et al., 2016; Zhu et al., 2018). Deep ensembles can also be used to improve the predictive uncertainty estimation (Lakshminarayanan et al., 2017; Fort et al., 2019). Lakshminarayanan et al. (2017) shows that random initialization of the NN parameters along with random shuffling of the data points are sufficient for deep ensembles to perform well in practice. However, training multiple DNNs from random initialization can be very expensive. To obtain deep ensembles more efficiently, recent papers explore using checkpoints during training to construct the ensemble (Wang et al., 2021; Huang et al., 2017a), or fine-tuning a single pre-trained model to create the ensemble (Kobayashi et al., 2022). In our work, we use the checkpoints during fine-tuning a source-trained model via active learning as the ensemble and further boost the ensemble's performance via self-training. We also use the ensemble's uncertainty measured by a margin to select samples for labeling in active learning.

**Self-training.** Self-training is a common algorithmic paradigm for leveraging unlabeled data with DNNs. Self-training methods train a model to fit pseudo-labels (i.e., predictions on unlabeled data made by a previously-learned model) to boost the model's performance (Yarowsky, 1995; Grandvalet & Bengio, 2004; Lee et al., 2013; Wei et al., 2020; Sohn et al., 2020). In this work, we use self-training to improve selective prediction performance. Instead of using predicted labels as pseudo-labels as a common practice in prior works, we use the average softmax outputs of the checkpoints during training as the pseudo-labels and self-train the models in the ensemble on them with the KL-Divergence loss to improve selective prediction performance.

## B  Evaluation Metrics

We have introduced the accuracy and coverage metrics in Section 3.2. The accuracy is measured on the predictions made by the model without human intervention while the coverage is the fraction of remaining unlabeled data points where we can rely on the model's prediction without human intervention. The accuracy and coverage metrics depend on the threshold $\tau$. The following evaluation metrics are proposed to be agnostic to the threshold $\tau$:

**Maximum Accuracy at a Target Coverage.** Given a target coverage $t_c$, the maximum accuracy is defined as:

$$\max_{\tau} \quad acc(f_s, \tau), \quad s.t. \quad cov(f_s, \tau) \geq t_c \tag{11}$$

We denote this metric as $acc|cov \geq t_c$.

**Maximum Coverage at a Target Accuracy.** Given a target accuracy $t_a$, the maximum coverage is defined as:

$$\max_{\tau} \quad cov(f_s, \tau), \quad s.t. \quad acc(f_s, \tau) \geq t_a \tag{12}$$

When $\tau = \infty$, we define $cov(f_s, \tau) = 0$ and $acc(f_s, \tau) = 1$. We denote this metric as $cov|acc \geq t_a$.

**Area Under Accuracy-Coverage Curve (AUACC).** We define the AUACC metric as:

$$\text{AUACC}(f_s) = \int_0^1 acc(f_s, \tau) \mathrm{d}cov(f_s, \tau) \tag{13}$$

We use the composite trapezoidal rule to estimate the integration.

## C   Baselines

We consider two selective classification baselines Softmax Response (SR) (Geifman & El-Yaniv, 2017) and Deep Ensembles (DE) (Lakshminarayanan et al., 2017) and combine them with active learning techniques. We describe them in detail below.

### C.1   Softmax Response

Suppose the neural network classifier is $f$ where the last layer is a softmax. Let $f(\mathbf{x} \mid k)$ be the soft response output for the $k$-th class. Then the classifier is defined as $f(\mathbf{x}) = \arg\max_{k \in \mathcal{Y}} f(\mathbf{x} \mid k)$ and the selection scoring function is defined as $g(\mathbf{x}) = \max_{k \in \mathcal{Y}} f(\mathbf{x} \mid k)$, which is also known as the Maximum Softmax Probability (MSP) of the neural network. Recall that with $f$ and $g$, the selective classifier is defined in Eq (1). We use active learning to fine-tune the model $f$ to improve selective prediction performance of SR on the unlabeled test dataset $U_X$. The complete algorithm is presented in Algorithm 1. In our experiments, we always set $\lambda = 1$. We use the joint training objective (22) to avoid over-fitting to the small labeled test set $\cup_{l=1}^t \tilde{B}_l$ and prevent the model from forgetting the source training knowledge. The algorithm can be combined with different kinds of acquisition functions. We describe the acquisition functions considered for SR below.

**Uniform.** In the $t$-th round of active learning, we select $\lceil \frac{M}{T} \rceil$ data points as the batch $B_t$ from $U_X \setminus \cup_{l=1}^{t-1} B_l$ via uniform random sampling. The corresponding acquisition function is: $a(B, f_{t-1}, g_{t-1}) = 1$. When solving the objective (21), the tie is broken randomly.

**Confidence.** We define the confidence score of $f$ on the input $\mathbf{x}$ as

$$S_{\text{conf}}(\mathbf{x}; f) = \max_{k \in \mathcal{Y}} \quad f(\mathbf{x} \mid k) \tag{14}$$

Then the acquisition function in the $t$-th round of active learning is defined as:

$$a(B, f_{t-1}, g_{t-1}) = -\sum_{\mathbf{x} \in B} S_{\text{conf}}(\mathbf{x}; f_{t-1}) \tag{15}$$

That is we select those test examples with the lowest confidence scores for labeling.

**Entropy.** We define the entropy score of $f$ on the input $\mathbf{x}$ as

$$S_{\text{entropy}}(\mathbf{x}; f) = \sum_{k \in \mathcal{Y}} \quad -f(\mathbf{x} \mid k) \cdot \log f(\mathbf{x} \mid k) \tag{16}$$

Then the acquisition function in the $t$-th round of active learning is defined as:

$$a(B, f_{t-1}, g_{t-1}) = \sum_{\mathbf{x} \in B} S_{\text{entropy}}(\mathbf{x}; f_{t-1}) \tag{17}$$

That is we select those test examples with the highest entropy scores for labeling.

**Margin.** We define the margin score of $f$ on the input $\mathbf{x}$ as

$$S_{\mathrm{margin}}(\mathbf{x}; f) = f(\mathbf{x} \mid \hat{y}) - \max_{k \in \mathcal{Y} \setminus \{\hat{y}\}} f(\mathbf{x} \mid k) \tag{18}$$

$$s.t. \quad \hat{y} = \arg\max_{k \in \mathcal{Y}} f(\mathbf{x} \mid k) \tag{19}$$

Then the acquisition function in the $t$-th round of active learning is defined as:

$$a(B, f_{t-1}, g_{t-1}) = -\sum_{\mathbf{x} \in B} S_{\mathrm{margin}}(\mathbf{x}; f_{t-1}) \tag{20}$$

That is we select those test examples with lowest margin scores for labeling.

**kCG.** We use the k-Center-Greedy algorithm proposed in (Sener & Savarese, 2017) to select test examples for labeling in each round.

**CLUE.** We use the Clustering Uncertainty-weighted Embeddings (CLUE) proposed in (Prabhu et al., 2021) to select test examples for labeling in each round. Following (Prabhu et al., 2021), we set the hyper-parameter $T = 0.1$ on DomainNet and set $T = 1.0$ on other datasets.

**BADGE.** We use the Diverse Gradient Embeddings (BADGE) proposed in (Ash et al., 2019) to select test examples for labeling in each round.

---

**Algorithm 1** Softmax Response with Active Learning

---

**Input:** A training dataset $\mathcal{D}^{\mathrm{tr}}$, an unlabeled test dataset $U_X$, the number of rounds $T$, the labeling budget $M$, a source-trained model $\bar{f}$, an acquisition function $a$ and a hyper-parameter $\lambda$.
  Let $f_0 = \bar{f}$.
  Let $B_0 = \emptyset$.
  Let $g_t(\mathbf{x}) = \max_{k \in \mathcal{Y}} f_t(\mathbf{x} \mid k)$.
  **for** $t = 1, \cdots, T$ **do**
    Select a batch $B_t$ with a size of $m = \lceil \frac{M}{T} \rceil$ from $U_X$ for labeling via:

$$B_t = \arg\max_{B \subset U_X \setminus (\cup_{l=0}^{t-1} B_l), |B|=m} a(B, f_{t-1}, g_{t-1}) \tag{21}$$

   Use an oracle to assign ground-truth labels to the examples in $B_t$ to get $\tilde{B}_t$.
   Fine-tune the model $f_{t-1}$ using the following training objective:

$$\min_{\theta} \quad \mathbb{E}_{(\mathbf{x},y) \in \cup_{l=1}^{t} \tilde{B}_l} \ell_{CE}(\mathbf{x}, y; \theta) + \lambda \cdot \mathbb{E}_{(\mathbf{x},y) \in \mathcal{D}^{\mathrm{tr}}} \ell_{CE}(\mathbf{x}, y; \theta) \tag{22}$$

   where $\theta$ is the model parameters of $f_{t-1}$ and $\ell_{CE}$ is the cross-entropy loss function.
   Let $f_t = f_{t-1}$.
  **end for**
**Output:** The classifier $f = f_T$ and the selection scoring function $g = \max_{k \in \mathcal{Y}} f(\mathbf{x} \mid k)$.

---

## C.2 Deep Ensembles

It has been shown that deep ensembles can significantly improve selective prediction performance (Lakshminarayanan et al., 2017), not only because deep ensembles are more accurate than a single model, but also because deep ensembles yield more calibrated confidence.

Suppose the ensemble model $f$ contains $N$ models $f^1, \ldots, f^N$. Let $f^j(\mathbf{x} \mid k)$ denote the predicted probability of the model $f^j$ on the $k$-th class. We define the predicted probability of the ensemble model $f$ on the $k$-th

class as:

$$f(\mathbf{x} \mid k) = \frac{1}{N} \sum_{j=1}^{N} f^j(\mathbf{x} \mid k). \tag{23}$$

The classifier is defined as $f(\mathbf{x}) = \arg\max_{k \in \mathcal{Y}} f(\mathbf{x} \mid k)$ and the selection scoring function is defined as $g(\mathbf{x}) = \max_{k \in \mathcal{Y}} f(\mathbf{x} \mid k)$. We use active learning to fine-tune each model $f^j$ in the ensemble to improve selective prediction performance of the ensemble on the unlabeled test dataset $U_X$. Each model $f^j$ is first initialized by the source-trained model $\bar{f}$, and then fine-tuned independently via Stochastic Gradient Decent (SGD) with different sources of randomness (e.g., different random order of the training batches) on the training dataset $\mathcal{D}^{\text{tr}}$ and the selected labeled test data. Note that this way to construct the ensembles is different from the standard Deep Ensembles method, which trains the models from different random initialization. We use this way to construct the ensemble due to the constraint in our problem setting, which requires us to fine-tune a given source-trained model $\bar{f}$. Training the models from different random initialization might lead to an ensemble with better performance, but it is much more expensive, especially when the training dataset and the model are large (e.g., training foundation models). Thus, the constraint in our problem setting is feasible in practice. The complete algorithm is presented in Algorithm 2. In our experiments, we always set $\lambda = 1$, $N = 5$, and $n_s = 1000$. We also use joint training here and the reasons are the same as those for the Softmax Response baseline. The algorithm can be combined with different kinds of acquisition functions. We describe the acquisition functions considered below.

**Uniform.** In the $t$-th round of active learning, we select $\lceil \frac{M}{T} \rceil$ data points as the batch $B_t$ from $U_X \setminus \cup_{l=0}^{t-1} B_l$ via uniform random sampling. The corresponding acquisition function is: $a(B, f_{t-1}, g_{t-1}) = 1$. When solving the objective (30), the tie is broken randomly.

**Confidence.** The confidence scoring function $S_{\text{conf}}$ for the ensemble model $f$ is the same as that in Eq. (14) ($f(\mathbf{x} \mid k)$ for the ensemble model $f$ is defined in Eq. (23)). The acquisition function in the $t$-th round of active learning is defined as:

$$a(B, f_{t-1}, g_{t-1}) = -\sum_{\mathbf{x} \in B} S_{\text{conf}}(\mathbf{x}; f_{t-1}) \tag{24}$$

That is we select those test examples with the lowest confidence scores for labeling.

**Entropy.** The entropy scoring function $S_{\text{entropy}}$ for the ensemble model $f$ is the same as that in Eq. (16). The acquisition function in the $t$-th round of active learning is defined as:

$$a(B, f_{t-1}, g_{t-1}) = \sum_{\mathbf{x} \in B} S_{\text{entropy}}(\mathbf{x}; f_{t-1}), \tag{25}$$

That is we select those test examples with the highest entropy scores for labeling.

**Margin.** The margin scoring function $S_{\text{margin}}$ for the ensemble model $f$ is the same as that in Eq. (18). The acquisition function in the $t$-th round of active learning is defined as:

$$a(B, f_{t-1}, g_{t-1}) = -\sum_{\mathbf{x} \in B} S_{\text{margin}}(\mathbf{x}; f_{t-1}) \tag{26}$$

That is we select those test examples with the lowest margin scores for labeling.

**Avg-KLD.** The Average Kullback-Leibler Divergence (Avg-KLD) is proposed in (McCallum et al., 1998) as a disagreement measure for the model ensembles, which can be used for sample selection in active learning. The Avg-KLD score of the ensemble model $f$ on the input $\mathbf{x}$ is defined as:

$$S_{\text{kl}}(\mathbf{x}; f) = \frac{1}{N} \sum_{j=1}^{N} \sum_{k \in \mathcal{Y}} f^j(\mathbf{x} \mid k) \cdot \log \frac{f^j(\mathbf{x} \mid k)}{f(\mathbf{x} \mid k)}. \tag{27}$$

Then the acquisition function in the $t$-th round of active learning is defined as:

$$a(B, f_{t-1}, g_{t-1}) = \sum_{\mathbf{x} \in B} S_{\text{kl}}(\mathbf{x}; f_{t-1}), \tag{28}$$

That is we select those test examples with the highest Avg-KLD scores for labeling.

**CLUE.** CLUE (Prabhu et al., 2021) is proposed for a single model. Here, we adapt CLUE for the ensemble model, which requires a redefinition of the entropy function $\mathcal{H}(Y \mid \mathbf{x})$ and the embedding function $\phi(\mathbf{x})$ used in the CLUE algorithm. We define the entropy function as Eq. (16) with the ensemble model $f$. Suppose $\phi^j$ is the embedding function for the model $f^j$ in the ensemble. Then, the embedding of the ensemble model $f$ on the input $\mathbf{x}$ is $[\phi^1(\mathbf{x}), \ldots, \phi^N(\mathbf{x})]$, which is the concatenation of the embeddings of the models $f^1, \ldots, f^N$ on $\mathbf{x}$. Following (Prabhu et al., 2021), we set the hyper-parameter $T = 0.1$ on DomainNet and set $T = 1.0$ on other datasets.

**BADGE.** BADGE (Ash et al., 2019) is proposed for a single model. Here, we adapt BADGE for the ensemble model, which requires a redefinition of the gradient embedding $g_x$ in the BADGE algorithm. Towards this end, we propose the gradient embedding $g_x$ of the ensemble model $f$ as the concatenation of the gradient embeddings of the models $f^1, \ldots, f^N$.

---

**Algorithm 2** Deep Ensembles with Active Learning

---

**Input:** A training dataset $\mathcal{D}^{\mathrm{tr}}$, An unlabeled test dataset $U_X$, the number of rounds $T$, the total labeling budget $M$, a source-trained model $\bar{f}$, an acquisition function $a(B, f, g)$, the number of models in the ensemble $N$, the number of initial training steps $n_s$, and a hyper-parameter $\lambda$.

Let $f_0^j = \bar{f}$ for $j = 1, \ldots, N$.

Fine-tune each model $f_0^j$ in the ensemble via SGD for $n_s$ training steps independently using the following training objective with different randomness:

$$\min_{\theta^j} \quad \mathbb{E}_{(\mathbf{x},y)\in\mathcal{D}^{\mathrm{tr}}} \quad \ell_{CE}(\mathbf{x}, y; \theta^j) \tag{29}$$

where $\theta^j$ is the model parameters of $f_0^j$ and $\ell_{CE}$ is the cross-entropy loss function.

Let $B_0 = \emptyset$.

Let $g_t(\mathbf{x}) = \max_{k\in\mathcal{Y}} f_t(\mathbf{x} \mid k)$.

**for** $t = 1, \cdots, T$ **do**

    Select a batch $B_t$ with a size of $m = \lceil \frac{M}{T} \rceil$ from $U_X$ for labeling via:

$$B_t = \arg \max_{B \subset U_X \setminus (\cup_{l=0}^{t-1} B_l), |B|=m} a(B, f_{t-1}, g_{t-1}) \tag{30}$$

    Use an oracle to assign ground-truth labels to the examples in $B_t$ to get $\tilde{B}_t$.

    Fine-tune each model $f_{t-1}^j$ in the ensemble via SGD independently using the following training objective with different randomness:

$$\min_{\theta^j} \quad \mathbb{E}_{(\mathbf{x},y)\in\cup_{l=1}^t \tilde{B}_l} \quad \ell_{CE}(\mathbf{x}, y; \theta^j) + \lambda \cdot \mathbb{E}_{(\mathbf{x},y)\in\mathcal{D}^{\mathrm{tr}}} \quad \ell_{CE}(\mathbf{x}, y; \theta^j) \tag{31}$$

    where $\theta^j$ is the model parameters of $f_{t-1}^j$.

    Let $f_t^j = f_{t-1}^j$ for $j = 1, \ldots, N$.

**end for**

**Output:** The classifier $f = f_T$ and the selection scoring function $g = \max_{k\in\mathcal{Y}} f(\mathbf{x} \mid k)$.

---

## D ASPEST Algorithm and its Computational Complexity

Algorithm 3 presents the overall ASPEST method. Figure 3 illustrates how the checkpoint ensemble and the pseudo-labeled set are constructed in the proposed ASPEST. Next, we will analyze the computational complexity of ASPEST.

Let the complexity for one step of updating $P$ and $N_e$ be $t_u$ (mainly one forward pass of DNN); for one DNN gradient update step is $t_g$ (mainly one forward and backward pass of DNN); and for sample selection is $t_s$ (mainly sorting test examples). Then, the total complexity of ASPEST would be $O\Big(N \cdot \frac{n_s}{c_s} \cdot t_u + N \cdot$

---

**Algorithm 3** Active Selective Prediction using Ensembles and Self-Training

---

**Input:** A training set $\mathcal{D}^{tr}$, a unlabeled test set $U_X$, the number of rounds $T$, the labeling budget $M$, the number of models $N$, the number of initial training steps $n_s$, the initial checkpoint steps $c_s$, a checkpoint epoch $c_e$, a threshold $\eta$, a sub-sampling fraction $p$, and a hyper-parameter $\lambda$.
    Let $f_0^j = \bar{f}$ for $j = 1, \ldots, N$.
    Set $N_e = 0$ and $P = \mathbf{0}_{n \times K}$.
    Fine-tune each $f_0^j$ for $n_s$ training steps using the following training objective:

$$\min_{\theta^j} \mathbb{E}_{(\mathbf{x}, y) \in \mathcal{D}^{\mathrm{tr}}} \quad \ell_{CE}(\mathbf{x}, y; \theta^j),$$

    and update $P$ and $N_e$ using Eq. (6) every $c_s$ training steps.
    **for** $t = 1, \cdots, T$ **do**
        Select a batch $B_t$ from $U_X$ for labeling using the sample selection objective (7).
        Use an oracle to assign ground-truth labels to the examples in $B_t$ to get $\tilde{B}_t$.
        Set $N_e = 0$ and $P = \mathbf{0}_{n \times K}$.
        Fine-tune each $f_{t-1}^j$ using objective (8), while updating $P$ and $N_e$ using Eq (6) every $c_e$ training epochs.
        Let $f_t^j = f_{t-1}^j$.
        Construct the pseudo-labeled set $R$ via Eq (9) and create $R_{\mathrm{sub}}$ by randomly sampling up to $[p \cdot n]$ data points from $R$.
        Train each $f_t^j$ further via SGD using the objective (10) and update $P$ and $N_e$ using Eq (6) every $c_e$ training epochs.
    **end for**
**Output:** The classifier $f(\mathbf{x}_i) = \arg\max_{k \in \mathcal{Y}} P_{i,k}$ and the selection scoring function $g(\mathbf{x}_i) = \max_{k \in \mathcal{Y}} P_{i,k}$.

---

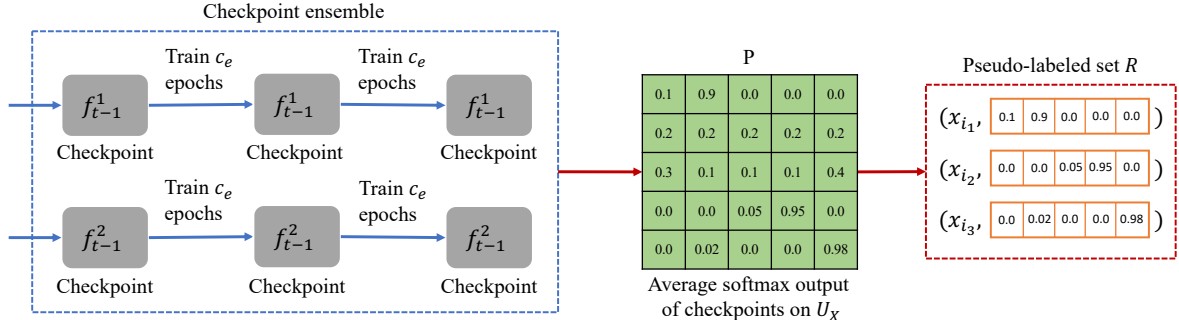

Figure 3: Illustration of the checkpoint ensemble and pseudo-labeled set construction in the proposed ASPEST.

$n_s \cdot t_g + T \cdot [t_s + N \cdot (e_f + e_s \cdot p) \cdot \frac{n}{b} \cdot t_g + N \cdot \frac{e_s + e_f}{c_e} \cdot t_u])$, where $e_f$ is the number of fine-tuning epochs and $e_s$ is the number of self-training epochs and $b$ is the batch size. Although the training objectives include training on $\mathcal{D}^{\mathrm{tr}}$, the complexity doesn't depend on the size of $\mathcal{D}^{\mathrm{tr}}$ since we measure $e_f$ over $\cup_{l=1}^t \tilde{B}_l$ in training objective (8) and measure $e_s$ over $R_{\mathrm{sub}}$ in training objective (10). In practice, we usually have $t_s \ll t_g$ and $t_u \ll t_g$. Also, we set $e_s \cdot p < e_f$, $n_s \ll \frac{n}{b} \cdot T \cdot e_f$ and $\frac{e_s + e_f}{c_e} \ll (e_f + e_s \cdot p) \cdot \frac{n}{b}$. So the complexity of ASPEST is $O\left(N \cdot T \cdot \frac{n}{b} \cdot e_f \cdot t_g\right)$. Suppose the size of $\mathcal{D}^{\mathrm{tr}}$ is $n_{tr}$ and the number of source training epochs is $e_p$. Then, the complexity for source training is $O(\frac{n_{tr}}{b} \cdot e_p \cdot t_g)$. In practice, we usually have $N \cdot T \cdot n \cdot e_f \ll n_{tr} \cdot e_p$. Overall, the complexity of ASPEST would be much smaller than that of source training.

It should be noted that ASPEST shares the same computational complexity as the DE+Margin baseline, which also has a computational complexity of $O\left(N \cdot T \cdot \frac{n}{b} \cdot e_f \cdot t_g\right)$. While both methods utilize an ensemble approach, a key distinction is that DE+Margin does not incorporate self-training, unlike ASPEST. In contrast, the SR+Margin baseline, which does not utilize ensembles, has a lower computational complexity of $O\left(T \cdot \frac{n}{b} \cdot e_f \cdot t_g\right)$.

While DE+Margin and ASPEST indeed have higher computational complexities compared to SR+Margin due to the inclusion of ensembles, this does not necessarily translate to proportionally longer running times. With careful implementation and adequate computational resources, it is feasible to train ensemble models in parallel, potentially aligning the actual running times of ASPEST with those of the baseline methods, SR+Margin and DE+Margin. This parallelization strategy could mitigate the increased computational demand, making the running times of ASPEST comparable to, if not competitive with, the baselines under similar resource conditions.

# E   Details of Experimental Setup

## E.1   Computing Infrastructure and Runtime

We run all experiments with TensorFlow 2.0 on NVIDIA A100 GPUs in the Debian GNU/Linux 10 system. We report the total runtime of the proposed method ASPEST on each dataset in Table 6. Note that in our implementation, we train models in the ensemble sequentially. However, it is possible to train models in the ensemble in parallel, which can significantly reduce the runtime. With the optimal implementation, the inference latency of the ensemble can be as low as the inference latency of a single model.

| Dataset | Total Runtime |
|---|---|
| MNIST→SVHN | 24 min |
| CIFAR-10→CINIC-10 | 1 hour |
| FMoW | 2 hour 48 min |
| Amazon Review | 1 hour 34 min |
| DomainNet (R→C) | 2 hours 10 min |
| DomainNet (R→P) | 1 hour 45 min |
| DomainNet (R→S) | 1 hour 51 min |
| Otto | 18 min |

Table 6: The runtime of ASPEST when the labeling budget $M = 500$. We use the default hyper-parameters for ASPEST described in Section 5.1.

## E.2   Datasets

We describe the datasets used below. For all image datasets, we normalize the range of pixel values to [0,1].

**MNIST→SVHN.** The source training dataset $\mathcal{D}^{\text{tr}}$ is MNIST (LeCun, 1998) while the target test dataset $U_X$ is SVHN (Netzer et al., 2011). MNIST consists 28×28 grayscale images of handwritten digits, containing in total 5,500 training images and 1,000 test images. We resize each image to be 32×32 resolution and change them to be colored. We use the training set of MNIST as $\mathcal{D}^{\text{tr}}$ and the test set of MNIST as the source validation dataset. SVHN consists 32×32 colored images of digits obtained from house numbers in Google Street View images. The training set has 73,257 images and the test set has 26,032 images. We use the test set of SVHN as $U_X$.

**CIFAR-10→CINIC-10.** The source training dataset $\mathcal{D}^{\text{tr}}$ is CIFAR-10 (Krizhevsky et al., 2009) while the target test dataset $U_X$ is CINIC-10 (Darlow et al., 2018). CIFAR-10 consists 32×32 colored images with ten classes (dogs, frogs, ships, trucks, etc.), each consisting of 5,000 training images and 1,000 test images. We use the training set of CIFAR-10 as $\mathcal{D}^{\text{tr}}$ and the test set of CIFAR-10 as the source validation dataset. During training, we apply random horizontal flipping and random cropping with padding data augmentations to the training images. CINIC-10 is an extension of CIFAR-10 via the addition of downsampled ImageNet images. CINIC-10 has a total of 270,000 images equally split into training, validation, and test. In each subset (90,000 images) there are ten classes (identical to CIFAR-10 classes). There are 9,000 images per class per subset. We use a subset of the CINIC-10 test set containing 30,000 images as $U_X$.

**FMoW.** We use the FMoW-WILDS dataset from (Koh et al., 2021). FMoW-wilds is based on the Functional Map of the World dataset (Christie et al., 2018), which collected and categorized high-resolution satellite

images from over 200 countries based on the functional purpose of the buildings or land in the image, over the years 2002–2018. The task is multi-class classification, where the input $\mathbf{x}$ is an RGB satellite image, the label $y$ is one of 62 building or land use categories, and the domain $d$ represents both the year the image was taken as well as its geographical region (Africa, the Americas, Oceania, Asia, or Europe). The FMoW dataset encompasses both temporal and geographical distribution shifts, covering diverse regions across various continents over a span of 16 years. The training set contains 76,863 images from the years 2002-2013. The In-Distribution (ID) validation set contains 11,483 images from the years 2002-2013. The OOD test set contains 22,108 images from the years 2016-2018. We resize each image to be $96 \times 96$ resolution to save computational cost. We use the training set as $\mathcal{D}^{\mathrm{tr}}$ and the ID validation set as the source validation dataset. During training, we apply random horizontal flipping and random cropping with padding data augmentations to the training images. We use the OOD test set as $U_X$.

**Amazon Review.** We use the Amazon Review WILDS dataset from (Koh et al., 2021). The dataset comprises 539,502 customer reviews on Amazon taken from the Amazon Reviews dataset (Ni et al., 2019). The task is multi-class sentiment classification, where the input $\mathbf{x}$ is the text of a review, the label $y$ is a corresponding star rating from 1 to 5, and the domain $d$ is the identifier of the reviewer who wrote the review. The issue of distribution shifts in the Amazon Review dataset stems from the diverse writing styles and perspectives of different reviewers. The training set contains 245,502 reviews from 1,252 reviewers. The In-Distribution (ID) validation set contains 46,950 reviews from 626 of the 1,252 reviewers in the training set. The Out-Of-Distribution (OOD) test set contains 100,050 reviews from another set of 1,334 reviewers, distinct from those of the training set. We use the training set as $\mathcal{D}^{\mathrm{tr}}$ and the ID validation set as the source validation dataset. We use a subset of the OOD test set containing 22,500 reviews from 300 reviewers as $U_X$.

**DomainNet.** DomainNet (Peng et al., 2019) is a dataset of common objects in six different domains. All domains include 345 categories (classes) of objects such as Bracelet, plane, bird, and cello. We use five domains from DomainNet including: (1) Real: photos and real world images. The training set from the Real domain has 120,906 images while the test set has 52,041 images; (2) Clipart: a collection of clipart images. The training set from the Clipart domain has 33,525 images while the test set has 14,604 images; (3) Sketch: sketches of specific objects. The training set from the Sketch has 48,212 images while the test set has 20,916 images; (4) Painting: artistic depictions of objects in the form of paintings. The training set from the Painting domain has 50,416 images while the test set has 21,850 images. (5) Infograph: infographic images with specific objects. The training set from the Infograph domain has 36,023 images while the test set has 15,582 images. We resize each image from all domains to be $96 \times 96$ resolution to save computational cost. We use the training set from the Real domain as $\mathcal{D}^{\mathrm{tr}}$ and the test set from the Real domain as the source validation dataset. During training, we apply random horizontal flipping and random cropping with padding data augmentations to the training images. We use the test sets from three domains Clipart, Sketch, and Painting as three different $U_X$ for evaluation. So we evaluate three shifts: Real→Clipart (R→C), Real→Sketch (R→S), and Real→Painting (R→P). We use the remaining shift Real→Infograph (R→I) as a validation dataset for tuning the hyper-parameters.

**Otto.** The Otto Group Product Classification Challenge (Benjamin Bossan, 2015) is a tabular dataset hosted on Kaggle [2]. The task is to classify each product with 93 features into 9 categories. Each target category represents one of the most important product categories (like fashion, electronics, etc). It contains $61,878$ training data points. Since it only provides labels for the training data, we need to create the training, validation and test set. To create a test set that is from a different distribution than the training set, we apply the Local Outlier Factor (LOF) (Breunig et al., 2000), which is an unsupervised outlier detection method, on the Otto training data to identify a certain fraction (e.g., 0.2) of outliers as the test set. Specifically, we apply the *LocalOutlierFactor* function provided by scikit-learn (Pedregosa et al., 2011) on the training data with a contamination of 0.2 (contamination value determines the proportion of outliers in the data set) to identify the outliers. We identify $12,376$ outlier data points and use them as the test set $U_X$. We then randomly split the remaining data into a training set $\mathcal{D}^{\mathrm{tr}}$ with $43,314$ data points and a source validation set with $6,188$ data points. We show that the test set indeed has a distribution shift compared to the source

---

[2]URL: `https://kaggle.com/competitions/otto-group-product-classification-challenge`

validation set, which causes the model trained on the training set to have a drop in performance (see Table 7 in Appendix F.1).

### E.3 Details on Model Architectures and Training on Source Data

On all datasets, we use the following supervised training objective for training models on the source training set $\mathcal{D}^{\mathrm{tr}}$:

$$\min_{\theta} \quad \mathbb{E}_{(\mathbf{x},y)\in\mathcal{D}^{\mathrm{tr}}} \quad \ell_{CE}(\mathbf{x}, y; \theta) \tag{32}$$

where $\ell_{CE}$ is the cross-entropy loss and $\theta$ is the model parameters.

On MNIST→SVHN, we use the Convolutional Neural Network (CNN) (LeCun et al., 1989) consisting of four convolutional layers followed by two fully connected layers with batch normalization and dropout layers. We train the model on the training set of MNIST for 20 epochs using the Adam optimizer (Kingma & Ba, 2014) with a learning rate of $10^{-3}$ and a batch size of 128.

On CIFAR-10→CINIC-10, we use the ResNet-20 network (He et al., 2016b). We train the model on the training set of CIFAR-10 for 200 epochs using the SGD optimizer with a learning rate of 0.1, a momentum of 0.9, and a batch size of 128. The learning rate is multiplied by 0.1 at the 80, 120, and 160 epochs, respectively, and is multiplied by 0.5 at the 180 epoch.

On the FMoW dataset, we use the DensetNet-121 network (Huang et al., 2017b) pre-trained on ImageNet. We train the model further for 50 epochs using the Adam optimizer with a learning rate of $10^{-4}$ and a batch size of 128.

On the Amazon Review dataset, we use the pre-trained RoBERTa base model (Liu et al., 2019) to extract the embedding of the input sentence for classification (i.e., RoBERTa's output for the [CLS] token) and then build an eight-layer fully connected neural network (also known as a multi-layer perceptron) with batch normalization, dropout layers and L2 regularization on top of the embedding. Note that we only update the parameters of the fully connected neural network without updating the parameters of the pre-trained RoBERTa base model during training (i.e., freeze the parameters of the RoBERTa base model during training). We train the model for 200 epochs using the Adam optimizer with a learning rate of $10^{-3}$ and a batch size of 128.

On the DomainNet dataset, we use the ResNet-50 network (He et al., 2016a) pre-trained on ImageNet. We train the model further on the training set from the Real domain for 50 epochs using the Adam optimizer with a learning rate of $10^{-4}$ and a batch size of 128.

On the Otto dataset, we use a six-layer fully connected neural network (also known as a multi-layer perceptron) with batch normalization, dropout layers and L2 regularization. We train the model on the created training set for 200 epochs using the Adam optimizer with a learning rate of $10^{-3}$ and a batch size of 128.

### E.4 Active learning hyper-parameters

During the active learning process, we fine-tune the model on the selected labeled test data. During fine-tuning, we don't apply any data augmentation to the test data. We use the same fine-tuning hyper-parameters for different methods to ensure a fair comparison. The optimizer used is the same as that in the source training stage (described in Appendix E.3). On MNIST→SVHN, we use a learning rate of $10^{-3}$; On CIFAR-10→CINIC-10, we use a learning rate of $5 \times 10^{-3}$; On FMoW, we use a learning rate of $10^{-4}$; On Amazon Review, we use a learning rate of $10^{-3}$; On DomainNet, we use a learning rate of $10^{-4}$; On Otto, we use a learning rate of $10^{-3}$. On all datasets, we fine-tune the model for at least 50 epochs and up to 200 epochs with a batch size of 128 and early stopping using 10 patient epochs.

## F   Additional Experimental Results

### F.1   Evaluate Source-Trained Models

In this section, we evaluate the accuracy of the source-trained models on the source validation dataset and the target test dataset $U_X$. The models are trained on the source training set $\mathcal{D}^{\mathrm{tr}}$ (refer to Appendix E.3 for the details of source training). The source validation data are randomly sampled from the training data distribution while the target test data are sampled from a different distribution than the training data distribution. The results in Table 7 show that the models trained on $\mathcal{D}^{\mathrm{tr}}$ always suffer a drop in accuracy when evaluating them on the target test dataset $U_X$.

| Dataset | Source Accuracy | Target Accuracy |
|---|---|---|
| MNIST→SVHN | 99.40 | 24.68 |
| CIFAR-10→CINIC-10 | 90.46 | 71.05 |
| FMoW | 46.25 | 38.01 |
| Amazon Review | 65.39 | 61.40 |
| DomainNet (R→C) | 63.45 | 33.37 |
| DomainNet (R→P) | 63.45 | 26.29 |
| DomainNet (R→S) | 63.45 | 16.00 |
| Otto | 80.72 | 66.09 |

Table 7: Results of evaluating the accuracy of the source-trained models on the source validation dataset and the target test dataset $U_X$. All numbers are percentages.

### F.2   Evaluate Softmax Response with Various Active Learning Methods

To see whether combining existing selective prediction and active learning approaches could solve the active selective prediction problem, we evaluate the existing selective prediction method Softmax Response (SR) with active learning methods based on uncertainty or diversity. The results in Table 8 show that the methods based on uncertainty sampling (SR+Confidence, SR+Entropy and SR+Margin) achieve relatively high accuracy of $f$, but suffer from the overconfidence issue (i.e., mis-classification with high confidence). The method based on diversity sampling (SR+kCG) doesn't have the overconfidence issue, but suffers from low accuracy of $f$. Also, the hybrid methods based on uncertainty and diversity sampling (SR+CLUE and SR+BADGE) still suffer from the overconfidence issue. In contrast, the proposed method ASPEST achieves much higher accuracy of $f$, effectively alleviates the overconfidence issue, and significantly improves the selective prediction performance.

| Method | Accuracy of $f$ ↑ | Overconfidence ratio ↓ | AUACC↑ |
|---|---|---|---|
| SR+Confidence | 45.29±3.39 | 16.91±2.24 | 64.14±2.83 |
| SR+Entropy | 45.78±6.36 | 36.84±18.96 | 65.88±4.74 |
| SR+Margin | 58.10±0.55 | 13.18±1.85 | 76.79±0.45 |
| SR+kCG | 32.68±3.87 | **0.04**±0.01 | 48.83±7.21 |
| SR+CLUE | 55.22±2.27 | 9.47±0.94 | 73.15±2.68 |
| SR+BADGE | 56.55±1.62 | 8.37±2.56 | 76.06±1.63 |
| ASPEST (ours) | **71.82**±1.49 | 0.10±0.02 | **88.84**±1.02 |

Table 8: Evaluating the Softmax Response (SR) method with various active learning methods and the proposed ASPEST on MNIST→SVHN. The experimental setup is describe in Section 5.1. The labeling budget $M$ is 100. The overconfidence ratio is the ratio of *mis-classified* unlabeled test inputs that have confidence $\geq 1$ (the highest confidence). The mean and std of each metric over three random runs are reported (mean±std). All numbers are percentages. **Bold** numbers are superior results.

### F.3 Complete Evaluation Results

We give complete experimental results for the baselines and the proposed method ASPEST on all datasets in this section. We repeat each experiment three times with different random seeds and report the mean and standard deviation (std) values. These results are shown in Table 9 (MNIST→SVHN), Table 10 (CIFAR-10→CINIC-10), Table 11 (FMoW), Table 12 (Amazon Review), Table 13 (DomainNet R→C), Table 14 (DomainNet R→P), Table 15 (DomainNet R→S) and Table 16 (Otto). Our results show that the proposed method ASPEST consistently outperforms the baselines across different image, text and structured datasets.

| Dataset | MNIST→SVHN | | | | | | | | |
|---|---|---|---|---|---|---|---|---|---|
| Metric | $cov\|acc \geq 90\%$ ↑ | | | $acc\|cov \geq 90\%$ ↑ | | | AUACC ↑ | | |
| Labeling Budget | 100 | 500 | 1000 | 100 | 500 | 1000 | 100 | 500 | 1000 |
| SR+Uniform | 0.00±0.0 | 51.46±3.7 | 75.57±0.9 | 58.03±1.5 | 76.69±1.2 | 84.39±0.2 | 74.08±1.5 | 88.80±0.8 | 93.57±0.2 |
| SR+Confidence | 0.00±0.0 | 55.32±5.1 | 82.22±1.3 | 47.66±3.4 | 79.02±0.7 | 87.19±0.4 | 64.14±2.8 | 89.93±0.6 | 94.62±0.2 |
| SR+Entropy | 0.00±0.0 | 0.00±0.0 | 75.08±2.4 | 47.93±7.0 | 77.09±1.0 | 84.81±0.7 | 65.88±4.7 | 88.19±0.8 | 93.37±0.5 |
| SR+Margin | 0.00±0.0 | 63.60±2.7 | 82.19±0.3 | 61.39±0.5 | 80.96±0.9 | 86.97±0.2 | 76.79±0.5 | 91.24±0.5 | 94.82±0.1 |
| SR+kCG | 2.52±1.3 | 23.04±0.3 | 38.97±2.6 | 34.57±4.4 | 52.76±1.1 | 64.34±4.8 | 48.83±7.2 | 73.65±1.0 | 83.16±2.0 |
| SR+CLUE | 0.00±0.0 | 62.03±2.4 | 81.29±1.1 | 57.35±1.9 | 79.55±0.8 | 86.28±0.5 | 72.72±1.9 | 90.98±0.5 | 94.99±0.2 |
| SR+BADGE | 0.00±0.0 | 62.55±4.4 | 82.39±2.8 | 59.82±1.7 | 79.49±1.6 | 86.96±0.9 | 76.06±1.6 | 91.09±0.9 | 95.16±0.6 |
| DE+Uniform | 24.71±5.6 | 68.98±1.6 | 83.67±0.1 | 63.22±1.7 | 81.67±0.4 | 87.32±0.1 | 79.36±1.7 | 92.47±0.2 | 95.48±0.0 |
| DE+Entropy | 6.24±8.8 | 63.30±6.5 | 84.62±1.5 | 56.61±0.6 | 80.16±2.0 | 88.05±0.5 | 72.51±1.5 | 91.21±1.4 | 95.45±0.5 |
| DE+Confidence | 14.92±5.1 | 67.87±1.4 | 89.41±0.3 | 61.11±2.9 | 81.80±0.5 | 89.75±0.1 | 75.85±3.0 | 92.16±0.2 | 96.19±0.1 |
| DE+Margin | 21.59±3.8 | 77.84±2.8 | 92.75±0.3 | 62.88±1.2 | 85.11±1.1 | 91.17±0.1 | 78.59±1.4 | 94.31±0.6 | 97.00±0.0 |
| DE+Avg-KLD | 10.98±4.6 | 61.45±3.4 | 88.06±2.2 | 54.80±1.6 | 78.21±1.6 | 89.23±0.9 | 71.67±2.2 | 90.92±0.8 | 96.23±0.4 |
| DE+CLUE | 22.34±1.4 | 69.23±1.9 | 82.80±1.0 | 59.47±1.3 | 81.05±0.9 | 86.78±0.4 | 76.88±1.0 | 92.70±0.5 | 95.56±0.2 |
| DE+BADGE | 22.02±4.5 | 72.31±1.2 | 88.23±0.4 | 61.23±1.9 | 82.69±0.5 | 89.15±0.2 | 77.65±1.9 | 93.38±0.2 | 96.51±0.1 |
| ASPEST (ours) | **52.10**±4.0 | **89.22**±0.9 | **98.70**±0.4 | **76.10**±1.5 | **89.62**±0.4 | **93.92**±0.3 | **88.84**±1.0 | **96.62**±0.2 | **98.06**±0.1 |

Table 9: Results of comparing ASPEST to the baselines on MNIST→SVHN. The mean and std of each metric over three random runs are reported (mean±std). All numbers are percentages. **Bold** numbers are superior results.

| Dataset | CIFAR-10→CINIC-10 | | | | | | | | |
|---|---|---|---|---|---|---|---|---|---|
| Metric | $cov\|acc \geq 90\%$ ↑ | | | $acc\|cov \geq 90\%$ ↑ | | | AUACC ↑ | | |
| Labeling Budget | 500 | 1000 | 2000 | 500 | 1000 | 2000 | 500 | 1000 | 2000 |
| SR+Uniform | 57.43±0.2 | 57.15±0.6 | 58.37±0.7 | 75.67±0.2 | 75.69±0.1 | 76.11±0.3 | 89.77±0.0 | 89.81±0.1 | 90.09±0.2 |
| SR+Confidence | 57.96±0.6 | 57.05±0.7 | 61.11±1.1 | 76.49±0.2 | 76.87±0.2 | 78.77±0.4 | 90.00±0.2 | 89.92±0.2 | 90.91±0.3 |
| SR+Entropy | 57.78±0.7 | 57.07±1.4 | 61.07±0.4 | 76.57±0.3 | 76.71±0.5 | 78.85±0.2 | 90.01±0.2 | 89.94±0.3 | 90.88±0.0 |
| SR+Margin | 57.72±0.8 | 57.98±0.7 | 61.71±0.2 | 76.24±0.2 | 76.90±0.2 | 78.42±0.2 | 89.95±0.2 | 90.14±0.1 | 91.02±0.0 |
| SR+kCG | 57.90±0.5 | 57.81±0.7 | 60.36±0.3 | 75.59±0.1 | 75.73±0.2 | 76.68±0.2 | 89.78±0.1 | 89.79±0.2 | 90.41±0.2 |
| SR+CLUE | 57.29±0.5 | 58.89±0.5 | 62.28±0.2 | 75.74±0.2 | 76.68±0.3 | 78.10±0.2 | 89.67±0.2 | 90.15±0.1 | 91.03±0.1 |
| SR+BADGE | 58.58±0.6 | 58.63±0.3 | 61.95±0.4 | 76.33±0.5 | 76.58±0.1 | 78.26±0.2 | 90.05±0.2 | 90.16±0.1 | 90.99±0.0 |
| DE+Uniform | 58.06±0.3 | 58.72±0.1 | 59.54±0.3 | 76.65±0.1 | 77.06±0.2 | 77.46±0.1 | 90.26±0.1 | 90.45±0.1 | 90.73±0.1 |
| DE+Entropy | 58.91±0.6 | 60.96±0.2 | 63.85±0.2 | 77.66±0.1 | 79.14±0.1 | 80.82±0.2 | 90.55±0.1 | 91.16±0.1 | 91.89±0.0 |
| DE+Confidence | 58.53±0.3 | 61.03±0.6 | 64.42±0.2 | 77.73±0.2 | 79.00±0.1 | 80.87±0.0 | 90.53±0.0 | 91.11±0.1 | 91.96±0.0 |
| DE+Margin | 58.76±0.5 | 61.60±0.5 | 64.92±0.5 | 77.61±0.2 | 78.91±0.1 | 80.59±0.1 | 90.56±0.1 | 91.11±0.1 | 91.98±0.1 |
| DE+Avg-KLD | 59.99±0.6 | 62.05±0.3 | 65.02±0.5 | 77.84±0.1 | 79.15±0.0 | 81.04±0.1 | 90.74±0.1 | 91.30±0.1 | 92.10±0.1 |
| DE+CLUE | 59.27±0.1 | 61.16±0.4 | 64.42±0.0 | 77.19±0.1 | 78.37±0.2 | 79.44±0.1 | 90.44±0.1 | 91.03±0.1 | 91.74±0.0 |
| DE+BADGE | 59.37±0.4 | 61.61±0.1 | 64.53±0.4 | 77.13±0.1 | 78.33±0.2 | 79.44±0.3 | 90.49±0.1 | 91.12±0.0 | 91.78±0.1 |
| ASPEST (ours) | **60.38**±0.3 | **63.34**±0.2 | **66.81**±0.3 | **78.23**±0.1 | **79.49**±0.1 | **81.25**±0.1 | **90.95**±0.0 | **91.60**±0.0 | **92.33**±0.1 |

Table 10: Results of comparing ASPEST to the baselines on CIFAR-10→CINIC-10. The mean and std of each metric over three random runs are reported (mean±std). All numbers are percentages. **Bold** numbers are superior results.

### F.4 Effect of combining selective prediction with active learning

Selective prediction without active learning corresponds to the case where the labeling budget $M = 0$ and the selected set $B^* = \emptyset$. To make fair comparisons with selective prediction methods without active learning, we define a new coverage metric:

$$cov^*(f_s, \tau) = \mathbb{E}_{\mathbf{x} \sim U_X} \mathbb{I}[g(\mathbf{x}) \geq \tau \wedge \mathbf{x} \notin B^*] \tag{33}$$

| Dataset | FMoW | | | | | | | | |
|---|---|---|---|---|---|---|---|---|---|
| Metric | $cov\|acc \geq 70\%$ ↑ | | | $acc\|cov \geq 70\%$ ↑ | | | AUACC ↑ | | |
| Labeling Budget | 500 | 1000 | 2000 | 500 | 1000 | 2000 | 500 | 1000 | 2000 |
| SR+Uniform | 38.50±0.7 | 42.00±0.5 | 52.34±1.1 | 51.76±0.7 | 54.27±0.2 | 60.31±0.7 | 65.75±0.4 | 67.67±0.3 | 72.73±0.3 |
| SR+Confidence | 37.34±0.3 | 42.28±1.2 | 53.72±0.7 | 52.24±0.1 | 55.52±0.5 | 61.76±0.4 | 65.57±0.1 | 68.03±0.5 | 73.14±0.5 |
| SR+Entropy | 37.42±0.3 | 42.08±0.2 | 51.18±0.4 | 51.74±0.4 | 54.94±0.2 | 60.62±0.2 | 65.31±0.2 | 68.00±0.1 | 71.99±0.2 |
| SR+Margin | 38.40±1.4 | 44.67±0.7 | 55.68±1.5 | 52.88±0.3 | 56.66±0.4 | 62.98±0.7 | 66.11±0.6 | 69.12±0.4 | 73.86±0.5 |
| SR+kCG | 36.50±0.8 | 39.76±1.2 | 45.87±0.6 | 49.36±0.7 | 51.45±0.5 | 55.47±0.1 | 64.34±0.5 | 66.21±0.6 | 69.63±0.2 |
| SR+CLUE | 38.65±0.7 | 44.50±1.8 | 54.71±0.5 | 52.23±0.4 | 55.54±1.0 | 61.13±0.4 | 65.78±0.3 | 68.76±0.9 | 73.80±0.1 |
| SR+BADGE | 40.47±1.5 | 45.65±1.2 | 57.59±0.4 | 53.08±1.0 | 56.63±0.3 | 63.57±0.2 | 66.74±0.8 | 69.43±0.6 | 74.76±0.2 |
| DE+Uniform | 44.74±0.4 | 51.57±1.1 | 61.92±0.4 | 56.39±0.5 | 60.01±0.5 | 65.74±0.2 | 69.44±0.3 | 72.48±0.5 | 77.02±0.1 |
| DE+Entropy | 43.76±0.3 | 50.52±1.4 | 62.73±0.4 | 56.29±0.3 | 60.31±0.3 | 66.53±0.2 | 69.02±0.1 | 72.10±0.5 | 76.65±0.2 |
| DE+Confidence | 45.23±0.6 | 50.11±0.9 | 64.29±0.3 | 57.18±0.4 | 60.46±0.3 | 67.46±0.0 | 69.80±0.3 | 72.11±0.4 | 77.37±0.1 |
| DE+Margin | 46.35±0.6 | 54.79±1.3 | 69.70±0.8 | 57.84±0.3 | 62.43±0.5 | 69.87±0.4 | 70.18±0.3 | 73.62±0.3 | 78.88±0.4 |
| DE+Avg-KLD | 46.29±0.3 | 53.63±0.8 | 68.18±0.9 | 57.75±0.4 | 61.60±0.3 | 69.11±0.4 | 70.16±0.1 | 73.09±0.2 | 78.48±0.3 |
| DE+CLUE | 45.22±0.2 | 49.97±0.3 | 58.05±0.5 | 56.39±0.1 | 59.05±0.1 | 63.23±0.4 | 69.53±0.0 | 71.95±0.1 | 75.72±0.3 |
| DE+BADGE | 47.39±0.7 | 53.83±0.7 | 66.45±0.8 | 57.71±0.4 | 61.16±0.2 | 68.13±0.4 | 70.59±0.4 | 73.40±0.3 | 78.66±0.1 |
| ASPEST (ours) | **53.05**±0.4 | **59.86**±0.4 | **76.52**±0.6 | **61.18**±0.2 | **65.18**±0.2 | **72.86**±0.3 | **71.12**±0.2 | **74.25**±0.2 | **79.93**±0.1 |

Table 11: Results of comparing ASPEST to the baselines on FMoW. The mean and std of each metric over three random runs are reported (mean±std). All numbers are percentages. **Bold** numbers are superior results.

| Dataset | Amazon Review | | | | | | | | |
|---|---|---|---|---|---|---|---|---|---|
| Metric | $cov\|acc \geq 80\%$ ↑ | | | $acc\|cov \geq 80\%$ ↑ | | | AUACC ↑ | | |
| Labeling Budget | 500 | 1000 | 2000 | 500 | 1000 | 2000 | 500 | 1000 | 2000 |
| SR+Uniform | 13.71±11.3 | 24.10±5.3 | 24.87±2.6 | 65.13±0.8 | 66.33±0.6 | 66.26±0.3 | 72.71±1.5 | 73.64±0.7 | 73.53±0.7 |
| SR+Confidence | 11.28±8.9 | 17.96±4.0 | 33.19±1.4 | 65.15±0.7 | 66.29±0.4 | 68.94±0.1 | 72.89±0.7 | 73.25±0.7 | 76.17±0.2 |
| SR+Entropy | 5.55±7.8 | 13.32±9.5 | 25.47±1.8 | 65.11±1.1 | 66.56±0.7 | 67.31±0.7 | 71.96±1.6 | 72.53±1.1 | 74.19±0.5 |
| SR+Margin | 14.48±10.9 | 22.61±4.2 | 28.35±6.1 | 65.75±0.5 | 66.31±0.4 | 68.15±0.4 | 73.25±1.0 | 73.65±0.5 | 75.17±0.8 |
| SR+kCG | 20.02±11.0 | 17.02±12.2 | 29.08±4.2 | 64.03±3.1 | 66.17±0.5 | 66.63±1.0 | 72.34±3.2 | 74.35±0.7 | 74.49±1.0 |
| SR+CLUE | 4.15±5.9 | 25.15±4.9 | 31.88±2.1 | 66.17±0.4 | 66.30±0.4 | 67.12±0.7 | 73.43±0.4 | 74.07±0.7 | 75.29±0.9 |
| SR+BADGE | 22.58±0.4 | 23.78±6.4 | 30.71±4.6 | 66.29±0.4 | 66.31±0.6 | 68.58±0.7 | 73.80±0.6 | 74.00±1.0 | 75.76±0.8 |
| DE+Uniform | 34.35±1.4 | 33.15±1.1 | 36.55±1.8 | 68.13±0.4 | 68.12±0.6 | 68.88±0.2 | 76.20±0.3 | 76.16±0.4 | 77.07±0.3 |
| DE+Entropy | 31.74±1.4 | 36.29±1.6 | 40.33±1.7 | 68.19±0.3 | 69.44±0.2 | 71.27±0.3 | 75.98±0.4 | 77.10±0.3 | 78.53±0.3 |
| DE+Confidence | 35.12±1.8 | 34.48±1.4 | 40.46±0.5 | 69.07±0.3 | 69.47±0.2 | 71.08±0.2 | 76.63±0.2 | 76.87±0.3 | 78.27±0.1 |
| DE+Margin | 33.42±1.3 | 35.03±1.3 | 41.20±0.4 | 68.45±0.3 | 69.30±0.2 | 70.88±0.1 | 76.18±0.2 | 76.91±0.3 | 78.31±0.1 |
| DE+Avg-KLD | 33.03±1.5 | 38.55±3.2 | 41.75±1.8 | 68.63±0.3 | 69.95±0.4 | 71.10±0.3 | 76.21±0.4 | 77.62±0.6 | 78.62±0.3 |
| DE+CLUE | 33.92±3.0 | 35.27±1.4 | 34.83±3.1 | 68.09±0.3 | 68.07±0.3 | 68.40±0.6 | 76.27±0.6 | 76.65±0.3 | 76.69±0.7 |
| DE+BADGE | 32.23±3.7 | 36.18±1.5 | 40.58±3.3 | 68.34±0.4 | 68.87±0.2 | 70.29±0.3 | 76.13±0.7 | 77.09±0.2 | 78.44±0.5 |
| ASPEST (ours) | **38.44**±0.7 | **40.96**±0.8 | **45.77**±0.1 | **69.31**±0.3 | **70.17**±0.2 | **71.60**±0.2 | **77.69**±0.1 | **78.35**±0.2 | **79.51**±0.2 |

Table 12: Results of comparing ASPEST to the baselines on Amazon Review. The mean and std of each metric over three random runs are reported (mean±std). All numbers are percentages. **Bold** numbers are superior results.

The range of $cov^*(f_s, \tau)$ is $[0, 1 - \frac{M}{n}]$, where $M = |B^*|$ and $n = |U_X|$. If we use a larger labeling budget $M$ for active learning, then the upper bound of $cov^*(f_s, \tau)$ will be smaller. Thus, in order to beat selective classification methods without active learning, active selective prediction methods need to use a small labeling budget to achieve significant accuracy and coverage improvement. We still use the accuracy metric defined in (4).

We then define a new maximum coverage at a target accuracy $t_a$ metric as:

$$\max_{\tau} \quad cov^*(f_s, \tau), \quad s.t. \quad acc(f_s, \tau) \geq t_a \tag{34}$$

We denote this metric as $cov^*|acc \geq t_a$. We also measure the accuracy of $f$ on the remaining unlabeled test data for different approaches.

The results under these metrics are shown in Table 3 (MNIST→SVHN), Table 17 (CIFAR-10→CINIC-10 and Otto), Table 18 (FMoW and Amazon Review) and Table 19 (DomainNet). The results show that the coverage achieved by the selective prediction baselines SR and DE is usually low and the active learning baselines Margin, kCG, CLUE and BADGE fail to achieve the target accuracy with a labeling budget of 1K.

| Dataset | DomainNet R→C (easy) | | | | | | | | |
|---|---|---|---|---|---|---|---|---|---|
| Metric | $cov\|acc \geq 80\%$ ↑ | | | $acc\|cov \geq 80\%$ ↑ | | | AUACC ↑ | | |
| Labeling Budget | 500 | 1000 | 2000 | 500 | 1000 | 2000 | 500 | 1000 | 2000 |
| SR+Uniform | 25.56±0.6 | 27.68±0.8 | 29.86±0.0 | 43.63±0.4 | 45.57±0.3 | 47.27±0.4 | 63.31±0.4 | 65.11±0.5 | 66.70±0.2 |
| SR+Confidence | 25.96±0.2 | 27.80±1.2 | 32.51±1.3 | 44.90±0.8 | 47.26±0.4 | 52.04±0.8 | 64.20±0.6 | 65.88±0.6 | 69.70±0.7 |
| SR+Entropy | 25.44±1.0 | 27.79±0.4 | 33.51±1.1 | 44.46±0.7 | 46.96±0.3 | 52.25±0.5 | 63.52±0.6 | 65.72±0.2 | 70.03±0.5 |
| SR+Margin | 26.28±1.2 | 27.77±1.0 | 32.92±0.4 | 45.24±1.0 | 47.12±0.7 | 52.29±0.4 | 64.37±0.8 | 65.91±0.6 | 70.01±0.4 |
| SR+kCG | 21.12±0.3 | 21.79±0.4 | 23.43±0.5 | 39.19±0.1 | 40.59±0.4 | 41.11±0.3 | 58.88±0.0 | 60.11±0.4 | 60.89±0.1 |
| SR+CLUE | 27.17±0.8 | 29.78±0.8 | 34.82±0.6 | 44.57±0.7 | 46.79±0.1 | 49.70±0.3 | 64.38±0.6 | 66.47±0.3 | 69.59±0.1 |
| SR+BADGE | 27.78±0.8 | 30.78±0.6 | 36.00±0.6 | 45.36±0.6 | 48.43±0.6 | 53.00±0.4 | 64.90±0.5 | 67.56±0.4 | 71.39±0.4 |
| DE+Uniform | 30.82±0.8 | 33.05±0.4 | 36.80±0.2 | 48.19±0.3 | 50.09±0.3 | 52.98±0.5 | 67.60±0.4 | 69.31±0.3 | 71.64±0.4 |
| DE+Entropy | 29.13±0.9 | 34.07±0.3 | 40.82±0.3 | 48.67±0.4 | 51.66±0.2 | 57.81±0.2 | 67.48±0.3 | 70.05±0.2 | 74.64±0.2 |
| DE+Confidence | 29.90±0.8 | 33.73±0.2 | 40.80±0.2 | 48.60±0.3 | 52.03±0.3 | 58.43±0.1 | 67.45±0.3 | 70.19±0.2 | 74.80±0.1 |
| DE+Margin | 31.82±1.3 | 35.68±0.2 | 43.39±0.7 | 50.12±0.4 | 53.19±0.4 | 59.17±0.2 | 68.85±0.4 | 71.29±0.3 | 75.79±0.3 |
| DE+Avg-KLD | 32.23±0.2 | 36.09±0.6 | 44.00±0.5 | 49.81±0.3 | 53.38±0.3 | 58.93±0.1 | 68.73±0.2 | 71.40±0.2 | 75.73±0.2 |
| DE+CLUE | 30.80±0.3 | 33.04±0.4 | 35.52±0.2 | 48.56±0.3 | 49.91±0.3 | 51.40±0.2 | 67.82±0.2 | 69.10±0.2 | 70.62±0.2 |
| DE+BADGE | 30.16±1.3 | 36.18±0.3 | 43.34±0.3 | 49.78±0.3 | 53.26±0.1 | 58.65±0.4 | 68.46±0.3 | 71.35±0.2 | 75.37±0.3 |
| ASPEST (ours) | **37.38**±0.1 | **39.98**±0.3 | **48.29**±1.0 | **54.56**±0.3 | **56.95**±0.1 | **62.69**±0.2 | **71.61**±0.2 | **73.27**±0.2 | **77.40**±0.4 |

Table 13: Results of comparing ASPEST to the baselines on DomainNet R→C. The mean and std of each metric over three random runs are reported (mean±std). All numbers are percentages. **Bold** numbers are superior results.

| Dataset | DomainNet R→P (medium) | | | | | | | | |
|---|---|---|---|---|---|---|---|---|---|
| Metric | $cov\|acc \geq 70\%$ ↑ | | | $acc\|cov \geq 70\%$ ↑ | | | AUACC ↑ | | |
| Labeling Budget | 500 | 1000 | 2000 | 500 | 1000 | 2000 | 500 | 1000 | 2000 |
| SR+Uniform | 21.01±1.0 | 21.35±0.3 | 22.64±0.5 | 36.78±0.6 | 37.18±0.2 | 38.20±0.4 | 51.87±0.7 | 52.31±0.0 | 53.34±0.4 |
| SR+Confidence | 20.64±0.6 | 22.15±0.8 | 23.60±0.6 | 37.01±0.3 | 38.46±0.7 | 40.23±0.4 | 51.77±0.3 | 53.33±0.8 | 54.80±0.5 |
| SR+Entropy | 20.76±0.7 | 22.11±0.3 | 23.56±0.3 | 37.09±0.2 | 38.38±0.3 | 40.30±0.1 | 51.86±0.4 | 53.29±0.3 | 54.81±0.2 |
| SR+Margin | 21.43±0.4 | 23.29±0.3 | 24.70±1.0 | 37.21±0.2 | 39.15±0.4 | 40.81±0.4 | 52.33±0.1 | 54.09±0.3 | 55.70±0.4 |
| SR+kCG | 17.33±0.4 | 17.62±0.2 | 18.49±0.2 | 33.97±0.3 | 34.12±0.1 | 34.36±0.1 | 48.61±0.5 | 48.65±0.2 | 49.25±0.2 |
| SR+CLUE | 21.15±0.6 | 22.49±0.5 | 24.84±0.7 | 36.96±0.2 | 37.93±0.5 | 39.31±0.4 | 51.97±0.4 | 53.20±0.5 | 54.84±0.5 |
| SR+BADGE | 20.07±0.3 | 22.21±0.5 | 24.92±0.2 | 36.10±0.1 | 38.11±0.4 | 40.40±0.5 | 50.99±0.0 | 53.10±0.4 | 55.40±0.4 |
| DE+Uniform | 25.42±0.2 | 26.38±0.2 | 28.83±0.3 | 40.83±0.1 | 41.66±0.2 | 43.93±0.2 | 55.86±0.1 | 56.62±0.1 | 58.80±0.2 |
| DE+Entropy | 25.74±0.4 | 27.11±0.4 | 30.39±0.1 | 41.34±0.1 | 42.92±0.3 | 45.92±0.3 | 56.06±0.2 | 57.51±0.3 | 60.10±0.2 |
| DE+Confidence | 25.69±0.4 | 27.38±0.7 | 30.47±0.1 | 41.45±0.2 | 43.12±0.3 | 45.88±0.1 | 56.13±0.2 | 57.68±0.3 | 60.20±0.2 |
| DE+Margin | 25.78±0.3 | 27.88±0.5 | 31.03±0.4 | 41.26±0.2 | 43.13±0.3 | 46.23±0.4 | 56.23±0.2 | 57.90±0.3 | 60.49±0.3 |
| DE+Avg-KLD | 26.30±0.7 | 28.00±0.1 | 31.97±0.2 | 41.80±0.3 | 43.17±0.1 | 46.32±0.2 | 56.65±0.3 | 57.99±0.1 | 60.82±0.2 |
| DE+CLUE | 25.38±0.6 | 26.65±0.4 | 27.89±0.1 | 40.86±0.3 | 41.62±0.2 | 42.46±0.1 | 55.79±0.4 | 56.65±0.2 | 57.71±0.1 |
| DE+BADGE | 26.27±0.7 | 27.69±0.1 | 31.84±0.2 | 42.02±0.6 | 43.41±0.2 | 46.37±0.1 | 56.67±0.5 | 58.03±0.1 | 60.84±0.1 |
| ASPEST (ours) | **29.69**±0.1 | **32.50**±0.3 | **35.46**±0.6 | **44.96**±0.1 | **46.77**±0.2 | **49.42**±0.1 | **58.74**±0.0 | **60.36**±0.0 | **62.84**±0.2 |

Table 14: Results of comparing ASPEST to the baselines on DomainNet R→P. The mean and std of each metric over three random runs are reported (mean±std). All numbers are percentages. **Bold** numbers are superior results.

In contrast, the proposed active selective prediction method ASPEST achieves the target accuracy with a much higher coverage.

### F.5 Effect of joint training

In the problem setup, we assume that we have access to the training dataset $\mathcal{D}^{tr}$ and can use joint training to improve selective prediction performance. In this section, we perform experiments to study the effect of joint training and the effect of the loss coefficient $\lambda$ when performing joint training. We consider three active selective prediction methods: SR+margin (Algorithm 1 with margin sampling), DE+margin (Algorithm 2 with margin sampling), and ASPEST (Algorithm 3). We consider $\lambda \in \{0, 0.5, 1.0, 2.0\}$. When $\lambda = 0$, we don't use joint training; when $\lambda > 0$, we use joint training. The results are shown in Table 20. From the results, we can see that using joint training (i.e., when $\lambda > 0$) can improve performance, especially when the labeling budget is small. Also, setting a too large value for $\lambda$ (e.g., $\lambda = 2$) will lead to worse performance. Setting $\lambda = 0.5$ or 1 usually leads to better performance. In our experiments, we simply set $\lambda = 1$ by default.

| Dataset | DomainNet R→S (hard) | | | | | | | | |
|---|---|---|---|---|---|---|---|---|---|
| Metric | $cov\|acc \geq 70\%$ ↑ | | | $acc\|cov \geq 70\%$ ↑ | | | AUACC ↑ | | |
| Labeling Budget | 500 | 1000 | 2000 | 500 | 1000 | 2000 | 500 | 1000 | 2000 |
| SR+Uniform | 12.12±0.7 | 12.42±0.4 | 15.88±0.2 | 27.01±0.6 | 27.74±0.3 | 31.29±0.3 | 41.12±0.8 | 41.89±0.2 | 46.17±0.3 |
| SR+Confidence | 11.06±1.1 | 11.48±0.5 | 14.49±1.5 | 26.53±1.4 | 27.98±0.2 | 31.31±0.7 | 40.26±1.6 | 41.65±0.2 | 45.46±1.1 |
| SR+Entropy | 10.91±0.3 | 12.45±0.6 | 14.65±0.6 | 26.84±0.5 | 28.72±0.5 | 31.07±0.6 | 40.47±0.6 | 42.61±0.8 | 45.31±0.4 |
| SR+Margin | 12.23±0.4 | 13.06±0.4 | 15.31±0.4 | 27.87±0.2 | 29.19±0.4 | 31.51±0.8 | 41.91±0.3 | 43.22±0.4 | 45.97±0.8 |
| SR+kCG | 9.03±0.2 | 9.76±0.2 | 11.41±0.2 | 23.32±0.4 | 24.06±0.4 | 25.68±0.4 | 36.63±0.3 | 37.57±0.4 | 39.80±0.3 |
| SR+CLUE | 12.39±0.3 | 14.17±1.0 | 15.80±0.8 | 27.82±0.4 | 29.68±0.4 | 30.62±0.8 | 42.00±0.4 | 44.19±0.7 | 45.58±0.9 |
| SR+BADGE | 12.18±0.9 | 13.13±1.0 | 15.83±0.7 | 27.68±1.0 | 28.96±0.7 | 32.00±0.4 | 41.72±1.1 | 43.28±0.9 | 46.60±0.6 |
| DE+Uniform | 15.91±0.5 | 17.55±0.4 | 21.33±0.3 | 31.37±0.5 | 32.57±0.4 | 36.12±0.2 | 46.28±0.5 | 47.79±0.4 | 51.64±0.2 |
| DE+Entropy | 13.70±0.3 | 16.31±0.5 | 19.58±0.4 | 30.38±0.4 | 32.45±0.2 | 36.18±0.2 | 44.79±0.5 | 47.15±0.2 | 50.87±0.3 |
| DE+Confidence | 13.73±0.2 | 16.21±0.2 | 19.22±0.4 | 30.55±0.3 | 33.02±0.1 | 36.29±0.5 | 45.05±0.3 | 47.59±0.0 | 50.84±0.4 |
| DE+Margin | 14.99±0.2 | 17.45±0.4 | 21.74±0.7 | 31.67±0.5 | 33.51±0.5 | 37.88±0.3 | 46.38±0.5 | 48.44±0.5 | 52.78±0.4 |
| DE+Avg-KLD | 15.75±0.5 | 18.14±0.7 | 22.15±0.3 | 31.36±0.2 | 33.79±0.2 | 37.96±0.2 | 46.29±0.1 | 48.77±0.3 | 53.02±0.3 |
| DE+CLUE | 14.76±0.5 | 17.38±0.1 | 19.75±0.4 | 31.05±0.4 | 32.58±0.2 | 34.61±0.4 | 45.80±0.3 | 47.74±0.1 | 50.09±0.2 |
| DE+BADGE | 14.97±0.1 | 17.49±0.3 | 21.71±0.3 | 31.35±0.2 | 33.46±0.1 | 37.35±0.3 | 46.03±0.1 | 48.31±0.1 | 52.33±0.2 |
| ASPEST (ours) | **17.86**±0.4 | **20.42**±0.4 | **25.87**±0.4 | **35.17**±0.1 | **37.28**±0.3 | **41.46**±0.2 | **49.62**±0.1 | **51.61**±0.4 | **55.90**±0.2 |

Table 15: Results of comparing ASPEST to the baselines on DomainNet R→S. The mean and std of each metric over three random runs are reported (mean±std). All numbers are percentages. **Bold** numbers are superior results.

| Dataset | Otto | | | | | | | | |
|---|---|---|---|---|---|---|---|---|---|
| Metric | $cov\|acc \geq 80\%$ ↑ | | | $acc\|cov \geq 80\%$ ↑ | | | AUACC ↑ | | |
| Labeling Budget | 500 | 1000 | 2000 | 500 | 1000 | 2000 | 500 | 1000 | 2000 |
| SR+Uniform | 63.58±0.7 | 64.06±0.4 | 67.49±0.9 | 73.56±0.3 | 73.57±0.6 | 75.21±0.2 | 84.46±0.2 | 84.61±0.3 | 85.72±0.2 |
| SR+Confidence | 69.63±1.7 | 73.41±0.6 | 84.19±0.5 | 75.96±0.5 | 77.57±0.2 | 81.39±0.2 | 85.91±0.3 | 86.86±0.1 | 88.93±0.1 |
| SR+Entropy | 67.79±0.8 | 73.83±1.0 | 83.12±0.7 | 75.43±0.4 | 77.91±0.3 | 81.07±0.2 | 85.41±0.3 | 86.94±0.2 | 88.86±0.1 |
| SR+Margin | 68.10±0.1 | 74.10±0.4 | 82.53±0.2 | 75.52±0.0 | 77.66±0.1 | 80.93±0.1 | 85.56±0.1 | 86.99±0.1 | 88.83±0.1 |
| SR+kCG | 64.84±0.7 | 62.90±1.1 | 59.85±1.0 | 73.75±0.3 | 73.03±0.2 | 71.90±0.3 | 85.08±0.2 | 84.67±0.2 | 83.79±0.3 |
| SR+CLUE | 68.21±1.2 | 70.85±0.6 | 78.26±0.9 | 75.26±0.5 | 76.32±0.2 | 79.30±0.3 | 85.82±0.3 | 86.69±0.2 | 88.53±0.2 |
| SR+BADGE | 67.23±1.0 | 73.52±0.2 | 83.17±0.4 | 74.74±0.3 | 77.43±0.2 | 81.20±0.2 | 85.41±0.3 | 87.10±0.2 | 89.25±0.1 |
| DE+Uniform | 70.74±0.5 | 72.20±0.6 | 75.58±0.5 | 76.40±0.1 | 77.06±0.2 | 78.35±0.2 | 86.78±0.1 | 87.26±0.1 | 88.11±0.1 |
| DE+Entropy | 75.71±0.3 | 80.91±0.2 | 92.62±0.2 | 78.44±0.1 | 80.29±0.1 | 84.05±0.1 | 87.87±0.1 | 88.77±0.1 | 90.99±0.1 |
| DE+Confidence | 75.52±0.2 | 81.69±0.7 | 92.15±0.9 | 78.28±0.1 | 80.49±0.2 | 83.83±0.1 | 87.84±0.1 | 89.05±0.1 | 90.98±0.1 |
| DE+Margin | 75.49±0.8 | 81.36±0.8 | 92.49±0.4 | 78.41±0.3 | 80.50±0.2 | 84.06±0.2 | 87.89±0.2 | 89.10±0.2 | 90.95±0.2 |
| DE+Avg-KLD | 75.91±0.2 | 80.97±0.5 | 91.94±0.8 | 78.50±0.1 | 80.33±0.2 | 83.80±0.2 | 87.89±0.0 | 89.06±0.1 | 90.98±0.1 |
| DE+CLUE | 69.66±0.5 | 70.52±0.1 | 70.17±0.4 | 76.09±0.3 | 76.32±0.1 | 76.31±0.2 | 86.67±0.1 | 87.11±0.0 | 87.06±0.1 |
| DE+BADGE | 73.23±0.2 | 77.89±0.6 | 86.32±0.5 | 77.33±0.1 | 79.21±0.3 | 82.32±0.2 | 87.55±0.1 | 88.75±0.1 | 90.58±0.0 |
| ASPEST (ours) | **77.85**±0.2 | **84.20**±0.6 | **94.26**±0.6 | **79.28**±0.1 | **81.40**±0.1 | **84.62**±0.1 | **88.28**±0.1 | **89.61**±0.1 | **91.49**±0.0 |

Table 16: Results of comparing ASPEST to the baselines on Otto. The mean and std of each metric over three random runs are reported (mean±std). All numbers are percentages. **Bold** numbers are superior results.

| Dataset | | CIFAR-10→CINIC-10 | | Otto | |
|---|---|---|---|---|---|
| Method | | $cov^*\|acc \geq 90\%$ ↑ | Accuracy of $f$ ↑ | $cov^*\|acc \geq 80\%$ ↑ | Accuracy of $f$ ↑ |
| Selective Prediction | SR (M=0) | 57.43±0.0 | 71.05±0.0 | 62.90±0.0 | 66.09±0.0 |
| | DE (M=0) | 56.64±0.2 | 71.33±0.1 | 67.69±0.4 | 68.12±0.3 |
| Active Learning | Margin (M=1000) | N/A | 72.74±0.1 | N/A | 70.19±0.2 |
| | kCG (M=1000) | N/A | 71.16±0.1 | N/A | 65.88±0.2 |
| | CLUE (M=1000) | N/A | 72.32±0.2 | N/A | 68.64±0.2 |
| | BADGE (M=1000) | N/A | 72.11±0.1 | N/A | 69.80±0.4 |
| Active Selective Prediction | ASPEST (M=1000) | **61.23**±0.2 | **74.89**±0.1 | **77.40**±0.5 | **74.13**±0.1 |

Table 17: Results on CIFAR-10→CINIC-10 and Otto for studying the effect of combining selective prediction with active learning. The mean and std of each metric over three random runs are reported (mean±std). All numbers are percentages. **Bold** numbers are superior results.

| Dataset | | FMoW | | Amazon Review | |
|---|---|---|---|---|---|
| Method | | $cov^*|acc \geq 70\%$ ↑ | Accuracy of $f$ ↑ | $cov^*|acc \geq 80\%$ ↑ | Accuracy of $f$ ↑ |
| Selective Prediction | SR (M=0) | 32.39±0.0 | 38.01±0.0 | 26.79±0.0 | 61.40±0.0 |
| | DE (M=0) | 37.58±0.3 | 41.08±0.0 | 35.81±1.9 | 63.89±0.1 |
| Active Learning | Margin (M=1000) | N/A | 45.17±0.5 | N/A | 62.64±0.6 |
| | kCG (M=1000) | N/A | 40.20±0.3 | N/A | 62.03±0.3 |
| | CLUE (M=1000) | N/A | 43.75±0.7 | N/A | 62.32±0.4 |
| | BADGE (M=1000) | N/A | 44.73±0.3 | N/A | 62.44±0.5 |
| Active Selective Prediction | ASPEST (M=1000) | **57.15**±0.4 | **52.42**±0.1 | **39.14**±0.8 | **65.47**±0.2 |

Table 18: Results on FMoW and Amazon Review for studying the effect of combining selective prediction with active learning. The mean and std of each metric over three random runs are reported (mean±std). All numbers are percentages. **Bold** numbers are superior results.

| Dataset | | DomainNet R→C (easy) | | DomainNet R→P (medium) | | DomainNet R→S (hard) | |
|---|---|---|---|---|---|---|---|
| Method | | $cov^*|acc \geq 80\%$ ↑ | Accuracy of $f$ ↑ | $cov^*|acc \geq 70\%$ ↑ | Accuracy of $f$ ↑ | $cov^*|acc \geq 70\%$ ↑ | Accuracy of $f$ ↑ |
| Selective Prediction | SR (M=0) | 21.50±0.0 | 33.37±0.0 | 18.16±0.0 | 26.29±0.0 | 7.16±0.0 | 15.99±0.0 |
| | DE (M=0) | 26.15±0.2 | 37.12±0.1 | 22.44±0.2 | 29.73±0.1 | 9.90±0.4 | 19.15±0.0 |
| Active Learning | Margin (M=1000) | N/A | 39.42±0.6 | N/A | 29.83±0.3 | N/A | 22.14±0.3 |
| | kCG (M=1000) | N/A | 33.74±0.4 | N/A | 25.92±0.0 | N/A | 18.14±0.3 |
| | CLUE (M=1000) | N/A | 38.97±0.1 | N/A | 28.88±0.5 | N/A | 22.44±0.3 |
| | BADGE (M=1000) | N/A | 40.32±0.5 | N/A | 28.91±0.4 | N/A | 21.87±0.5 |
| Active Selective Prediction | ASPEST (M=1000) | **37.24**±0.3 | **47.76**±0.1 | **31.01**±0.3 | **35.69**±0.1 | **19.45**±0.3 | **28.16**±0.3 |

Table 19: Results on DomainNet R→C, R→P and R→S for studying the effect of combining selective prediction with active learning. The mean and std of each metric over three random runs are reported (mean±std). All numbers are percentages. **Bold** numbers are superior results.

### F.6 Effect of the number of rounds T

In this section, we study the effect of the number of rounds $T$ in active learning. The results in Table 21 show that larger $T$ usually leads to better performance, and the proposed method ASPEST has more improvement as we increase $T$ compared to SR+Margin and DE+Margin. Also, when $T$ is large enough, the improvement becomes minor (or can even be worse). Considering that in practice, we might not be able to set a large $T$ due to resource constraints, we thus set $T = 10$ by default.

### F.7 Effect of the number of models N in the ensemble

In this section, we study the effect of the number of models $N$ in the ensemble for DE+Margin and ASPEST. The results in Table 22 show that larger $N$ usually leads to better results. However, larger $N$ also means a larger computational cost. In our experiments, we simply set $N = 5$ by default.

### F.8 Effect of the upper bound in pseudo-labeled set construction

When constructing the pseudo-labeled set $R$ using Eq. (9), we exclude those test data points with confidence equal to 1. In this section, we study whether setting such an upper bound can improve performance. The results in Table 23 show that when the labeling budget is small, setting such an upper bound can improve performance significantly. However, when the labeling budget is large, setting such an upper bound may not improve the performance. Since we focus on the low labeling budget region, we decide to set such an upper bound for the proposed ASPEST method.

### F.9 Ensemble accuracy after each round of active learning

We evaluate the accuracy of the ensemble model $f_t$ in the ASPEST algorithm after the $t$-th round of active learning. Recall that $f_t$ contains $N$ models $f_t^1, \ldots, f_t^N$ and $f_t(\mathbf{x}) = \arg\max_{k \in \mathcal{Y}} \frac{1}{N} \sum_{j=1}^{N} f_t^j(\mathbf{x} \mid k)$. The results in Table 24 show that after each round of active learning, the accuracy of the ensemble model will be improved significantly.

| Dataset | MNIST→SVHN | | DomainNet R→C (easy) | |
|---|---|---|---|---|
| Metric | AUACC ↑ | | AUACC ↑ | |
| Labeling Budget | 100 | 500 | 500 | 1000 |
| SR+Margin ($\lambda = 0$) | 71.90±3.1 | 90.56±0.8 | 60.05±0.9 | 60.34±1.2 |
| SR+Margin ($\lambda = 0.5$) | 75.54±1.7 | 91.43±0.5 | 64.99±0.7 | 66.81±0.5 |
| SR+Margin ($\lambda = 1$) | 76.79±0.5 | 91.24±0.5 | 64.37±0.8 | 65.91±0.6 |
| SR+Margin ($\lambda = 2$) | 72.71±2.5 | 90.80±0.3 | 64.17±0.3 | 66.21±0.2 |
| DE+Margin ($\lambda = 0$) | 77.12±0.5 | 94.26±0.5 | 66.86±0.5 | 69.29±0.6 |
| DE+Margin ($\lambda = 0.5$) | 79.35±1.4 | 94.22±0.2 | 69.28±0.3 | 71.60±0.2 |
| DE+Margin ($\lambda = 1$) | 78.59±1.4 | 94.31±0.6 | 68.85±0.4 | 71.29±0.3 |
| DE+Margin ($\lambda = 2$) | 77.64±2.3 | 93.81±0.4 | 68.54±0.1 | 71.28±0.2 |
| ASPEST ($\lambda = 0$) | 84.48±2.5 | 96.99±0.2 | 68.61±1.2 | 73.21±1.2 |
| ASPEST ($\lambda = 0.5$) | 86.46±3.1 | 97.01±0.0 | 71.53±0.1 | 73.69±0.1 |
| ASPEST ($\lambda = 1$) | 88.84±1.0 | 96.62±0.2 | 71.61±0.2 | 73.27±0.2 |
| ASPEST ($\lambda = 2$) | 85.46±1.7 | 96.43±0.1 | 70.54±0.3 | 73.02±0.1 |

Table 20: Ablation study results for the effect of using joint training and the effect of the loss coefficient $\lambda$. The mean and std of each metric over three random runs are reported (mean±std). All numbers are percentages.

| Dataset | MNIST→SVHN | | DomainNet R→C (easy) | |
|---|---|---|---|---|
| Metric | AUACC ↑ | | AUACC ↑ | |
| Labeling Budget | 100 | 500 | 500 | 1000 |
| SR+Margin (T=1) | 63.10±2.7 | 75.42±3.6 | 65.16±0.4 | 66.76±0.3 |
| SR+Margin (T=2) | 68.09±3.1 | 87.45±1.6 | 64.64±0.8 | 66.91±0.1 |
| SR+Margin (T=5) | 74.87±1.7 | 91.32±0.5 | 64.35±0.2 | 66.76±0.3 |
| SR+Margin (T=10) | 76.79±0.5 | 91.24±0.5 | 64.37±0.8 | 65.91±0.6 |
| SR+Margin (T=20) | 72.81±1.5 | 90.34±1.3 | 63.65±0.6 | 66.08±0.4 |
| DE+Margin (T=1) | 69.85±0.5 | 82.74±2.1 | 68.39±0.2 | 70.55±0.0 |
| DE+Margin (T=2) | 75.25±1.0 | 90.90±1.0 | 68.79±0.2 | 70.95±0.5 |
| DE+Margin (T=5) | 78.41±0.2 | 93.26±0.3 | 68.80±0.2 | 71.21±0.2 |
| DE+Margin (T=10) | 78.59±1.4 | 94.31±0.6 | 68.85±0.4 | 71.29±0.3 |
| DE+Margin (T=20) | 76.84±0.4 | 94.67±0.2 | 68.50±0.5 | 71.39±0.2 |
| ASPEST (T=1) | 62.53±1.0 | 80.72±1.5 | 69.44±0.1 | 71.79±0.2 |
| ASPEST (T=2) | 75.08±1.4 | 89.70±0.7 | 70.68±0.2 | 72.56±0.3 |
| ASPEST (T=5) | 81.57±1.8 | 95.43±0.1 | 71.23±0.1 | 73.19±0.1 |
| ASPEST (T=10) | 88.84±1.0 | 96.62±0.2 | 71.61±0.2 | 73.27±0.2 |
| ASPEST (T=20) | 91.26±0.9 | 97.32±0.1 | 70.57±0.4 | 73.32±0.3 |

Table 21: Ablation study results for the effect of the number of rounds $T$. The mean and std of each metric over three random runs are reported (mean±std). All numbers are percentages.

## F.10 Empirical analysis for checkpoint ensemble

In this section, we analyze why the proposed checkpoint ensemble can improve selective prediction performance. We postulate the rationales: (1) the checkpoint ensemble can help with generalization; (2) the checkpoint ensemble can help with reducing overconfident wrong predictions.

Regarding (1), when fine-tuning the model on the small set of selected labeled test data, we hope that the fine-tuned model could generalize to remaining unlabeled test data. However, since the selected test set is small, we might have an overfitting issue. So possibly some intermediate checkpoints along the training path achieve better generalization than the end checkpoint. By using checkpoint ensemble, we might get an

| Dataset | MNIST→SVHN | | DomainNet R→C (easy) | |
|---|---|---|---|---|
| Metric | AUACC ↑ | | AUACC ↑ | |
| Labeling Budget | 100 | 500 | 500 | 1000 |
| DE+Margin (N=2) | 67.41±3.9 | 91.20±0.8 | 65.82±0.5 | 67.72±0.4 |
| DE+Margin (N=3) | 77.53±1.5 | 93.41±0.1 | 67.54±0.4 | 69.61±0.2 |
| DE+Margin (N=4) | 74.46±2.7 | 93.65±0.3 | 68.09±0.2 | 70.65±0.3 |
| DE+Margin (N=5) | 78.59±1.4 | 94.31±0.6 | 68.85±0.4 | 71.29±0.3 |
| DE+Margin (N=6) | 79.34±0.7 | 94.40±0.1 | 68.63±0.2 | 71.65±0.3 |
| DE+Margin (N=7) | 80.30±1.5 | 93.97±0.2 | 69.41±0.1 | 71.78±0.3 |
| DE+Margin (N=8) | 78.91±1.5 | 94.52±0.2 | 69.00±0.0 | 71.88±0.4 |
| ASPEST (N=2) | 80.38±1.2 | 96.26±0.0 | 69.14±0.3 | 71.36±0.3 |
| ASPEST (N=3) | 84.86±1.0 | 96.60±0.2 | 69.91±0.2 | 72.25±0.2 |
| ASPEST (N=4) | 84.94±0.3 | 96.76±0.1 | 70.68±0.2 | 73.09±0.2 |
| ASPEST (N=5) | 88.84±1.0 | 96.62±0.2 | 71.61±0.2 | 73.27±0.2 |
| ASPEST (N=6) | 84.51±0.5 | 96.66±0.2 | 71.20±0.2 | 73.42±0.3 |
| ASPEST (N=7) | 86.70±2.3 | 96.90±0.2 | 71.16±0.2 | 73.50±0.1 |
| ASPEST (N=8) | 88.59±0.9 | 97.01±0.1 | 71.62±0.3 | 73.76±0.2 |

Table 22: Ablation study results for the effect of the number of models $N$ in the ensemble. The mean and std of each metric over three random runs are reported (mean±std). All numbers are percentages.

| Dataset | MNIST→SVHN | | DomainNet R→C (easy) | |
|---|---|---|---|---|
| Metric | AUACC ↑ | | AUACC ↑ | |
| Labeling Budget | 100 | 500 | 500 | 1000 |
| ASPEST without upper bound | 86.95±1.4 | 96.59±0.1 | 71.39±0.1 | **73.52**±0.2 |
| ASPEST | **88.84**±1.0 | **96.62**±0.2 | **71.61**±0.2 | 73.27±0.2 |

Table 23: Ablation study results for the effect of setting an upper bound when constructing the pseudo-labeled set $R$ in ASPEST. The mean and std of each metric over three random runs are reported (mean±std). All numbers are percentages. **Bold** numbers are superior results.

ensemble that achieves better generalization to remaining unlabeled test data. Although standard techniques like cross-validation and early stopping can also reduce overfitting, they are not suitable in the active selective prediction setup since the amount of labeled test data is small.

Regarding (2), when fine-tuning the model on the small set of selected labeled test data, the model can get increasingly confident on the test data. Since there exist high-confidence mis-classified test points, incorporating intermediate checkpoints along the training path into the ensemble can reduce the average confidence of the ensemble on those mis-classified test points. By using checkpoint ensemble, we might get an ensemble that has better confidence estimation for selective prediction on the test data.

We perform experiments on the image dataset MNIST→SVHN and the text dataset Amazon Review to verify these two hypotheses. We employ one-round active learning with a labeling budget of 100 samples. We use the margin sampling method for sample selection and fine-tune a single model on the selected labeled test data for 200 epochs. We first evaluate the median confidence of the model on the correctly classified and mis-classified test data respectively when fine-tuning the model on the selected labeled test data. In Figure 4, we show that during fine-tuning, the model gets increasingly confident not only on the correctly classified test data, but also on the mis-classified test data.

We then evaluate the Accuracy, the area under the receiver operator characteristic curve (AUROC) and the area under the accuracy-coverage curve (AUACC) metrics of the checkpoints during fine-tuning and the checkpoint ensemble constructed after fine-tuning on the target test dataset. The AUROC metric is equivalent to the probability that a randomly chosen correctly classified input has a higher confidence score

| Metric | Ensemble Test Accuracy | | | |
|---|---|---|---|---|
| Dataset | MNIST→SVHN | | DomainNet R→C (easy) | |
| Labeling Budget | 100 | 500 | 500 | 1000 |
| Round 0 | 24.67 | 24.87 | 37.33 | 37.46 |
| Round 1 | 24.91 | 43.80 | 39.61 | 39.67 |
| Round 2 | 37.75 | 54.91 | 41.15 | 41.55 |
| Round 3 | 45.62 | 64.15 | 41.97 | 43.24 |
| Round 4 | 50.94 | 71.65 | 42.57 | 45.09 |
| Round 5 | 56.75 | 77.23 | 43.85 | 45.62 |
| Round 6 | 59.82 | 79.97 | 44.20 | 46.60 |
| Round 7 | 63.10 | 81.43 | 45.02 | 47.51 |
| Round 8 | 67.49 | 82.78 | 45.17 | 48.59 |
| Round 9 | 69.93 | 84.70 | 45.80 | 48.66 |
| Round 10 | 71.14 | 85.48 | 46.36 | 49.70 |

Table 24: Ensemble test accuracy of ASPEST after each round of active learning. All numbers are percentages.

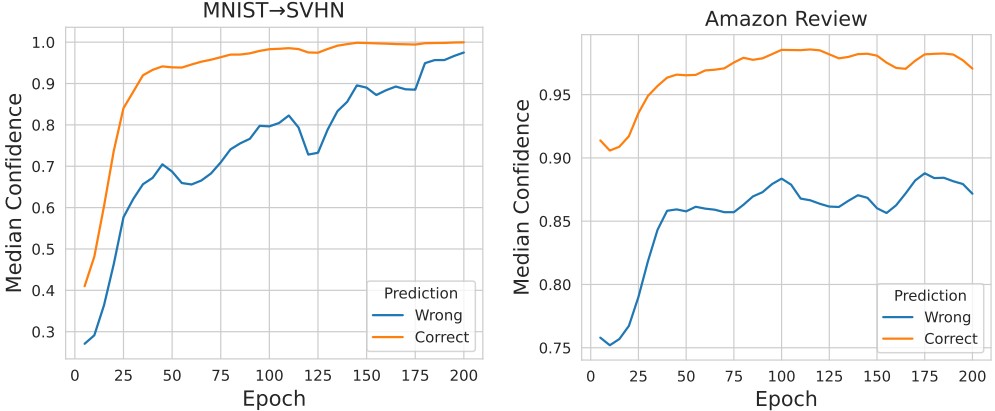

Figure 4: Evaluating the median confidence of the model on the correctly classified and mis-classified test data respectively when fine-tuning the model on the selected labeled test data.

than a randomly chosen mis-classified input. Thus, the AUROC metric can measure the quality of the confidence score for selective prediction. The results in Figure 5 show that in the fine-tuning path, different checkpoints have different target test accuracy and the end checkpoint may not have the optimal target test accuracy. The checkpoint ensemble can have better target test accuracy than the end checkpoint. Also, in the fine-tuning path, different checkpoints have different confidence estimation (the quality of confidence estimation is measured by the metric AUROC) on the target test data and the end checkpoint may not have the optimal confidence estimation. The checkpoint ensemble can have better confidence estimation than the end checkpoint. Furthermore, in the fine-tuning path, different checkpoints have different selective prediction performance (measured by the metric AUACC) on the target test data and the end checkpoint may not have the optimal selective prediction performance. The checkpoint ensemble can have better selective prediction performance than the end checkpoint.

## F.11 Empirical analysis for self-training

In this section, we analyze why the proposed self-training can improve selective prediction performance. Our hypothesis is that after fine-tuning the models on the selected labeled test data, the checkpoint ensemble constructed is less confident on the test data $U_X$ compared to the deep ensemble (obtained by ensembling the end checkpoints). Thus, using the softmax outputs of the checkpoint ensemble as soft pseudo-labels for self-training can alleviate the overconfidence issue and improve selective prediction performance.

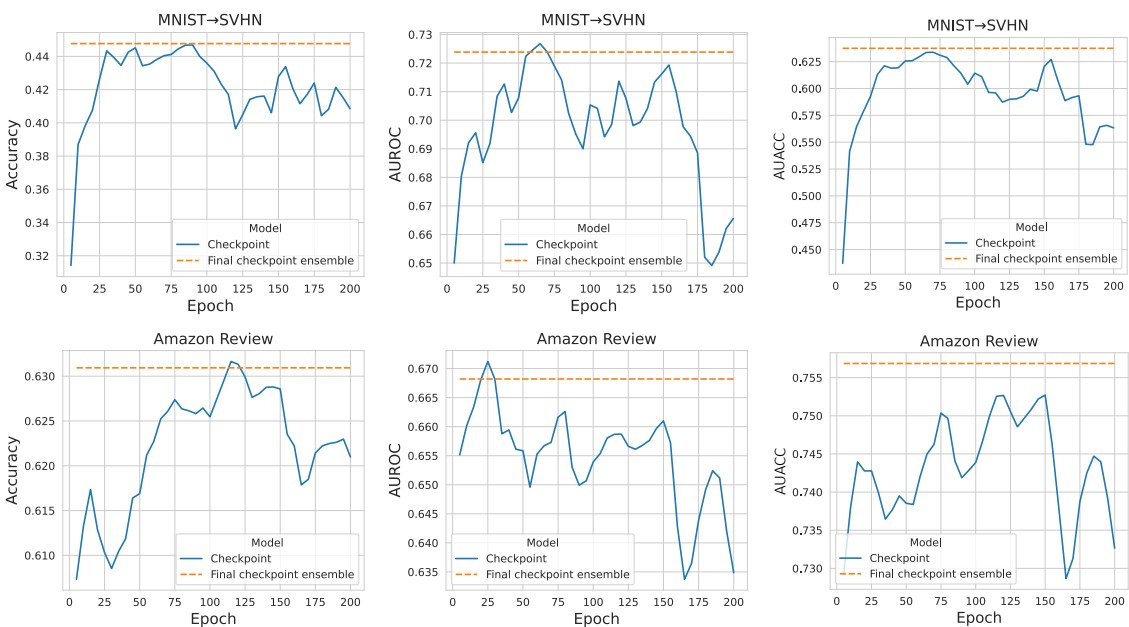

Figure 5: Evaluating the checkpoints during fine-tuning and the checkpoint ensemble constructed after fine-tuning on the target test dataset.

We perform experiments on the image dataset MNIST→SVHN and the text dataset Amazon Review to verity this hypothesis. To see the effect of self-training better, we only employ one-round active learning (i.e., only apply one-round self-training) with a labeling budget of 100 samples. We visualize the histogram of the confidence scores on the test data $U_X$ for the deep ensemble and the checkpoint ensemble after fine-tuning. We also evaluate the receiver operator characteristic curve (AUROC) and the area under the accuracy-coverage curve (AUACC) metrics of the checkpoint ensemble before and after the self-training. We use the AUROC metric to measure the quality of the confidence score for selective prediction. The results in Figure 6 show that the checkpoint ensemble is less confident on the test data $U_X$ compared to the deep ensemble. On the high-confidence region (i.e., confidence$\geq \eta$. Recall that $\eta$ is the confidence threshold used for constructing the pseudo-labeled set $R$. We set $\eta = 0.9$ in our experiments), the checkpoint ensemble is also less confident than the deep ensemble. Besides, the results in Table 25 show that after self-training, both AUROC and AUACC metrics of the checkpoint ensemble are improved significantly. Therefore, the self-training can alleviate the overconfidence issue and improve selective prediction performance.

| Dataset | MNIST→SVHN | | Amazon Review | |
|---|---|---|---|---|
| Metric | AUROC↑ | AUACC↑ | AUROC↑ | AUACC↑ |
| Before self-training | 73.92 | 66.75 | 67.44 | 76.24 |
| After self-training | 74.31 | 67.37 | 67.92 | 76.80 |

Table 25: Evaluating the AUROC and AUACC metrics of the checkpoint ensemble before and after self-training. All numbers are percentages.

## F.12 Training with unsupervised domain adaptation

In this section, we study whether incorporating Unsupervised Domain Adaptation (UDA) techniques into training could improve the selective prediction performance. UDA techniques are mainly proposed to adapt the representation learned on the labeled source domain data to the target domain with unlabeled data from the target domain (Liu et al., 2022). We can easily incorporate those UDA techniques into SR (Algorithm 1),

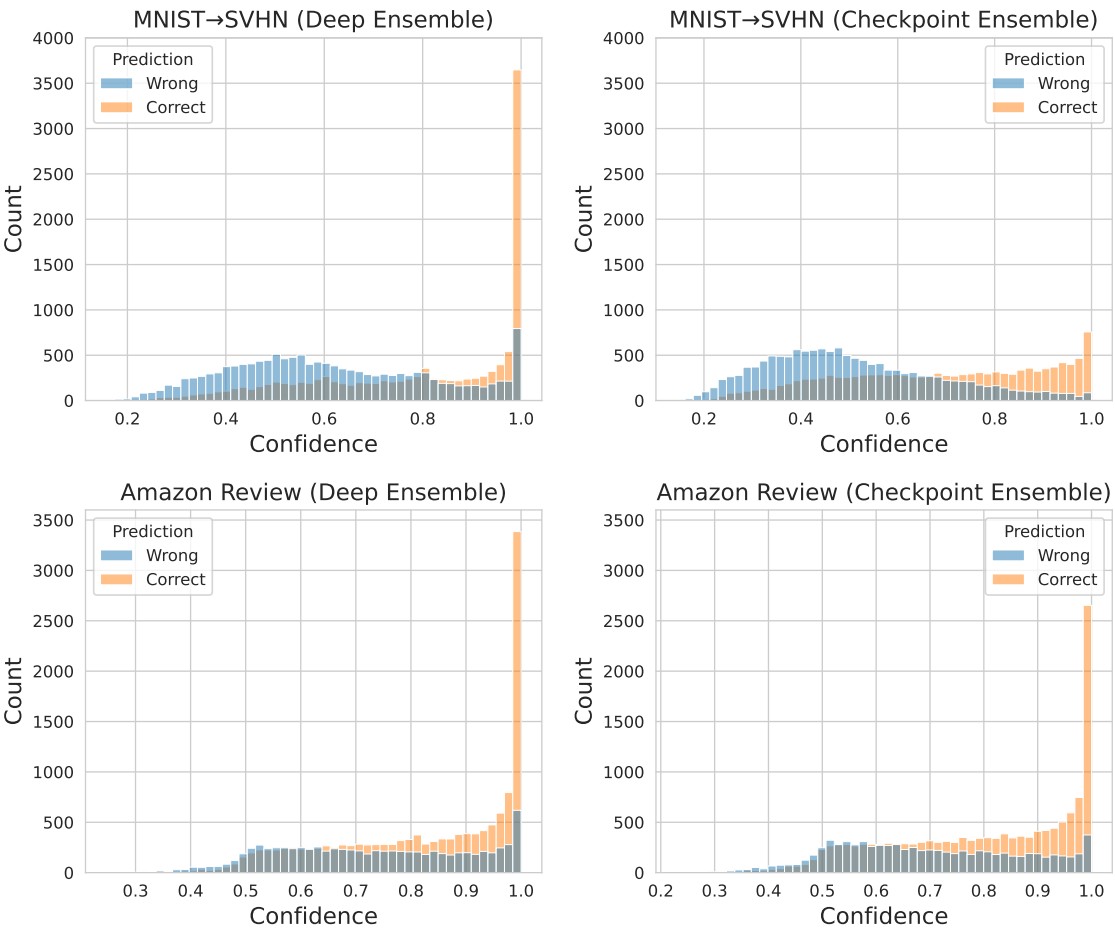

Figure 6: Plot the histogram of the confidence scores on the test data $U_X$ for the deep ensemble and the checkpoint ensemble after fine-tuning.

DE (Algorithm 2), and the proposed ASPEST (Algorithm 3) by adding unsupervised training losses into the training objectives.

We consider the method `DE with UDA` and the method `ASPEST with UDA`. The algorithm for DE with UDA is presented in Algorithm 4 and the algorithm for ASPEST with UDA is presented in Algorithm 5. We consider UDA techniques based on representation matching where the goal is to minimize the distance between the distribution of the representation on $\mathcal{D}^{\text{tr}}$ and that on $U_X$. Suppose the model $f$ is a composition of a prediction function $h$ and a representation function $\phi$ (i.e., $f(x) = h(\phi(x))$). Then $L_{UDA}(\mathcal{D}^{\text{tr}}, U_X; \theta) = d(p^{\phi}_{\mathcal{D}^{\text{tr}}}, p^{\phi}_{U_X})$, which is a representation matching loss. We consider the representation matching losses from the state-of-the-art UDA methods DANN (Ganin et al., 2016) and CDAN (Long et al., 2018).

We evaluate two instantiations of Algorithm 4 – `DE with DANN` and `DE with CDAN`, and two instantiations of Algorithm 5 – `ASPEST with DANN` and `ASPEST with CDAN`. The values of the hyper-parameters are the same as those described in the paper except that we set $n_s = 20$. For DANN and CDAN, we set the hyper-parameter between the source classifier and the domain discriminator to be 0.1. The results are shown in Table 26 (MNIST→SVHN), Table 27 (CIFAR-10→CINIC-10), Table 28 (FMoW), Table 29 (Amazon Review), Table 30 (DomainNet R→C), Table 31 (DomainNet R→P), Table 32 (DomainNet R→S) and Table 33 (Otto).

From the results, we can see that ASPEST outperforms (or on par with) `DE with DANN` and `DE with CDAN` across different datasets, although ASPEST doesn't use UDA techniques. We further show that by combining

ASPEST with UDA, it might achieve even better performance. For example, on MNIST→SVHN, `ASPEST with DANN` improves the mean AUACC from 96.62% to 97.03% when the labeling budget is 500. However, in some cases, combining ASPEST with DANN or CDAN leads to much worse results. For example, on MNIST→SVHN, when the labeling budget is 100, combining ASPEST with DANN or CDAN will reduce the mean AUACC by over 4%. It might be because in those cases, DANN or CDAN fails to align the representations between the source and target domains. Existing work also show that UDA methods may not have stable performance across different kinds of distribution shifts and sometimes they can even yield accuracy degradation (Johansson et al., 2019; Sagawa et al., 2021). So our findings align with those of existing work.

---

**Algorithm 4** DE with Unsupervised Domain Adaptation

---

**Input:** A training dataset $\mathcal{D}^{\mathrm{tr}}$, An unlabeled test dataset $U_X$, the number of rounds $T$, the total labeling budget $M$, a source-trained model $\bar{f}$, an acquisition function $a(B, f, g)$, the number of models in the ensemble $N$, the number of initial training epochs $n_s$, and a hyper-parameter $\lambda$.
Let $f_0^j = \bar{f}$ for $j = 1, \ldots, N$.
Fine-tune each model $f_0^j$ in the ensemble via SGD for $n_s$ training epochs independently using the following training objective with different randomness:

$$\min_{\theta^j} \quad \mathbb{E}_{(\mathbf{x}, y) \in \mathcal{D}^{\mathrm{tr}}} \quad \ell_{CE}(\mathbf{x}, y; \theta^j) + L_{UDA}(\mathcal{D}^{\mathrm{tr}}, U_X; \theta^j) \tag{35}$$

where $L_{UDA}$ is a loss function for unsupervised domain adaptation.
Let $B_0 = \emptyset$.
Let $g_t(\mathbf{x}) = \max_{k \in \mathcal{Y}} f_t(\mathbf{x} \mid k)$.
**for** $t = 1, \cdots, T$ **do**
    Select a batch $B_t$ with a size of $m = \lceil \frac{M}{T} \rceil$ from $U_X$ for labeling via:

$$B_t = \arg \max_{B \subset U_X \setminus (\cup_{l=0}^{t-1} B_l), |B|=m} a(B, f_{t-1}, g_{t-1}) \tag{36}$$

    Use an oracle to assign ground-truth labels to the examples in $B_t$ to get $\tilde{B}_t$.
    Fine-tune each model $f_{t-1}^j$ in the ensemble via SGD independently using the following training objective with different randomness:

$$\min_{\theta^j} \quad \mathbb{E}_{(\mathbf{x}, y) \in \cup_{l=1}^t \tilde{B}_l} \quad \ell_{CE}(\mathbf{x}, y; \theta^j) + \lambda \cdot \mathbb{E}_{(\mathbf{x}, y) \in \mathcal{D}^{\mathrm{tr}}} \quad \ell_{CE}(\mathbf{x}, y; \theta^j) + L_{UDA}(\mathcal{D}^{\mathrm{tr}}, U_X; \theta^j) \tag{37}$$

    where $\theta^j$ is the model parameters of $f_{t-1}^j$.
    Let $f_t^j = f_{t-1}^j$.
**end for**
**Output:** The classifier $f = f_T$ and the selection scoring function $g = \max_{k \in \mathcal{Y}} f(\mathbf{x} \mid k)$.

---

---

**Algorithm 5** ASPEST with Unsupervised Domain Adaptation

---

**Input:** A training set $\mathcal{D}^{tr}$, a unlabeled test set $U_X$, the number of rounds $T$, the labeling budget $M$, the number of models $N$, the number of initial training epochs $n_s$, a checkpoint epoch $c_e$, a threshold $\eta$, a sub-sampling fraction $p$, and a hyper-parameter $\lambda$.

Let $f_0^j = \bar{f}$ for $j = 1, \ldots, N$.

Set $N_e = 0$ and $P = \mathbf{0}_{n \times K}$.

Fine-tune each $f_0^j$ for $n_s$ training epochs using the following training objective:

$$\min_{\theta^j} \quad \mathbb{E}_{(\mathbf{x},y)\in\mathcal{D}^{tr}} \quad \ell_{CE}(\mathbf{x}, y; \theta^j) + L_{UDA}(\mathcal{D}^{tr}, U_X; \theta^j), \tag{38}$$

where $L_{UDA}$ is a loss function for unsupervised domain adaptation. During fine-tuning, update $P$ and $N_e$ using Eq. (6) every $c_e$ training epochs.

**for** $t = 1, \cdots, T$ **do**

    Select a batch $B_t$ from $U_X$ for labeling using the sample selection objective (7).

    Use an oracle to assign ground-truth labels to the examples in $B_t$ to get $\tilde{B}_t$.

    Set $N_e = 0$ and $P = \mathbf{0}_{n \times K}$.

    Fine-tune each $f_{t-1}^j$ using the following training objective:

$$\min_{\theta^j} \quad \mathbb{E}_{(\mathbf{x},y)\in\cup_{l=1}^t \tilde{B}_l} \quad \ell_{CE}(\mathbf{x}, y; \theta^j) + \lambda \cdot \mathbb{E}_{(\mathbf{x},y)\in\mathcal{D}^{tr}} \quad \ell_{CE}(\mathbf{x}, y; \theta^j) + L_{UDA}(\mathcal{D}^{tr}, U_X; \theta^j), \tag{39}$$

    During fine-tuning, update $P$ and $N_e$ using Eq (6) every $c_e$ training epochs.

    Let $f_t^j = f_{t-1}^j$.

    Construct the pseudo-labeled set $R$ via Eq (9) and create $R_{\text{sub}}$ by randomly sampling up to $\lceil p \cdot n \rceil$ data points from $R$.

    Train each $f_t^j$ further via SGD using the objective (10) and update $P$ and $N_e$ using Eq (6) every $c_e$ training epochs.

**end for**

**Output:** The classifier $f(\mathbf{x}_i) = \arg\max_{k\in\mathcal{Y}} P_{i,k}$ and the selection scoring function $g(\mathbf{x}_i) = \max_{k\in\mathcal{Y}} P_{i,k}$.

---

| Dataset | MNIST→SVHN | | | | | | | | |
|---|---|---|---|---|---|---|---|---|---|
| Metric | $cov\|acc \geq 90\%$ ↑ | | | $acc\|cov \geq 90\%$ ↑ | | | AUACC ↑ | | |
| Labeling Budget | 100 | 500 | 1000 | 100 | 500 | 1000 | 100 | 500 | 1000 |
| DE with DANN + Uniform | 27.27±1.8 | 72.78±2.0 | 87.05±0.5 | 63.95±1.4 | 82.99±0.8 | 88.64±0.2 | 80.37±0.7 | 93.25±0.4 | 96.05±0.1 |
| DE with DANN + Entropy | 11.33±8.2 | 74.04±2.2 | 91.06±1.4 | 58.28±2.1 | 83.64±0.9 | 90.41±0.5 | 74.62±1.6 | 93.45±0.5 | 96.47±0.2 |
| DE with DANN + Confidence | 15.68±6.3 | 76.34±3.1 | 93.96±1.2 | 61.32±3.0 | 85.02±0.9 | 91.64±0.4 | 76.43±3.0 | 93.85±0.6 | 96.97±0.3 |
| DE with DANN + Margin | 30.64±2.1 | 83.44±0.9 | 96.17±0.5 | 66.79±0.9 | 87.30±0.4 | 92.71±0.2 | 82.14±0.8 | 95.40±0.3 | 97.60±0.1 |
| DE with DANN + Avg-KLD | 22.30±3.0 | 78.13±2.1 | 93.42±1.0 | 63.22±2.0 | 85.40±0.8 | 91.47±0.5 | 78.88±1.6 | 94.25±0.5 | 97.02±0.2 |
| DE with DANN + CLUE | 16.42±13.6 | 72.27±2.8 | 86.71±0.4 | 61.79±2.7 | 82.72±1.1 | 88.46±0.2 | 77.47±3.4 | 93.33±0.5 | 96.21±0.0 |
| DE with DANN + BADGE | 25.41±10.9 | 78.83±1.2 | 90.94±1.1 | 63.93±4.4 | 85.27±0.5 | 90.45±0.5 | 79.82±4.1 | 94.58±0.3 | 96.89±0.1 |
| DE with CDAN + Uniform | 28.10±4.8 | 73.15±0.7 | 87.50±0.6 | 63.95±2.7 | 83.10±0.3 | 88.86±0.3 | 80.28±2.2 | 93.44±0.1 | 96.13±0.2 |
| DE with CDAN + Entropy | 6.94±9.8 | 74.38±1.5 | 90.77±1.3 | 59.90±2.3 | 84.14±0.4 | 90.32±0.6 | 76.04±2.0 | 93.48±0.3 | 96.38±0.2 |
| DE with CDAN + Confidence | 13.47±10.2 | 75.15±2.8 | 92.77±0.7 | 60.98±2.0 | 84.62±0.9 | 91.23±0.3 | 76.19±2.8 | 93.62±0.6 | 96.63±0.1 |
| DE with CDAN + Margin | 22.44±3.3 | 81.84±2.5 | 96.07±0.2 | 62.89±3.8 | 86.71±1.0 | 92.64±0.0 | 78.69±2.6 | 94.89±0.5 | 97.57±0.0 |
| DE with CDAN + Avg-KLD | 20.23±4.1 | 80.62±1.7 | 93.13±2.5 | 62.23±2.7 | 86.34±0.6 | 91.30±1.0 | 77.68±2.5 | 94.81±0.4 | 96.97±0.4 |
| DE with CDAN + CLUE | 7.47±6.4 | 72.61±2.9 | 87.22±0.2 | 57.82±2.9 | 82.50±1.3 | 88.62±0.1 | 73.33±2.3 | 93.38±0.7 | 96.31±0.0 |
| DE with CDAN + BADGE | 26.88±3.5 | 79.21±0.1 | 92.50±0.7 | 65.69±1.7 | 85.32±0.1 | 91.18±0.4 | 81.10±1.3 | 94.73±0.1 | 97.17±0.2 |
| ASPEST (ours) | **52.10**±4.0 | 89.22±0.9 | 98.70±0.4 | **76.10**±1.5 | 89.62±0.4 | 93.92±0.3 | **88.84**±1.0 | 96.62±0.2 | 98.06±0.1 |
| ASPEST with DANN (ours) | 37.90±2.4 | **91.61**±0.6 | 99.39±0.4 | 69.45±1.7 | **90.70**±0.3 | 94.42±0.4 | 84.55±1.0 | **97.03**±0.1 | 98.23±0.1 |
| ASPEST with CDAN (ours) | 30.97±11.7 | 91.39±0.6 | **99.50**±0.3 | 67.58±3.2 | 90.60±0.3 | **94.46**±0.2 | 82.20±3.3 | 96.95±0.1 | **98.26**±0.1 |

Table 26: Results of evaluating DE with UDA and ASPEST with UDA on MNIST→SVHN. The mean and std of each metric over three random runs are reported (mean±std). All numbers are percentages. **Bold** numbers are superior results.

| Dataset | CIFAR-10→CINIC-10 | | | | | | | | |
|---|---|---|---|---|---|---|---|---|---|
| Metric | $cov\|acc \geq 90\%$ ↑ | | | $acc\|cov \geq 90\%$ ↑ | | | AUACC ↑ | | |
| Labeling Budget | 500 | 1000 | 2000 | 500 | 1000 | 2000 | 500 | 1000 | 2000 |
| DE with DANN + Uniform | 58.85±0.3 | 59.39±0.2 | 60.04±0.1 | 77.06±0.2 | 77.33±0.2 | 77.84±0.1 | 90.40±0.1 | 90.60±0.1 | 90.73±0.1 |
| DE with DANN + Entropy | 59.42±0.4 | 60.86±0.3 | 64.52±0.3 | 78.14±0.2 | 79.20±0.1 | 81.31±0.1 | 90.72±0.0 | 91.06±0.1 | 92.02±0.0 |
| DE with DANN + Confidence | 59.44±0.6 | 61.08±0.3 | 65.12±0.2 | 78.19±0.1 | 79.38±0.0 | 81.29±0.1 | 90.73±0.1 | 91.26±0.1 | 92.06±0.0 |
| DE with DANN + Margin | 59.81±0.3 | 62.26±0.4 | 65.58±0.4 | 78.15±0.0 | 79.25±0.1 | 81.05±0.1 | 90.76±0.1 | 91.30±0.1 | 92.11±0.0 |
| DE with DANN + Avg-KLD | 60.50±0.5 | 62.04±0.1 | 65.08±0.2 | 78.32±0.1 | 79.31±0.1 | 81.07±0.0 | 90.89±0.1 | 91.34±0.0 | 92.11±0.0 |
| DE with DANN + CLUE | 60.20±0.5 | 61.69±0.2 | 64.08±0.2 | 77.84±0.2 | 78.35±0.2 | 79.38±0.1 | 90.73±0.2 | 91.07±0.1 | 91.63±0.0 |
| DE with DANN + BADGE | 60.18±0.4 | 62.15±0.2 | 65.31±0.6 | 77.70±0.1 | 78.54±0.1 | 79.81±0.2 | 90.72±0.1 | 91.19±0.1 | 91.86±0.1 |
| DE with CDAN + Uniform | 58.72±0.2 | 59.49±0.5 | 60.28±0.2 | 77.16±0.0 | 77.52±0.1 | 77.90±0.1 | 90.45±0.1 | 90.65±0.0 | 90.78±0.1 |
| DE with CDAN + Entropy | 58.73±0.4 | 60.82±0.5 | 64.45±0.2 | 77.95±0.1 | 79.20±0.1 | 81.04±0.1 | 90.57±0.1 | 91.10±0.1 | 91.86±0.1 |
| DE with CDAN + Confidence | 59.10±0.6 | 61.03±0.6 | 64.60±0.2 | 77.92±0.0 | 79.26±0.2 | 81.07±0.0 | 90.59±0.0 | 91.10±0.2 | 91.96±0.0 |
| DE with CDAN + Margin | 59.88±0.5 | 61.57±0.9 | 64.82±0.4 | 78.09±0.3 | 79.02±0.2 | 80.82±0.1 | 90.73±0.1 | 91.17±0.2 | 91.98±0.1 |
| DE with CDAN + Avg-KLD | 60.51±0.1 | 61.71±0.5 | 65.03±0.3 | 78.20±0.2 | 79.29±0.2 | 81.15±0.1 | 90.85±0.0 | 91.19±0.1 | 92.07±0.1 |
| DE with CDAN + CLUE | 60.12±0.5 | 61.77±0.3 | 64.06±0.2 | 77.88±0.1 | 78.38±0.2 | 79.42±0.2 | 90.73±0.0 | 91.08±0.1 | 91.64±0.0 |
| DE with CDAN + BADGE | 60.28±0.7 | 61.84±0.2 | 65.29±0.3 | 77.68±0.2 | 78.53±0.1 | 79.84±0.2 | 90.73±0.1 | 91.17±0.0 | 91.95±0.1 |
| ASPEST (ours) | 60.38±0.3 | 63.34±0.2 | 66.81±0.3 | 78.23±0.1 | 79.49±0.1 | 81.25±0.1 | 90.95±0.0 | 91.60±0.0 | 92.33±0.1 |
| ASPEST with DANN (ours) | **61.69**±0.2 | **63.58**±0.4 | **66.81**±0.4 | **78.68**±0.1 | **79.68**±0.1 | 81.42±0.1 | **91.16**±0.1 | **91.66**±0.1 | 92.37±0.1 |
| ASPEST with CDAN (ours) | 61.00±0.2 | 62.80±0.4 | 66.78±0.1 | 78.56±0.1 | 79.54±0.1 | **81.49**±0.0 | 91.13±0.0 | 91.57±0.1 | **92.41**±0.0 |

Table 27: Results of evaluating DE with UDA and ASPEST with UDA on CIFAR-10→CINIC-10. The mean and std of each metric over three random runs are reported (mean±std). All numbers are percentages. **Bold** numbers are superior results.

| Dataset | FMoW | | | | | | | | |
|---|---|---|---|---|---|---|---|---|---|
| Metric | $cov\|acc \geq 70\%$ ↑ | | | $acc\|cov \geq 70\%$ ↑ | | | AUACC ↑ | | |
| Labeling Budget | 500 | 1000 | 2000 | 500 | 1000 | 2000 | 500 | 1000 | 2000 |
| DE with DANN + Uniform | 46.11±0.6 | 51.77±0.3 | 62.76±0.5 | 57.62±0.3 | 60.67±0.4 | 66.21±0.2 | 70.17±0.3 | 72.46±0.3 | 76.83±0.2 |
| DE with DANN + Entropy | 44.36±0.7 | 48.19±0.3 | 59.52±0.8 | 56.78±0.1 | 59.51±0.0 | 65.75±0.3 | 69.09±0.2 | 71.02±0.2 | 75.15±0.3 |
| DE with DANN + Confidence | 44.46±0.5 | 49.32±0.1 | 61.47±0.3 | 57.04±0.3 | 60.51±0.3 | 66.61±0.1 | 69.14±0.1 | 71.50±0.1 | 75.70±0.1 |
| DE with DANN + Margin | 48.09±0.4 | 54.35±0.5 | 70.11±0.4 | 59.07±0.2 | 62.79±0.2 | 70.02±0.1 | 70.76±0.1 | 73.29±0.2 | 78.25±0.1 |
| DE with DANN + Avg-KLD | 48.42±0.1 | 55.95±0.2 | 68.73±1.1 | 59.06±0.2 | 63.44±0.2 | 69.41±0.5 | 70.84±0.1 | 73.83±0.1 | 77.91±0.4 |
| DE with DANN + CLUE | 44.14±0.6 | 46.15±0.2 | 49.02±0.5 | 56.01±0.3 | 56.89±0.2 | 58.66±0.3 | 69.11±0.2 | 70.16±0.2 | 71.46±0.2 |
| DE with DANN + BADGE | 48.57±0.5 | 54.47±0.5 | 67.69±0.9 | 58.61±0.2 | 61.67±0.0 | 68.71±0.5 | 71.17±0.2 | 73.64±0.1 | 78.65±0.3 |
| DE with CDAN + Uniform | 46.08±0.7 | 51.92±0.8 | 62.87±0.2 | 57.45±0.1 | 60.73±0.4 | 66.19±0.2 | 69.93±0.3 | 72.57±0.4 | 76.87±0.1 |
| DE with CDAN + Entropy | 44.42±0.3 | 49.32±0.1 | 60.11±0.3 | 56.83±0.1 | 60.04±0.2 | 65.95±0.2 | 69.18±0.2 | 71.34±0.3 | 75.44±0.3 |
| DE with CDAN + Confidence | 44.75±0.1 | 49.34±0.1 | 62.80±1.0 | 57.09±0.1 | 60.50±0.2 | 66.94±0.4 | 69.27±0.1 | 71.60±0.2 | 76.14±0.3 |
| DE with CDAN + Margin | 47.48±0.7 | 54.48±0.7 | 70.25±0.9 | 58.98±0.4 | 62.98±0.3 | 70.10±0.4 | 70.55±0.3 | 73.46±0.2 | 78.39±0.3 |
| DE with CDAN + Avg-KLD | 48.43±0.2 | 54.37±0.4 | 68.93±0.6 | 59.36±0.2 | 62.71±0.2 | 69.54±0.2 | 71.12±0.2 | 73.35±0.2 | 77.97±0.2 |
| DE with CDAN + CLUE | 44.09±0.3 | 46.11±0.5 | 48.90±0.1 | 55.78±0.3 | 56.98±0.2 | 58.46±0.2 | 69.03±0.1 | 70.02±0.2 | 71.31±0.1 |
| DE with CDAN + BADGE | 47.93±0.2 | 54.61±0.2 | 67.01±0.5 | 58.16±0.1 | 61.81±0.1 | 68.36±0.2 | 70.91±0.2 | 73.63±0.1 | 78.52±0.2 |
| ASPEST (ours) | **53.05**±0.4 | **59.86**±0.4 | **76.52**±0.6 | 61.18±0.2 | **65.18**±0.2 | **72.86**±0.3 | 71.12±0.2 | **74.25**±0.2 | **79.93**±0.1 |
| ASPEST with DANN (ours) | 51.02±0.9 | 58.63±1.1 | 72.97±0.9 | 61.10±0.5 | 64.98±0.4 | 71.21±0.4 | 71.03±0.3 | 73.79±0.4 | 77.84±0.3 |
| ASPEST with CDAN (ours) | 51.40±0.6 | 58.21±0.6 | 73.94±0.6 | **61.38**±0.2 | 65.04±0.2 | 71.63±0.2 | **71.17**±0.1 | 73.59±0.1 | 78.04±0.2 |

Table 28: Results of evaluating DE with UDA and ASPEST with UDA on FMoW. The mean and std of each metric over three random runs are reported (mean±std). All numbers are percentages. **Bold** numbers are superior results.

| Dataset | Amazon Review | | | | | | | | |
|---|---|---|---|---|---|---|---|---|---|
| Metric | $cov\|acc \geq 80\%$ ↑ | | | $acc\|cov \geq 80\%$ ↑ | | | AUACC ↑ | | |
| Labeling Budget | 500 | 1000 | 2000 | 500 | 1000 | 2000 | 500 | 1000 | 2000 |
| DE with DANN + Uniform | 38.55±3.3 | 37.25±1.8 | 39.21±1.9 | 69.06±0.6 | 68.94±0.1 | 69.41±0.2 | 77.52±0.7 | 77.03±0.4 | 77.70±0.2 |
| DE with DANN + Entropy | 38.22±2.3 | 41.85±0.8 | 41.57±1.3 | 69.48±0.3 | **70.71**±0.3 | 71.55±0.2 | 77.49±0.5 | 78.39±0.2 | 78.58±0.1 |
| DE with DANN + Confidence | 38.01±1.0 | 38.36±2.5 | 38.89±1.3 | 69.45±0.1 | 70.16±0.3 | 71.44±0.2 | 77.54±0.2 | 77.58±0.5 | 78.48±0.3 |
| DE with DANN + Margin | 36.82±1.3 | 36.89±1.3 | 41.98±1.5 | 69.35±0.3 | 69.63±0.3 | 71.27±0.2 | 77.30±0.3 | 77.23±0.3 | 78.34±0.3 |
| DE with DANN + Avg-KLD | 37.15±2.9 | 38.21±1.3 | 42.46±1.4 | 69.38±0.4 | 69.79±0.2 | 71.21±0.2 | 77.25±0.6 | 77.72±0.3 | 78.68±0.3 |
| DE with DANN + CLUE | **40.23**±4.0 | 34.71±1.8 | 31.38±0.9 | 68.95±0.7 | 68.07±0.2 | 67.44±0.3 | 77.62±1.0 | 76.27±0.6 | 75.60±0.2 |
| DE with DANN + BADGE | 37.51±1.8 | 37.00±0.9 | 41.62±2.3 | 68.98±0.4 | 69.27±0.1 | 70.20±0.4 | 77.20±0.4 | 77.21±0.1 | 78.31±0.5 |
| DE with CDAN + Uniform | 37.81±0.3 | 37.83±2.7 | 39.52±0.8 | 68.93±0.1 | 69.16±0.7 | 69.50±0.3 | 77.16±0.1 | 77.30±0.7 | 77.74±0.3 |
| DE with CDAN + Entropy | 37.99±0.8 | 37.68±1.1 | 42.55±0.9 | **69.54**±0.3 | 70.01±0.2 | 71.52±0.2 | 77.52±0.2 | 77.61±0.1 | 78.63±0.1 |
| DE with CDAN + Confidence | 35.76±0.9 | 38.69±2.8 | 41.43±2.1 | 69.24±0.0 | 70.45±0.4 | 71.50±0.4 | 77.08±0.2 | 77.82±0.4 | 78.47±0.3 |
| DE with CDAN + Margin | 37.68±2.9 | 37.43±1.0 | 42.18±1.3 | 69.50±0.3 | 69.80±0.4 | 71.29±0.0 | 77.50±0.5 | 77.31±0.3 | 78.46±0.3 |
| DE with CDAN + Avg-KLD | 37.85±1.6 | 40.71±0.9 | 44.35±0.9 | 69.41±0.3 | 70.29±0.1 | 71.28±0.2 | 77.28±0.5 | 78.11±0.2 | 78.86±0.2 |
| DE with CDAN + CLUE | 34.85±2.7 | 34.03±1.3 | 30.70±0.4 | 68.70±0.3 | 67.84±0.1 | 67.12±0.3 | 76.95±0.7 | 76.23±0.4 | 75.36±0.4 |
| DE with CDAN + BADGE | 39.47±0.2 | 39.29±1.1 | 41.64±0.9 | 69.33±0.0 | 69.34±0.2 | 70.58±0.2 | 77.52±0.2 | 77.49±0.2 | 78.24±0.3 |
| ASPEST (ours) | 38.44±0.7 | 40.96±0.8 | 45.77±0.1 | 69.31±0.3 | 70.17±0.2 | **71.60**±0.2 | 77.69±0.1 | 78.35±0.2 | **79.51**±0.2 |
| ASPEST with DANN (ours) | 40.22±0.5 | 41.99±1.4 | **45.84**±0.1 | 69.42±0.1 | 70.30±0.1 | 71.58±0.2 | **78.00**±0.1 | 78.34±0.3 | 79.43±0.1 |
| ASPEST with CDAN (ours) | 40.02±0.5 | **42.46**±0.6 | 44.95±0.4 | 69.50±0.1 | 70.37±0.2 | 71.42±0.0 | 77.80±0.1 | **78.57**±0.1 | 79.25±0.0 |

Table 29: Results of evaluating DE with UDA and ASPEST with UDA on Amazon Review. The mean and std of each metric over three random runs are reported (mean±std). All numbers are percentages. **Bold** numbers are superior results.

| Dataset | DomainNet R→C (easy) | | | | | | | | |
|---|---|---|---|---|---|---|---|---|---|
| Metric | $cov\|acc \geq 80\%$ ↑ | | | $acc\|cov \geq 80\%$ ↑ | | | AUACC ↑ | | |
| Labeling Budget | 500 | 1000 | 2000 | 500 | 1000 | 2000 | 500 | 1000 | 2000 |
| DE with DANN + Uniform | 33.53±0.5 | 36.28±0.3 | 40.13±1.0 | 50.57±0.5 | 52.19±0.1 | 55.15±0.1 | 69.34±0.3 | 70.98±0.2 | 73.50±0.3 |
| DE with DANN + Entropy | 28.66±1.0 | 34.47±0.1 | 42.77±0.7 | 48.13±0.6 | 52.70±0.3 | 59.01±0.2 | 66.60±0.5 | 70.64±0.1 | 75.45±0.2 |
| DE with DANN + Confidence | 29.92±0.4 | 35.29±1.0 | 43.33±0.4 | 48.61±0.1 | 53.36±0.5 | 59.72±0.3 | 67.23±0.2 | 70.92±0.5 | 75.89±0.3 |
| DE with DANN + Margin | 35.19±0.3 | 39.63±0.2 | 46.51±0.5 | 52.29±0.3 | 55.60±0.2 | 60.97±0.4 | 70.70±0.1 | 73.41±0.1 | 77.24±0.3 |
| DE with DANN + Avg-KLD | 36.02±0.6 | 39.67±0.5 | 47.20±0.8 | 53.00±0.3 | 55.75±0.3 | 61.22±0.3 | 71.19±0.3 | 73.51±0.2 | 77.46±0.2 |
| DE with DANN + CLUE | 32.26±1.5 | 35.09±0.4 | 35.66±0.3 | 50.21±0.0 | 50.90±0.1 | 51.50±0.1 | 69.17±0.2 | 70.20±0.2 | 70.82±0.1 |
| DE with DANN + BADGE | 35.27±0.5 | 38.88±0.3 | 45.97±0.7 | 52.15±0.3 | 54.89±0.1 | 60.03±0.3 | 70.65±0.1 | 72.95±0.1 | 76.87±0.1 |
| DE with CDAN + Uniform | 33.49±0.6 | 36.01±0.7 | 39.93±0.2 | 50.46±0.2 | 51.89±0.1 | 55.23±0.2 | 69.32±0.3 | 70.86±0.3 | 73.55±0.2 |
| DE with CDAN + Entropy | 29.50±0.5 | 33.86±0.3 | 42.24±0.5 | 48.01±0.1 | 52.52±0.3 | 58.96±0.2 | 66.82±0.2 | 70.28±0.1 | 75.33±0.1 |
| DE with CDAN + Confidence | 29.21±1.0 | 34.92±0.6 | 43.36±0.4 | 48.48±0.4 | 52.85±0.4 | 59.88±0.4 | 66.82±0.5 | 70.61±0.4 | 75.93±0.3 |
| DE with CDAN + Margin | 35.87±0.7 | 38.37±0.4 | 46.42±0.6 | 52.58±0.1 | 55.28±0.2 | 61.20±0.2 | 70.95±0.2 | 72.95±0.2 | 77.26±0.1 |
| DE with CDAN + Avg-KLD | 36.21±0.6 | 40.08±0.3 | 47.62±0.4 | 52.95±0.3 | 55.93±0.1 | 61.56±0.2 | 71.29±0.3 | 73.60±0.1 | 77.58±0.2 |
| DE with CDAN + CLUE | 31.74±2.1 | 35.11±0.2 | 35.87±0.5 | 49.99±0.2 | 51.39±0.2 | 51.43±0.2 | 69.04±0.3 | 70.35±0.0 | 70.82±0.3 |
| DE with CDAN + BADGE | 34.74±0.5 | 38.68±0.7 | 45.87±1.0 | 51.80±0.3 | 54.75±0.2 | 60.22±0.1 | 70.38±0.1 | 72.90±0.2 | 76.85±0.2 |
| ASPEST (ours) | 37.38±0.1 | 39.98±0.3 | 48.29±1.0 | 54.56±0.3 | 56.95±0.1 | 62.69±0.2 | 71.61±0.2 | 73.27±0.2 | 77.40±0.4 |
| ASPEST with DANN (ours) | **37.41**±0.8 | 42.45±1.0 | 49.74±0.6 | **55.60**±0.1 | 58.29±0.2 | 63.64±0.2 | 71.88±0.2 | 74.18±0.4 | 78.09±0.0 |
| ASPEST with CDAN (ours) | 36.60±1.2 | **42.96**±0.6 | **50.86**±0.2 | 55.55±0.2 | **58.71**±0.2 | **63.85**±0.2 | **71.99**±0.2 | **74.60**±0.2 | **78.45**±0.3 |

Table 30: Results of evaluating DE with UDA and ASPEST with UDA on DomainNet R→C. The mean and std of each metric over three random runs are reported (mean±std). All numbers are percentages. **Bold** numbers are superior results.

| Dataset | DomainNet R→P (medium) | | | | | | | | |
|---|---|---|---|---|---|---|---|---|---|
| Metric | $cov\|acc \geq 70\%$ ↑ | | | $acc\|cov \geq 70\%$ ↑ | | | AUACC ↑ | | |
| Labeling Budget | 500 | 1000 | 2000 | 500 | 1000 | 2000 | 500 | 1000 | 2000 |
| DE with DANN + Uniform | 26.98±0.1 | 28.34±0.5 | 30.63±0.2 | 41.96±0.2 | 42.89±0.2 | 44.73±0.1 | 57.04±0.1 | 58.10±0.2 | 59.87±0.1 |
| DE with DANN + Entropy | 24.75±0.4 | 27.02±0.5 | 30.10±0.2 | 40.29±0.4 | 42.34±0.2 | 45.78±0.2 | 55.19±0.3 | 57.12±0.3 | 60.21±0.1 |
| DE with DANN + Confidence | 22.41±0.9 | 27.03±0.6 | 31.70±0.6 | 39.05±0.5 | 42.61±0.2 | 46.60±0.2 | 53.66±0.6 | 57.35±0.3 | 60.93±0.4 |
| DE with DANN + Margin | 29.16±0.1 | 30.58±0.3 | 33.64±0.6 | 43.78±0.2 | 45.17±0.2 | 47.69±0.4 | 58.76±0.1 | 59.94±0.0 | 62.19±0.4 |
| DE with DANN + Avg-KLD | 29.52±0.1 | 31.17±0.4 | 34.09±0.3 | 43.84±0.3 | 45.33±0.2 | 48.18±0.2 | 58.89±0.2 | 60.25±0.2 | 62.54±0.2 |
| DE with DANN + CLUE | 27.48±0.5 | 27.83±0.2 | 28.39±0.5 | 42.05±0.3 | 42.34±0.2 | 42.65±0.1 | 57.32±0.3 | 57.64±0.2 | 57.99±0.2 |
| DE with DANN + BADGE | 28.92±0.1 | 30.36±0.2 | 33.86±0.3 | 43.38±0.1 | 44.85±0.1 | 47.64±0.3 | 58.38±0.0 | 59.82±0.1 | 62.26±0.2 |
| DE with CDAN + Uniform | 26.96±0.4 | 28.33±0.2 | 29.98±0.4 | 41.77±0.3 | 42.85±0.2 | 44.23±0.4 | 56.86±0.4 | 58.01±0.0 | 59.42±0.4 |
| DE with CDAN + Entropy | 24.91±0.4 | 26.30±0.9 | 30.33±0.4 | 40.34±0.3 | 42.07±0.6 | 45.79±0.2 | 55.38±0.4 | 56.70±0.8 | 60.23±0.2 |
| DE with CDAN + Confidence | 24.58±0.7 | 27.11±0.5 | 31.07±0.5 | 40.32±0.2 | 42.64±0.3 | 46.25±0.3 | 55.14±0.3 | 57.40±0.3 | 60.63±0.3 |
| DE with CDAN + Margin | 28.33±0.1 | 30.17±0.3 | 33.54±0.4 | 43.44±0.4 | 44.77±0.1 | 47.56±0.2 | 58.31±0.2 | 59.65±0.1 | 62.17±0.2 |
| DE with CDAN + Avg-KLD | 28.69±0.2 | 30.99±0.9 | 34.30±0.2 | 43.64±0.2 | 45.34±0.2 | 48.22±0.1 | 58.60±0.1 | 60.15±0.4 | 62.67±0.1 |
| DE with CDAN + CLUE | 27.52±0.6 | 27.96±0.2 | 28.18±0.5 | 42.02±0.2 | 42.44±0.1 | 42.67±0.2 | 57.21±0.3 | 57.70±0.1 | 58.04±0.3 |
| DE with CDAN + BADGE | 28.79±0.1 | 30.28±0.1 | 33.77±0.4 | 43.45±0.0 | 44.73±0.3 | 47.84±0.2 | 58.47±0.1 | 59.64±0.2 | 62.37±0.2 |
| ASPEST (ours) | 29.69±0.1 | 32.50±0.3 | 35.46±0.6 | 44.96±0.1 | 46.77±0.2 | 49.42±0.1 | 58.74±0.0 | 60.36±0.0 | 62.84±0.2 |
| ASPEST with DANN (ours) | **31.75**±0.4 | **33.58**±0.3 | 36.96±0.2 | **46.16**±0.1 | 47.64±0.2 | **50.37**±0.3 | **59.63**±0.2 | 61.06±0.1 | **63.75**±0.1 |
| ASPEST with CDAN (ours) | 30.39±0.4 | 33.57±0.3 | **37.53**±0.7 | 45.90±0.1 | **47.71**±0.2 | 50.31±0.2 | 59.13±0.3 | **61.17**±0.2 | 63.69±0.3 |

Table 31: Results of evaluating DE with UDA and ASPEST with UDA on DomainNet R→P. The mean and std of each metric over three random runs are reported (mean±std). All numbers are percentages. **Bold** numbers are superior results.

| Dataset | DomainNet R→S (hard) | | | | | | | | |
|---|---|---|---|---|---|---|---|---|---|
| Metric | $cov\|acc \geq 70\%$ ↑ | | | $acc\|cov \geq 70\%$ ↑ | | | AUACC ↑ | | |
| Labeling Budget | 500 | 1000 | 2000 | 500 | 1000 | 2000 | 500 | 1000 | 2000 |
| DE with DANN + Uniform | 17.55±0.4 | 19.82±0.3 | 23.57±0.4 | 32.61±0.5 | 34.56±0.3 | 37.73±0.2 | 47.60±0.5 | 49.92±0.4 | 53.52±0.1 |
| DE with DANN + Entropy | 10.77±0.8 | 15.38±0.5 | 20.11±0.5 | 27.78±0.7 | 31.09±0.2 | 36.39±0.3 | 41.69±0.7 | 45.62±0.3 | 51.05±0.4 |
| DE with DANN + Confidence | 10.64±1.2 | 15.22±0.4 | 20.25±0.5 | 28.09±1.0 | 31.76±0.3 | 36.86±0.8 | 41.94±1.3 | 46.19±0.3 | 51.48±0.7 |
| DE with DANN + Margin | 17.90±0.7 | 20.44±0.6 | 25.52±0.4 | 33.61±0.1 | 35.79±0.5 | 40.29±0.3 | 48.67±0.1 | 51.03±0.6 | 55.64±0.4 |
| DE with DANN + Avg-KLD | 18.02±1.0 | 21.22±0.2 | 25.46±0.2 | 34.00±0.2 | 36.51±0.2 | 40.72±0.2 | 49.05±0.2 | 51.79±0.2 | 55.95±0.2 |
| DE with DANN + CLUE | 15.77±0.3 | 18.14±0.7 | 19.49±0.4 | 32.10±0.1 | 33.42±0.3 | 34.50±0.3 | 47.18±0.2 | 48.63±0.3 | 50.03±0.3 |
| DE with DANN + BADGE | 16.84±0.9 | 20.88±0.3 | 25.11±0.3 | 33.97±0.1 | 36.20±0.2 | 40.01±0.3 | 48.87±0.2 | 51.46±0.2 | 55.33±0.2 |
| DE with CDAN + Uniform | 17.33±0.5 | 19.79±0.1 | 22.99±0.5 | 32.47±0.5 | 34.59±0.3 | 37.88±0.2 | 47.49±0.5 | 50.02±0.2 | 53.51±0.3 |
| DE with CDAN + Entropy | 12.48±0.8 | 15.19±0.8 | 20.23±0.0 | 28.83±0.1 | 32.41±0.4 | 36.57±0.1 | 42.93±0.5 | 47.00±0.3 | 51.24±0.2 |
| DE with CDAN + Confidence | 11.23±0.6 | 13.93±0.1 | 18.45±1.3 | 28.67±0.3 | 31.35±0.4 | 35.56±0.8 | 42.87±0.5 | 45.40±0.7 | 49.80±1.0 |
| DE with CDAN + Margin | 18.06±0.7 | 20.39±0.3 | 25.05±0.3 | 33.98±0.2 | 35.76±0.2 | 40.11±0.1 | 49.15±0.1 | 50.92±0.1 | 55.27±0.1 |
| DE with CDAN + Avg-KLD | 18.63±1.0 | 20.80±0.3 | 25.49±0.9 | 34.19±0.4 | 36.41±0.2 | 40.53±0.5 | 49.45±0.5 | 51.58±0.1 | 55.74±0.5 |
| DE with CDAN + CLUE | 16.51±0.3 | 18.82±0.1 | 19.47±0.1 | 32.23±0.2 | 33.83±0.4 | 34.72±0.3 | 47.40±0.2 | 49.11±0.2 | 49.98±0.3 |
| DE with CDAN + BADGE | 17.52±0.8 | 21.48±0.5 | 25.35±0.4 | 33.53±0.5 | 36.19±0.4 | 40.31±0.3 | 48.67±0.5 | 51.65±0.3 | 55.62±0.3 |
| ASPEST (ours) | 17.86±0.4 | 20.42±0.4 | 25.87±0.4 | 35.17±0.1 | 37.28±0.3 | 41.46±0.2 | 49.62±0.1 | 51.61±0.4 | 55.90±0.2 |
| ASPEST with DANN (ours) | 16.35±1.2 | **23.18**±0.4 | 28.00±0.1 | 36.56±0.2 | **39.40**±0.4 | 42.94±0.1 | 50.58±0.4 | **53.73**±0.3 | 57.25±0.1 |
| ASPEST with CDAN (ours) | **18.81**±1.1 | 22.95±0.8 | **28.17**±0.2 | **36.85**±0.3 | 39.10±0.2 | **43.25**±0.3 | **51.14**±0.3 | 53.47±0.2 | **57.26**±0.2 |

Table 32: Results of evaluating DE with UDA and ASPEST with UDA on DomainNet R→S. The mean and std of each metric over three random runs are reported (mean±std). All numbers are percentages. **Bold** numbers are superior results.

| Dataset | Otto | | | | | | | | |
|---|---|---|---|---|---|---|---|---|---|
| Metric | $cov\|acc \geq 80\%$ ↑ | | | $acc\|cov \geq 80\%$ ↑ | | | AUACC ↑ | | |
| Labeling Budget | 500 | 1000 | 2000 | 500 | 1000 | 2000 | 500 | 1000 | 2000 |
| DE with DANN + Uniform | 70.35±0.5 | 72.42±0.4 | 75.63±0.7 | 76.12±0.3 | 77.04±0.1 | 78.25±0.1 | 86.67±0.1 | 87.16±0.1 | 88.09±0.1 |
| DE with DANN + Entropy | 75.27±0.3 | 81.25±0.1 | 92.23±0.3 | 78.14±0.1 | 80.45±0.0 | 83.73±0.1 | 87.73±0.1 | 88.91±0.0 | 90.90±0.1 |
| DE with DANN + Confidence | 74.66±0.3 | 81.62±0.1 | 92.57±0.6 | 78.05±0.2 | 80.50±0.0 | 83.67±0.2 | 87.51±0.1 | 89.06±0.1 | 90.94±0.1 |
| DE with DANN + Margin | 75.47±0.4 | 82.56±0.7 | 91.86±0.9 | 78.26±0.1 | 80.79±0.2 | 83.61±0.3 | 87.87±0.1 | 89.08±0.0 | 90.88±0.1 |
| DE with DANN + Avg-KLD | 76.02±0.6 | 81.78±0.4 | 91.82±0.3 | 78.53±0.0 | 80.70±0.1 | 83.88±0.0 | 87.99±0.0 | 89.17±0.0 | 90.90±0.1 |
| DE with DANN + CLUE | 69.68±0.4 | 68.07±0.3 | 62.70±0.6 | 75.81±0.3 | 75.44±0.0 | 73.49±0.3 | 86.68±0.2 | 86.31±0.1 | 84.89±0.2 |
| DE with DANN + BADGE | 74.69±0.5 | 79.04±0.6 | 87.63±0.4 | 77.97±0.1 | 79.57±0.3 | 82.99±0.1 | 87.82±0.1 | 88.92±0.1 | 90.67±0.1 |
| DE with CDAN + Uniform | 70.25±0.9 | 72.43±0.4 | 75.21±0.7 | 76.09±0.3 | 76.94±0.1 | 78.13±0.1 | 86.56±0.3 | 87.14±0.2 | 87.90±0.1 |
| DE with CDAN + Entropy | 74.73±0.6 | 81.60±0.8 | 92.58±0.2 | 77.97±0.2 | 80.59±0.3 | 83.81±0.2 | 87.47±0.1 | 88.93±0.1 | 90.84±0.1 |
| DE with CDAN + Confidence | 74.88±0.6 | 81.30±0.8 | 92.53±0.9 | 78.06±0.2 | 80.51±0.3 | 83.85±0.3 | 87.43±0.2 | 88.99±0.1 | 90.95±0.1 |
| DE with CDAN + Margin | 76.68±1.0 | 81.57±0.4 | 92.20±0.5 | 78.74±0.5 | 80.62±0.2 | 84.01±0.2 | 88.08±0.2 | 88.85±0.2 | 91.09±0.0 |
| DE with CDAN + Avg-KLD | 75.88±0.4 | 81.82±0.8 | 91.43±1.1 | 78.45±0.1 | 80.72±0.3 | 83.72±0.3 | 87.92±0.2 | 89.12±0.2 | 90.91±0.2 |
| DE with CDAN + CLUE | 69.86±0.5 | 67.79±0.2 | 63.46±0.9 | 76.09±0.2 | 75.42±0.3 | 73.66±0.3 | 86.81±0.1 | 86.25±0.1 | 85.00±0.1 |
| DE with CDAN + BADGE | 74.68±0.4 | 79.46±0.3 | 87.57±0.4 | 77.89±0.1 | 79.78±0.1 | 82.85±0.1 | 87.78±0.1 | 88.90±0.1 | 90.72±0.1 |
| ASPEST (ours) | 77.85±0.2 | **84.20**±0.6 | 94.26±0.6 | 79.28±0.1 | **81.40**±0.1 | 84.62±0.1 | 88.28±0.1 | **89.61**±0.1 | **91.49**±0.0 |
| ASPEST with DANN (ours) | **78.14**±0.4 | 83.33±0.5 | 93.61±0.0 | **79.33**±0.1 | 81.23±0.1 | 84.21±0.1 | **88.36**±0.2 | 89.32±0.1 | 91.26±0.0 |
| ASPEST with CDAN (ours) | 77.75±0.3 | 83.68±0.5 | **94.44**±0.3 | 79.27±0.0 | 81.30±0.2 | **84.76**±0.1 | 88.35±0.1 | 89.59±0.0 | 91.41±0.0 |

Table 33: Results of evaluating DE with UDA and ASPEST with UDA on Otto. The mean and std of each metric over three random runs are reported (mean±std). All numbers are percentages. **Bold** numbers are superior results.

