# OpenReview forum: "ASPEST: Bridging the Gap Between Active Learning and Selective Prediction"
_TMLR — Accepted by TMLR_

### Review · Reviewer_HHqS · 2023-12-05

**Summary Of Contributions:**

The work presents ASPEST (Active Selective Prediction using Ensembles and Self-training). This method incorporates active learning with selective prediction to enhance accuracy and coverage in cases where the distribution of test data diverges from that of the training data (distribution shift). ASPEST queries highly informative samples from the shifted target domain, which significantly boosts the model accuracy and coverage breadth. Key to ASPEST's design is the use of ensemble model checkpoints, coupled with self-training techniques that leverage aggregated outputs as pseudo-labels.

**Audience:**

Yes

**Broader Impact Concerns:**

This work uses self-training, there is always a risk of inherent biases in the training data being perpetuated or amplified by the model and result in fairness issues. And such biases can be accumulated by increasing the rounds.

**Claims And Evidence:**

Yes

**Requested Changes:**

1. See the weaknesses part in the previous section.

2. Add a notion table.

3. Figure 1 could not become an overview of the proposed model, I cannot see how it works under the distribution shift data scenarios.

4. The evaluation metrics are a mess, sometimes it is ACC>90, ACC>80, and sometimes ACC>70.

5.  Is epochs=20 conduct a sufficient model training?

6. It is better to provide evaluation metrics vs. labeling budget curves for different methods instead of using tables.

7. The proposed method seems cannot give enough distinguishable model performance on standard datasets (without distribution shift problems), can add statistical tests.

**Strengths And Weaknesses:**

Strengths: The idea of this integration method looks interesting, a combination of different strategies, each addressing different aspects of the challenge. Self-training helps in adapting the model to new distributions, active learning efficiently expands the model's understanding of the new domain with limited labeled data, and selective prediction maintains prediction reliability in the face of uncertainty. An integrated approach, combining these elements, is often more effective in tackling distribution shifts in real-world scenarios.

Weaknesses:

1. This workflow induces too many hyperparameters. Although the authors do sensitive analysis on some hyperparameters, it is still not appropriate to introduce too many hyperparameters in the active learning process, since the validation dataset is insufficient, and the data scenarios vary.  The hyperparameters should be self-adaptive. For instance, in the joint training, in previous stages, the labeled data are insufficient, and the ensembles should be less reliable.

2. The use of ensembles and self-training might increase computational requirements. The authors only provide the running time of their methods, not comparison.

3. This work demonstrates effectiveness w.r.t. distribution shift on some specific datasets like MNIST and SVHN, it may not establish the generalizability of ASPEST across a more diverse range of domains and data types.

---

> ### Author Response · Authors · 2023-12-27
> **Response to Reviewer HHqS -- Part 1**
>
> - This workflow induces too many hyperparameters. Although the authors do sensitive analysis on some hyperparameters, it is still not appropriate to introduce too many hyperparameters in the active learning process, since the validation dataset is insufficient, and the data scenarios vary. Add a notion table.
>
> We appreciate the reviewer's concern regarding the number of hyperparameters in our ASPEST framework and their impact on the active learning process. To address this, we have now included a comprehensive table detailing all the hyperparameters used in ASPEST (see Table 2 in the paper).
>
> It's important to note that the majority of these hyperparameters are general training parameters, and as such, they would not significantly benefit from per-dataset tuning. This significantly reduces the complexity and the need for extensive validation datasets. Specifically, for hyperparameters $n_s$, $c_s$, $c_e$, and $p$, we employ fixed values based on empirical evidence, thus eliminating the need for their adjustment across different datasets.
>
> For other hyperparameters $\lambda$, $N$, $T$, and $\eta$, we perform tuning based on the performance observed on the DomainNet Real$\to$Infograph dataset. Specifically, we utilize the DomainNet Infograph subset, which comprises 15,582 images, as our validation set. This choice is driven by the fact that the Infograph subset is not part of our evaluation datasets, making it an ideal candidate for unbiased hyperparameter tuning. Once tuned, these values are kept consistent across all other datasets. This approach mirrors real-world scenarios where practitioners often rely on a single, representative validation dataset to tune key hyperparameters before applying the model to various datasets.
>
> This method of hyperparameter selection is not only realistic but also practical, ensuring that ASPEST remains adaptable and efficient across different data scenarios without requiring extensive tuning for each new dataset. We believe this approach strikes a balance between flexibility and robustness, addressing the concern of having an overly complex hyperparameter space in active learning.
>
> - The hyperparameters should be self-adaptive. For instance, in the joint training, in previous stages, the labeled data are insufficient, and the ensembles should be less reliable.
>
> Thank you for highlighting the importance of self-adaptive hyperparameters, especially in scenarios where labeled data are limited. Our experiments, indeed, cover cases with small labeling budgets (e.g., 100 or 500 samples), which are critical for evaluating the performance under constrained conditions.
>
> We found that even with these limited labeling budgets, ASPEST consistently outperforms the baseline methods. For instance, in the MNIST$\to$SVHN task, ASPEST improves the AUACC from 79.36% to 88.84% with just a 100-sample labeling budget. This significant improvement underlines the reliability and robustness of the ensembles in ASPEST, even when labeled data are scarce.
> While these results demonstrate the effectiveness of our current hyperparameter settings, we acknowledge the potential benefits of self-adaptive hyperparameters. They could further enhance the model's performance, particularly in the initial stages of training when labeled data are most limited. Therefore, we plan to explore adaptive hyperparameter tuning in future work to optimize the performance of ASPEST across various stages of the training process.
>
> This approach would not only address the concern of ensemble reliability in the face of insufficient labeled data but also potentially lead to more efficient and effective learning, especially in data-constrained environments.

---

> ### Author Response · Authors · 2023-12-27
> **Response to Reviewer HHqS -- Part 2**
>
> - The use of ensembles and self-training might increase computational requirements. The authors only provide the running time of their methods, not comparison.
>
> We acknowledge the reviewer's concern regarding the increased computational requirements associated with ensembles and self-training.  To provide a more standardized and fair comparison of computational costs, we have analyzed and compared the computational complexities of the proposed method and the baselines in Appendix D of our revised paper. The complexity of our proposed method, ASPEST, is expressed as $O\Big(N \cdot T \cdot \frac{n}{b} \cdot e_f \cdot t_g\Big)$, where $N$ represents the number of models in the ensemble, $T$ is the number of rounds in active learning, $n$ denotes the number of samples in the unlabeled test dataset, $b$ is the batch size, $e_f$ is the number of fine-tuning epochs, and $t_g$ is the complexity for one DNN gradient update step.
>
> It should be noted that ASPEST shares the same computational complexity as the DE+Margin baseline, which also has a computational complexity of $O\Big(N \cdot T \cdot \frac{n}{b} \cdot e_f \cdot t_g\Big)$. While both methods utilize an ensemble approach, a key distinction is that DE+Margin does not incorporate self-training, unlike ASPEST. In contrast, the SR+Margin baseline, which does not utilize ensembles, has a lower computational complexity of $O\Big(T \cdot \frac{n}{b} \cdot e_f \cdot t_g\Big)$.
>
> While DE+Margin and ASPEST indeed have higher computational complexities compared to SR+Margin due to the inclusion of ensembles, this does not necessarily translate to proportionally longer running times. With careful implementation and adequate computational resources, it is feasible to train ensemble models in parallel, potentially aligning the actual running times of ASPEST with those of the baseline methods, SR+Margin and DE+Margin. This parallelization strategy could mitigate the increased computational demand, making the running times of ASPEST comparable to, if not competitive with, the baselines under similar resource conditions.
>
> - This work demonstrates effectiveness w.r.t. distribution shift on some specific datasets like MNIST and SVHN, it may not establish the generalizability of ASPEST across a more diverse range of domains and data types.
>
> We appreciate the reviewer's concern regarding the generalizability of ASPEST across various domains and data types. To address this, our experimental setup was deliberately designed to encompass a broad spectrum of data categories and application domains.
> Specifically, we extended our evaluations beyond just image datasets like MNIST and SVHN. Our experiments encompass three distinct data types: image, text, and tabular data. This diverse selection is aimed at demonstrating the adaptability and effectiveness of ASPEST across different data modalities.
>
> For image data, we included well-known benchmarks such as CIFAR-10$\to$CINIC-10 and DomainNet for object classification, and satellite image classification datasets, which present unique challenges due to their complex spatial features. In the realm of text data, we conducted experiments using the Amazon Review dataset, which involves the classification of customer reviews—a task that entails handling natural language with its inherent variabilities. Additionally, for tabular data, we utilized the Otto Group Product Classification dataset, which requires the model to learn from structured data with multiple features.
>
> Each of these datasets represents a distinct domain with its own set of challenges: from digit classification in MNIST and SVHN to the more complex and varied tasks in datasets like DomainNet. This wide-ranging experimental setup was carefully chosen to robustly test ASPEST's performance and to establish its generalizability across different types of data and a variety of application domains.
>
> Through this comprehensive approach, we aim to demonstrate that ASPEST is not only effective in handling distribution shifts in specific cases but also possesses the versatility to adapt and perform consistently across a broader range of domains and data types.

---

> ### Author Response · Authors · 2023-12-27
> **Response to Reviewer HHqS -- Part 3**
>
> - The evaluation metrics are a mess, sometimes it is ACC>90, ACC>80, and sometimes ACC>70.
>
> We appreciate the reviewer's feedback on the clarity of our evaluation metrics. To ensure a comprehensive and fair assessment of our method, ASPEST, we employed three primary evaluation metrics, each with its specific purpose and applicability:
>
> 1. Maximum Accuracy at a Target Coverage ($t_c$): This metric measures the highest accuracy achievable at a pre-defined coverage level. The target coverage $t_c$ is tailored to each dataset, acknowledging the varying levels of difficulty inherent to different datasets. This customization allows for a more meaningful and context-specific evaluation.
>
> 2. Maximum Coverage at a Target Accuracy ($t_a$): Conversely, this metric gauges the maximum coverage attainable while maintaining a certain level of accuracy. Similar to $t_c$, the target accuracy $t_a$ is set according to the dataset's complexity. For more challenging datasets, where achieving high accuracy is difficult, lower target accuracy thresholds are set. Conversely, for datasets where models typically achieve higher accuracy, more stringent accuracy targets are established.
>
> 3. Area Under the Accuracy-Coverage Curve (AUACC): To provide a comprehensive and standardized comparison across different datasets and models, we utilize the AUACC metric. AUACC is independent of specific dataset difficulties and threshold variations, offering a uniform measure to evaluate and compare the performance of different methods.
>
> The variation in target accuracy and coverage thresholds for different datasets is intentional and reflective of their respective complexities. By adapting these thresholds to the dataset-specific challenges, we ensure that our evaluation is both rigorous and relevant to each dataset's context. The use of AUACC as a primary comparative metric further aids in standardizing our evaluation across diverse datasets, mitigating concerns regarding the inconsistency of threshold-based metrics.
>
> We hope this explanation clarifies our approach to selecting and applying these evaluation metrics, ensuring both dataset-specific relevance and methodological consistency across our analyses.
>
> - Is epochs=20 sufficient for model training?
>
> Yes, the 20 epochs for self-training are sufficient in our setup. This is because, in our active learning process, we fine-tune a source pre-trained model on the selected labeled test data for at least 50 epochs and up to 200 epochs, using a batch size of 128 and early stopping after 10 epochs without improvement. The initial pre-training on the source dataset and the fine-tuning on the selected labeled test data significantly reduces the need for extended training during the self-training phase, making 20 epochs of self-training adequate for this purpose.
>
> - It is better to provide evaluation metrics vs. labeling budget curves for different methods instead of using tables.
>
> We appreciate your suggestion on presenting our results. We opted for tables over curves to detail our findings due to their ability to convey precise numerical values and accommodate extensive results. Given the large number of baseline methods we analyzed, using curves would result in overly dense and potentially unclear visual representations, making it challenging to discern the performance differences between methods. Tables, in this context, provide a clearer and more direct comparison of the results across various methods.

---

> ### Author Response · Authors · 2023-12-27
> **Response to Reviewer HHqS -- Part 4**
>
> - It seems the proposed method can not give enough distinguishable model performance on standard datasets (without distribution shift problems), and can add statistical tests.
>
> We appreciate the reviewer's comments on our method's performance. Our study primarily targets datasets with distribution shifts, as standard datasets without such shifts typically exhibit satisfactory performance from models trained on the source distribution, reducing the need for adaptation. In scenarios with distribution shifts, ASPEST consistently outperforms baseline methods. For example, on MNIST$\to$SVHN, ASPEST improves AUACC from 79.36% to 88.84% with a labeling budget (M) of 100. Similarly, with M=500, ASPEST enhances AUACC on DomainNet R$\to$C from 68.85% to 71.61%, on Amazon Review from 76.63% to 77.69%, and on Otto from 87.89% to 88.28%. Each method was trained three times with varying random seeds, and we reported the mean and standard deviation for each metric. Our results indicate that the standard deviation is significantly smaller than the mean performance difference.
>
>
> -  Figure 1 could not become an overview of the proposed model. It is unclear how it works under the distribution shift data scenarios.
>
> We appreciate the reviewer's feedback regarding the clarity of Figure 1 in illustrating how our model addresses distribution shift scenarios. In response, we have revised Figure 1 by adding a new subfigure that explicitly depicts the distribution shift problem. This addition visually demonstrates how our proposed framework, *active selective prediction*, effectively operates under such scenarios.
>
> Furthermore, to enhance understanding, we have also updated the figure caption. The revised caption now clearly explains that the active selective prediction framework is specifically designed to tackle challenges posed by distribution shifts. It details how the model adapts to and manages these shifts, thereby providing a comprehensive overview of our approach in a visually intuitive manner.
>
> These modifications aim to clarify the functionality and significance of the active selective prediction framework in handling distribution shifts, ensuring that Figure 1 accurately and effectively conveys the essence of our proposed model.

---

> > ### Comment · Reviewer_HHqS · 2024-01-19
> > **response**
> >
> > I would appropriate the authors' responses, as it makes the paper clearer now, I have another question, In Table 5, it seems that $\eta$ is more sensitive when the labeling budget is small?

---

> > > ### Author Response · Authors · 2024-01-20
> > > **Response to Reviewer HHqS**
> > >
> > > We appreciate your feedback and are pleased to hear that our previous responses clarified the paper. Regarding your question about the sensitivity of $\eta$ under a small labeling budget, as noted in Table 5: yes, $\eta$ does exhibit increased sensitivity in such scenarios. For instance, this is evident when the labeling budget is limited to 100 in the MNIST$\to$SVHN case. Nevertheless, we find that selecting a sufficiently large value of $\eta$, such as 0.9, generally yields robust results, minimizing the need for extensive tuning of this parameter. The rationale behind this is that a higher $\eta$ value favors the selection of pseudo-labels with greater confidence. These labels are inherently more reliable for prediction accuracy and uncertainty estimation, thereby enhancing the efficacy of self-training and overall performance.

---

### Review · Reviewer_8iAq · 2023-12-12

**Summary Of Contributions:**

This paper proposes to tackle the active learning and selective prediction problems together in tandem.
They specifically investigate the setting of distribution shift where models trained on the source domain tend to be miscaliberated thus resulting in poor coverage of the selective predictor and resultant poor performance of the joint system.
The paper propose a combination of deep ensembling and self-training when leveraging labelled points queried from active learning, resulting in ASPEST which shows strong performance in the settings they investigate.

**Audience:**

Yes

**Claims And Evidence:**

Yes

**Requested Changes:**

1. Clarifications to Figure 2 as requested in above section
2. Positioning of the paper should be reconsidered, else a stronger case should be made for why active domain adaptation approaches  (along with an appropriately chosen threshold for selective prediction) cannot be used (or compared to) in the evaluation settings the paper presented.

**Strengths And Weaknesses:**

## Strengths
1. The method proposed by the authors -- ASPESTS -- produces strong results compared to reasonable baselines
2. The paper presents quite thorough experiments and ablations of the settings of interest.

## Weaknesses
Broadly, my primary concerns with this paper is that quite a few choices are not well motivated (at least from my read of the paper). There are several choices that I question :
1. Carving out active selective prediction as a niche, separate domain.  Specifically, the authors present *active selective prediction* as separate from active learning because the points are chosen not only to improve the model but also the selection scoring function. However, this strikes me as quite a cosmetic difference, since one can always re-define the utility function for active learning to fold in performance on the scoring function too. Thus the claim of **we introduce a new machine learning paradigm** seems overstated.
2. The motivation presented for why active domain adaptation (as a standalone approach) would not cover improvements in the selective predictor is unclear to me.  Specifically, the authors mention early in the intro that distribution shift can cause the selection predictor to fail due to miscalibration. It stands to reason that doing active domain adaption would fix the calibration problem (or if it doesn't, the authors provide no evidence of this), thus requiring minimal change to the selective predictor.
3. The authors state in section 3.1 **These two approaches to involve humans have different objectives and thus, their
joint optimization to best use the human labeling resources is not straightforward** -- however, it is unclear whether these two objectives are actually in opposition and not complementary. It would be good to either have an empirical or theoretical exploration of the compatibility of the two objectives.
4. Figure 2 was quite hard for me to understand. The distribution of already labelled data is not shown and thus makes it hard to understand why the decision bound changes the way the authors suggest when the marked points are selected. Also, the particular configuration shown (where diversity is the selection metric, and equal number of points are chosen from each cluster "where cluster is a group of points with the same latent label" but some clusters are bigger so one would apriori expect more points to be sampled from that cluster leading to a much different decision boundary change than the authors propose) seems contrived and I am unsure if the setup reflects the vast majority of typical problems.

---

> ### Author Response · Authors · 2023-12-27
> **Response to Reviewer 8iAq -- Part 1**
>
> - The claim of introducing a new machine learning paradigm seems overstated. One can always re-define the utility function for active learning to fold in performance on the scoring function too.
>
> We value the reviewer's insights regarding our claim of introducing a new machine learning paradigm. Our aim is to highlight the innovative integration of active learning and selective prediction, a fusion we believe is pioneering in this particular domain. Through our extensive analysis of existing methodologies in both active learning and selective prediction, we identified a notable performance disparity in scenarios demanding concurrent application of these strategies. This observation underscores a unique challenge, one not fully addressed by existing models.
>
> The essence of our contribution lies in the formulation of *active selective prediction*. This concept is not a mere amalgamation of two established methods, but a distinctive framework that thoughtfully combines the strengths of active learning and selective prediction. Our approach is characterized by the intricate design of the selection scoring function and the nuanced definition of the acquisition function within the active learning framework. These elements are key innovations of our work, representing more than straightforward extensions of existing techniques.
>
> We recognize that describing our approach as a new paradigm may appear ambitious. Nonetheless, we believe this terminology aptly conveys the originality and transformative potential of our work, particularly in light of its capacity to pioneer new research avenues. Our intention is for this work to serve as a foundational step in an evolving domain, stimulating further exploration and solidifying its place as a recognized paradigm in the machine learning landscape.
>
> That said, we are open to modifying our terminology if the reviewer maintains that our initial claim is overstated. We propose alternatively describing our work as a “novel integration” or an “innovative framework” within machine learning, should that better align with the reviewer's perspective.
>
> - Active domain adaptation would fix the calibration problem (or if it doesn't, the authors provide no evidence of this), thus requiring minimal change to the selective predictor.
>
> Thank you for pointing out the potential role of active domain adaptation in addressing the calibration problem within the context of selective prediction. To investigate this, we specifically included an evaluation of the active domain adaptation method CLUE [1], with results presented in Table 3 of our paper.
>
> Our experimental results reveal a critical insight: when the active domain adaptation method CLUE is employed in conjunction with Softmax Response (SR) or Deep Ensembles (DE) for constructing a selective predictor, it falls short in tackling the calibration problem. This limitation becomes clear upon examining the comparative performance metrics. Both baselines, CLUE+SR and CLUE+DE, demonstrate suboptimal performance when compared to our proposed method, ASPEST, in the same experimental setup. This outcome strongly indicates that the integration of CLUE with these selective prediction techniques does not adequately address the intricacies of calibration challenges within the active selective prediction framework.
>
> This outcome suggests that a naïve application of active domain adaptation methods like CLUE, although valuable, does not inherently resolve the calibration challenge posed in active selective prediction scenarios. Our findings indicate that the calibration issue, in this case, necessitates a more nuanced approach than what standard active domain adaptation methods can offer.
>
> Therefore, our proposed method ASPEST does not merely represent an incremental change to existing selective predictors but addresses a more complex interplay of factors unique to the active selective prediction setting. The empirical evidence from our experiments underscores the necessity and effectiveness of our approach in overcoming these specific challenges.
>
> [1] Prabhu, Viraj, et al. "Active domain adaptation via clustering uncertainty-weighted embeddings." Proceedings of the IEEE/CVF International Conference on Computer Vision. 2021.

---

> ### Author Response · Authors · 2023-12-27
> **Response to Reviewer 8iAq -- Part 2**
>
> - The authors state in section 3.1 These two approaches to involve humans have different objectives and thus, their joint optimization to best use the human labeling resources is not straightforward -- however, it is unclear whether these two objectives are actually in opposition and not complementary. It would be good to either have an empirical or theoretical exploration of the compatibility of the two objectives.
>
> We appreciate the opportunity to clarify the relationship between the objectives of active learning and selective prediction, as highlighted in section 3.1. While these two approaches have distinct goals, we recognize that they are neither inherently complementary nor oppositional in nature.
>
> Active learning primarily focuses on selecting samples for labeling to enhance the model's accuracy. However, it does not inherently aim to optimize the selection score (e.g., model confidence), which is crucial for effective selective prediction. This divergence in objectives can lead to scenarios where active learning choices, although beneficial for accuracy improvement, may adversely affect selective prediction performance. For example, the insights we gleaned from our experiments, as detailed in Appendix F.2, provide a concrete illustration of this concept. In these experiments, we employed methods based on uncertainty sampling, such as SR+Confidence, SR+Entropy, and SR+Margin. While these methods effectively enhanced the model's accuracy, they also led to an overconfidence issue, characterized by misclassifications made with high confidence. This presents a significant challenge for selective prediction: the high-confidence yet incorrectly classified samples are not aptly identified and rejected by the selective predictor. As a result, the performance of selective prediction is compromised, demonstrating its suboptimality in scenarios where the active learning component contributes to overconfidence in predictions.
>
> Conversely, selective prediction is designed to reject the samples that are expected to be incorrect, deferring their prediction to human judgment. However, the samples deemed unreliable by selective prediction may not align with those that would be most beneficial for active learning. Particularly under conditions of distribution shifts, where model accuracy on unlabeled test data is typically lower, an optimal selective prediction approach would defer many predictions on misclassified samples to humans. This leads to a significant demand for human labeling resources.
>
> The concept of active selective prediction emerges from the potential synergistic optimization of these two approaches. By judiciously integrating active learning and selective prediction, we aim to minimize the demand for human labeling resources. Active learning in this context is tailored to select a minimal yet impactful set of samples for labeling, which in turn informs the development of a highly accurate selective predictor. This predictor effectively rejects misclassified samples, deferring them for human evaluation. As a result of this optimal integration, the number of samples requiring human intervention is significantly reduced, thereby enhancing the overall efficiency and resource utilization.
>
> Through this explanation, we hope to elucidate that while active learning and selective prediction have different focal points, their joint optimization in the context of active selective prediction presents a unique and beneficial approach, significantly reducing the reliance on human labeling resources.

---

> ### Author Response · Authors · 2023-12-27
> **Response to Reviewer 8iAq -- Part 3**
>
> - Difficulties in understanding Figure 2.
>
> Thank you for your valuable feedback regarding Figure 2. We acknowledge that the figure, as presented, may not fully convey the complexities involved in real-world data and multi-round active learning scenarios. The figure is intended primarily for illustrative purposes to provide a simplified conceptual understanding of the active learning process in a single-round setup, without the inclusion of already labeled data.  The depicted current decision boundary is derived from labeled source training data. Our primary focus, however, is on assessing performance with respect to the unlabeled target test data, where our evaluation metrics are applied, and this is the reason for omitting source training data from this illustration. This simplification was chosen for clarity, but we recognize that it may not capture the full intricacy of real-world decision boundaries.
>
> In Figure 2(b), we illustrate the sample selection process using the k-Center-Greedy algorithm [2], which is a diversity-focused approach within the realm of active learning. The primary objective of this depiction is to demonstrate a potential pitfall of conventional diversity-based active learning methods. Specifically, we aim to highlight how these methods, while effective in diversifying the sample pool, can inadvertently contribute to the issue of low accuracy. This leads to rejection of many points, necessitating significant human intervention. Our intent with Figure 2(b) is to showcase this specific challenge, underscoring the need for more nuanced approaches in active learning that can mitigate such issues.
>
> We understand that this representation may seem somewhat contrived and might not reflect the more complex dynamics of typical problems. However, our aim was to use this simplified model to highlight specific challenges in a clear and accessible way. In practice, active learning, particularly in multiple rounds with existing labeled data, would result in more nuanced and complex decision boundary changes.
>
> We hope this clarification helps to better contextualize the purpose and limitations of Figure 2. To ensure comprehensive understanding for our readers, we have incorporated this explanation into the caption of Figure 2 in the revised version of our paper.
>
>
> [2] Sener, Ozan, and Silvio Savarese. "Active learning for convolutional neural networks: A core-set approach." arXiv preprint arXiv:1708.00489 (2017).

---

### Review · Reviewer_FWsG · 2023-12-20

**Summary Of Contributions:**

This paper introduces a new learning paradigm called active selective prediction that combines selective prediction (not predicting on uncertain points) and active learning (deciding which datapoints to obtain human labels for). The authors' new framework aims to query more informative sample that the model is uncertain on, and propose an approach called ASPEST that utilizes ensembles of model snapshots with self-training with their aggregated outputs as pseudo labels. They show that this approach is quite effective on several datasets for selective prediction, active learning, and distribution shifts.

**Audience:**

Yes

**Broader Impact Concerns:**

no concerns

**Claims And Evidence:**

No

**Requested Changes:**

see above weaknesses on real-world setting where we see this problem setting, new experimental results on this setting rather than artificially paired transfer datasets, and issues regarding baselines

**Strengths And Weaknesses:**

The paper is well-written, methods seems solid, experiments are shown on a wide range of datasets, and improvements are consistent. There is both theoretical and empirical analysis of the proposed problem setting and method. Results are carefully studied and analyzed.

My biggest issue with the paper is that I don't know if the problem is actually something we see in the real-world. The experiments seem to be on synthetic shifts from one dataset to another. It would be much more compelling if there is a unique real-world task where this problem setting and method is actually required, or performance would be close to 0% or random. This would significantly improve the paper and address my concerns.

Also, I am not an expert in this area, but I am curious whether the baselines quoted and compared in the paper are sufficient, or are they designed to trivially fail (i.e baselines for active learning combined with separate baselines for selective prediction, which obviously fail when applied to this new problem combining both challenges). If so, then there should be a much more significant effort towards testing and analyzing strong baselines for this new problem.

---

> ### Author Response · Authors · 2023-12-27
> **Response to Reviewer FWsG -- Part 1**
>
> - Concern about the practical relevance of the research, focusing on synthetic data.
>
> Thank you for raising this critical point about the practical applicability of our work. We concur that demonstrating real-world relevance is vital for the impact of our research. To this end, our work on active selective prediction is directly inspired by and applicable to genuine challenges faced in deploying deep learning models in dynamic real-world environments.
>
> In real-world settings, it's common for deployed models to encounter test data that significantly diverge from the training data distribution, leading to performance degradation. Our proposed approach, active selective prediction, is designed to address this precise issue of distribution shift, which is a pervasive and critical challenge in practical applications.
>
> To substantiate the real-world applicability of our method, we have conducted experiments not only on synthetic data but also on diverse real-world datasets, where distribution shifts are inherent and not artificially constructed. For instance, our experiments on the FMoW-WILDS dataset [1], derived from the Functional Map of the World (FMoW) dataset [2], showcase the challenges of dealing with temporal and geographical distribution shifts in high-resolution satellite imagery spanning a 16-year period across various continents.
> Similarly, our experiments on the Amazon Review dataset [3], drawn from a vast corpus of customer reviews on Amazon, illustrate the issue of distribution shifts stemming from the diverse writing styles and perspectives of different reviewers.
>
> These datasets have been specifically chosen for their origin in real-world scenarios, reflecting the complexities and variabilities inherent in practical applications. Each dataset represents a different aspect of real-world challenges, capturing diverse, naturally occurring distribution shifts. These datasets are not artificially contrived but are derived from genuine, real-world sources, thereby providing an authentic and practical framework for testing and demonstrating the effectiveness of our approach in handling distribution shifts as they naturally occur in the real world.
>
> In these real-world scenarios, our experimental setup involves training models on source domain data and evaluating their performance on target domain data, where a notable performance degradation is observed due to distribution shifts (please see Table 6 in Appendix F.1 for detailed results on performance degradation due to distribution shifts). The active selective prediction strategy, as embodied by our proposed ASPEST method, actively adapts the source-trained model to the target domain using principles of active learning, while simultaneously applying selective prediction to ensure reliable and robust predictions in the target domain.
>
> Our experimental results on numerous real-world image, text and structured datasets, which suffer from distribution shifts, demonstrated that the proposed method ASPEST significantly outperforms the baselines. For example, with a labeling budget (M) of 500, ASPEST enhances the AUACC on FMoW from 70.59% to 71.12% and on Amazon Review from 76.63% to 77.69%. The promising results we have obtained with ASPEST on these real-world datasets not only demonstrate its efficacy but also its potential for practical implementation in addressing real-world distribution shift challenges. While the deployment of ASPEST in actual real-world applications remains a part of our future work, we believe that the current evidence strongly supports its viability and necessity in practical scenarios.
>
> [1] Koh, Pang Wei, et al. "Wilds: A benchmark of in-the-wild distribution shifts." International Conference on Machine Learning. PMLR, 2021.
>
> [2] Christie, Gordon, et al. "Functional map of the world." Proceedings of the IEEE Conference on Computer Vision and Pattern Recognition. 2018.
>
> [3] Ni, Jianmo, Jiacheng Li, and Julian McAuley. "Justifying recommendations using distantly-labeled reviews and fine-grained aspects." Proceedings of the 2019 conference on empirical methods in natural language processing and the 9th international joint conference on natural language processing (EMNLP-IJCNLP). 2019.

---

> ### Author Response · Authors · 2023-12-27
> **Response to Reviewer FWsG -- Part 2**
>
> - Whether the baselines quoted and compared in the paper are sufficient, or are they designed to trivially fail.
>
> We appreciate the reviewer's concern regarding the adequacy and fairness of the baselines used in our study. It is important to clarify that our work pioneers the exploration of the active selective prediction problem, a domain where, to our knowledge, no specific methods have previously been established. In light of this, we crafted our baselines by integrating state-of-the-art methods from closely related fields: active learning and selective prediction. Our intention was to establish a robust and relevant foundation for comparison, not to set up baselines that are predisposed to underperform.
>
> Furthermore, to reinforce the strength and relevance of our baselines, we incorporated advanced techniques such as the active domain adaptation method CLUE [4], and the unsupervised domain adaptation methods DANN [5] and CDAN [6]. These were chosen to ensure that the baselines represent a meaningful and challenging benchmark for our proposed ASPEST method.
>
> The observed underperformance of these baselines in the active selective prediction setup is not a reflection of their triviality, but rather an indication of the unique challenges posed by this new problem. This underperformance underscores the necessity for innovative approaches, like ASPEST, specifically tailored to the active selective prediction problem.
>
> We believe that our efforts in testing and analyzing these strong baselines provide a comprehensive foundation for evaluating the novelty and effectiveness of ASPEST. Our approach demonstrates not only the feasibility of solving this new problem but also sets a precedent for future research in this area.
>
> [4] Prabhu, Viraj, et al. "Active domain adaptation via clustering uncertainty-weighted embeddings." Proceedings of the IEEE/CVF International Conference on Computer Vision. 2021.
>
> [5] Ganin, Yaroslav, et al. "Domain-adversarial training of neural networks." Journal of machine learning research 17.59 (2016): 1-35.
>
> [6] Long, Mingsheng, et al. "Conditional adversarial domain adaptation." Advances in neural information processing systems 31 (2018).

---

### Author Response · Authors · 2023-12-27
**Rebuttal and Paper Revision Summary**

We extend our gratitude to all reviewers for their insightful feedback. Based on your comments, we have thoroughly revised our paper. Revisions are highlighted in blue in the text for easy identification. Below is a summary of the key revisions and our responses to the reviewers’ concerns:

- **For Reviewer FWsG**:
  - **Real-World Applicability**: We emphasized that our work is inspired by real-world challenges, detailing experiments on diverse real-world datasets experiencing inherent distribution shifts. These clarifications are now included in Section 5.1 and Appendix E.2.
  - **Baseline Comparison**: We highlighted the establishment of strong baselines for the novel active selective prediction problem, underscoring the robustness of our comparative analysis.

- **For Reviewer 8iAq**:
  - **Machine Learning Paradigm**: We defended our claim of introducing a new paradigm in machine learning, expressing openness to modify this claim if it is still perceived as overstated.
  - **Active Domain Adaptation and Calibration**: We clarified that existing active domain adaptation methods do not resolve the calibration problem in active selective prediction.
  - **Objectives of Active Learning and Selective Prediction**: We elaborated on how these objectives are neither inherently complementary nor oppositional and described our integrated approach. Modifications are made in Section 3.1.
  - **Figure 2**: Improved the caption of Figure 2 for enhanced clarity and understanding.

- **For Reviewer HHqS**:
  - **Hyperparameters**: Included a detailed table (Table 2) explaining the hyperparameters used in ASPEST and the rationale behind their selection in Section 5.1.
  - **Computational Complexity**: Analyzed and compared the computational complexity of ASPEST with baseline methods in Appendix D.
  - **Performance Across Domains**: Showcased ASPEST's superior performance over baselines across various domains and data types.
  - **Evaluation Metrics**: Added a paragraph in Section 3.2 clarifying the evaluation metrics used in our study.
  - **Significance of results**: Confirmed that our study focuses on datasets with distribution shifts and highlighted the significant improvements achieved by ASPEST.
  - **Figure 1 Revision**: Revised Figure 1, adding a subfigure explicitly depicting the distribution shift problem.

We welcome any further questions or concerns regarding our paper and remain committed to contributing valuable insights to the field.

---

### Decision · Action_Editor_zSPc · 2024-02-05

**Recommendation:** Accept as is

**Comment:**

Although not all the reviewers view the problem of "active selective prediction" as novel necessarily, all the reviewers are in agreement with respect to the manuscript - leaning accept. All requested changes were either made or discussed to satisfaction of the reviewers. The extensive experiments clearly show the improved performance of the proposed approach, ASPEST, as compared to baseline methods. The authors made edits to the manuscript to clarify any points of confusion. Hence, the recommendation is to accept the paper as is.

**Audience:**

Safe handling of out-of-distribution scenarios is of interest to the community. As such, the proposed active selective prediction paradigm should be interesting to TMLR's audience.

**Claims And Evidence:**

The authors present a new formulation, active selective prediction, which balances active learning towards covering more of the out-of-distribution (OOD) spaces and selective prediction which avoids making predictions on uncertain inputs. As such, active selective prediction aims towards higher coverage of the OOD space while maintaining high accuracy. The authors propose ASPEST - a method that involves selecting samples for labeling based on uncertainty to achieve high accuracy (active learning) and using checkpoint ensembles and self-training to alleviate overconfidence. The reviewers found the paper well presented and well supported both theoretically and empirically, with extensive experiments. Most of the requested changes and questions, which the authors have implemented and answered, were regarding clarification of the method or the experimental setup.